# The 3D enhancer network of the developing T cell genome is shaped by SATB1

Tomas Zelenka [1,2,6], Antonios Klonizakis[1], Despina Tsoukatou[2], Dionysios-Alexandros Papamatheakis [1,2], Sören Franzenburg [3], Petros Tzerpos[1,7], Ioannis-Rafail Tzonevrakis [1], George Papadogkonas[1,2], Manouela Kapsetaki [2], Christoforos Nikolaou[1,2,8], Dariusz Plewczynski [4,5] & Charalampos Spilianakis [1,2] ✉

Mechanisms of tissue-specific gene expression regulation via 3D genome organization are poorly understood. Here we uncover the regulatory chromatin network of developing T cells and identify SATB1, a tissue-specific genome organizer, enriched at the anchors of promoter-enhancer loops. We have generated a T-cell specific *Satb1* conditional knockout mouse which allows us to infer the molecular mechanisms responsible for the deregulation of its immune system. H3K27ac HiChIP and Hi-C experiments indicate that SATB1-dependent promoter-enhancer loops regulate expression of master regulator genes (such as *Bcl6*), the T cell receptor locus and adhesion molecule genes, collectively being critical for cell lineage specification and immune system homeostasis. SATB1-dependent regulatory chromatin loops represent a more refined layer of genome organization built upon a high-order scaffold provided by CTCF and other factors. Overall, our findings unravel the function of a tissue-specific factor that controls transcription programs, via spatial chromatin arrangements complementary to the chromatin structure imposed by ubiquitously expressed genome organizers.

In order to store the large amount of genetic information, higher eukaryotes developed spatial and functional genome organization into compartments and domains. The A and B compartments represent the largest organizational units and they functionally correspond to active and inactive chromatin regions, respectively[1,2]. These compartments are further partitioned into topologically associated domains (TADs)[3,4], although due to their heterogeneous nature, new terminology better reflecting the reality is slowly being adopted[5]. Structural segmentation of chromatin in mammals is driven by architectural proteins such as CTCF and the cohesin complex[6–8]. Depletion of either

CTCF[6], cohesin[7] or its loading factor NIPBL[8] leads to global disruption of TAD organization, yet surprisingly with only modest transcriptional changes and unaffected A/B compartments. This, together with the recent findings at base-pair resolution[9], indicates the presence of additional mechanisms of the three-dimensional (3D) chromatin organization. The elimination of cohesin loading factor NIPBL unveiled a finer compartment structure that reflected the underlying epigenetic landscape[8]. This observation is in line with the model in which the primary driver of chromatin organization is the actual transcriptional state[10]. Indeed, RNA polymerase II and transcription itself are tightly

[1]Department of Biology, University of Crete, Heraklion, Crete, Greece. [2]Institute of Molecular Biology and Biotechnology—Foundation for Research and Technology Hellas, Heraklion, Crete, Greece. [3]University Hospital Schleswig Holstein, Kiel, Germany. [4]Laboratory of Bioinformatics and Computational Genomics, Faculty of Mathematics and Information Science, Warsaw University of Technology, Warsaw, Poland. [5]Laboratory of Functional and Structural Genomics, Centre of New Technologies, University of Warsaw, Warsaw, Poland. [6]Present address: Department of Immunology, H. Lee Moffitt Cancer Center and Research Institute, Tampa, FL, USA. [7]Present address: Department of Biochemistry and Molecular Biology, Faculty of Medicine, University of Debrecen, Debrecen HU-4032, Hungary. [8]Present address: Institute for Bioinnovation, Biomedical Sciences Research Centre "Alexander Fleming", 16672 Vari, Greece. ✉e-mail: spiliana@imbb.forth.gr

linked to the formation of finer-scale structures of chromatin organization[11]. Nonetheless, transcriptional inhibition has only a modest effect on promoter-enhancer contacts[11]. Similarly, the inhibition of BET proteins, degradation of BRD4 or dissolution of transcriptional phase condensates all yield in the disrupted transcription, however they also have just a little impact on promoter-enhancer interactions[12]. In contrast, an experimental disruption of TADs[13] or direct manipulation of promoter-enhancer contacts[14] both result in alterations of gene expression. Additionally, establishment of regulatory 3D chromatin arrangements often precedes changes in transcription during development and differentiation[15–18], suggesting that in many scenarios 3D genome organization instructs the transcriptional programs. However, the precise mechanisms on how chromatin organization is linked to gene expression regulation, especially in the context of cell lineage specification, still remain poorly understood. Several transcription factors have been shown to mediate promoter-enhancer interactions and thus also the underlying transcriptional programs in a tissue-specific manner and often even independent of CTCF and cohesin[19–21]. Such an additional regulatory layer of chromatin organization, provided by transcription factors, may represent the missing link between high order chromatin structure and transcriptional regulation.

In this work, we aimed to identify drivers of regulatory chromatin loops in developing murine T cells, as a great model of tissue-specific gene expression regulation. We have identified SATB1 to be enriched at gene promoters and enhancers involved in long-range chromatin interactions. SATB1 has been attributed to many biological roles, mostly related to T cell biology[22], but it also regulates the function of cell types such as the epidermis[23] and neurons[24,25]. Mice with conditionally depleted SATB1 from T cells display impaired T cell development accompanied with an autoimmune-like phenotype[17,26,27]. Moreover, SATB1 is also overexpressed in a wide array of cancers and is positively associated with increased tumor size, metastasis, tumor progression, poor prognosis and reduced overall survival[28]. Originally described as a Special AT-rich Binding protein[29], it is known for its binding to TAATA DNA motif[30–32] and generally in regions with more negative torsional stress[30]. In a proposed model, SATB1 dimers bound to DNA interact with each other to form a tetramer in order to mediate long-range chromatin loops[33,34]. The regulatory function of SATB1 is controlled by post-translational modifications[35] and indirectly also by protein-protein interactions with chromatin modifying complexes[36–40].

To understand the principles of chromatin organization in murine thymocytes and their impact on physiology, we perform Hi-C and HiChIP experiments and compare the roles of tissue-specific SATB1 and ubiquitously expressed CTCF genome organizers. Our findings are complemented by ATAC-seq, RNA-seq and H3K27ac HiChIP experiments in WT and *Satb1* cKO thymocytes to further unravel the functional roles of SATB1. This represents a comprehensive genome-wide study, systematically probing all SATB1-dependent chromatin loops in the T cell nucleus. A number of datasets combined with unbiased analytical approaches indicate the presence of a functional organizational layer built upon a general chromatin scaffold dependent on conventional genome organizers, such as CTCF, specifically regulating expression of master regulator genes and adhesion molecule genes essential for proper T cell development.

## Results

### Detection of regulatory chromatin loops in T cells
In order to unravel the active promoter-enhancer connectome in T cells, we performed H3K27ac HiChIP experiments[41] in C57BL/6J (WT) thymocytes. Loop calling at 5 kbp resolution (FDR ≤ 0.01) yielded 16,458 regulatory loops. To identify the prospective protein factors associated with these regulatory loops, we intersected the anchors of these loops with all the available murine ChIP-seq datasets from blood

cells, using the enrichment analysis of ChIP-Atlas[42]. The most highly enriched protein factors included RAG1, RAG2, BCL11b, SATB1 and TCF1 (Supplementary Data 1). Both RAG1/2 proteins are known to be associated with the H3K27ac histone modification[43,44], however their main known role relies in the recombination of B and T cell receptor loci[45]. BCL11b and TCF1 are well-studied factors specifying T cell lineage commitment, whose roles in forming the chromatin landscape in T cells have been recently addressed[46–49]. We drew our attention to SATB1 which displayed significant enrichment at the H3K27ac loop anchors (Fig. 1a). Moreover, it is a known genome organizer[50,51], which was already found to be associated with enhancers in double positive CD4+CD8+ (DP)[17,27], single positive CD4+ and developing thymic regulatory T cells[17], yet with a limited number of genome-wide studies targeting its role in 3D chromatin organization of T cells.

### Deregulated thymic development in *Satb1* cKO mice
In order to link the molecular mechanisms governing T cell chromatin organization to physiology, we generated a *Satb1*fl/fl*Cd4*-Cre+ (*Satb1* cKO; Supplementary Figs. 1–2) conditional knockout mouse and characterized its phenotype. The knockout animals displayed problems with their skin and fur, inflammation in various tissues and affected lymphoid organs (Supplementary Fig. 3a). The thymi of *Satb1* cKO animals had impaired structural integrity (Fig. 1b) and the intercellular contacts were largely disrupted (Fig. 1c). This was reflected in deregulation of T cell populations (Fig. 1d, Supplementary Fig. 3b, c). In line with previous studies[26,27], mice with SATB1 depleted from T cells had increased percentage of DP cells and decreased percentage of CD4+ and CD8+ single positive (SP) cells in the thymus, indicating a developmental blockade at the DP stage[26,27,52]. Additionally, here we showed that the impaired developmental processes were accompanied by increased apoptosis rate in the thymus (Fig. 1d). This likely contributed to the lower number of thymocytes in the knockout (Fig. 1e) and smaller size of their thymi (Supplementary Fig. 3a), as also previously documented for the SATB1-null mouse[52]. However, as a result of the disrupted thymic structure, we also observed increased exit rate of T cells from the knockout thymus (Fig. 1f). Consequently, these improperly developed T cells altered the composition of T cell populations in the periphery (Fig. 1g, h, Supplementary Fig. 3b, c). In line with the data from the *Satb1*fl/fl*Vav*-Cre+ mouse[26], there was a diminished pool of naïve CD62LhiCD44lo peripheral T cells and an increased fraction of CD44hi T cells displaying an activated (and/or memory) T cell phenotype (Fig. 1g, lower panel) in the *Satb1* cKO animals. The absence of naïve CD4+ T cells together with increased levels of DP T cells in the spleen and lymph nodes (Fig. 1g, h) were suggestive of an autoimmune-like phenotype[53], consistently with other studies[17,26,27]. In line with these, we observed infiltration of T cells in the pancreas of the *Satb1* cKO animals (Fig. 1i, Supplementary Fig. 3c), causing damage to the islets of Langerhans (Supplementary Fig. 3d) and leading to impaired glucose metabolism (Supplementary Fig. 3e). Deregulation of the thymic developmental programs was also supported by the altered cytokine milieu in the blood serum, with prevailing IL-17 cell responses and increased levels of pro-inflammatory cytokines such as IFNγ and TNFα detected in *Satb1* cKO sera (Fig. 1j).

Impaired T cell development in *Satb1* cKO animals is associated with changes to the transcription programs as previously documented[17,27]. Here and in our accompanying publication[54], we have described the vast transcriptional deregulation in the *Satb1*fl/fl*Cd4*-Cre+ animals. Stranded total RNA sequencing revealed that 922 genes were significantly underexpressed and 719 genes were significantly overexpressed in the *Satb1* cKO compared to WT thymocytes (FDR < 0.05; Supplementary Data 2). Additionally, here we identified double negative (DN), DP and SP T cell signature genes (Supplementary Fig. 4a) using data available from sorted T cell subsets[55]. In line with the hypothesis that the DP stage is the most affected in *Satb1* cKO animals,

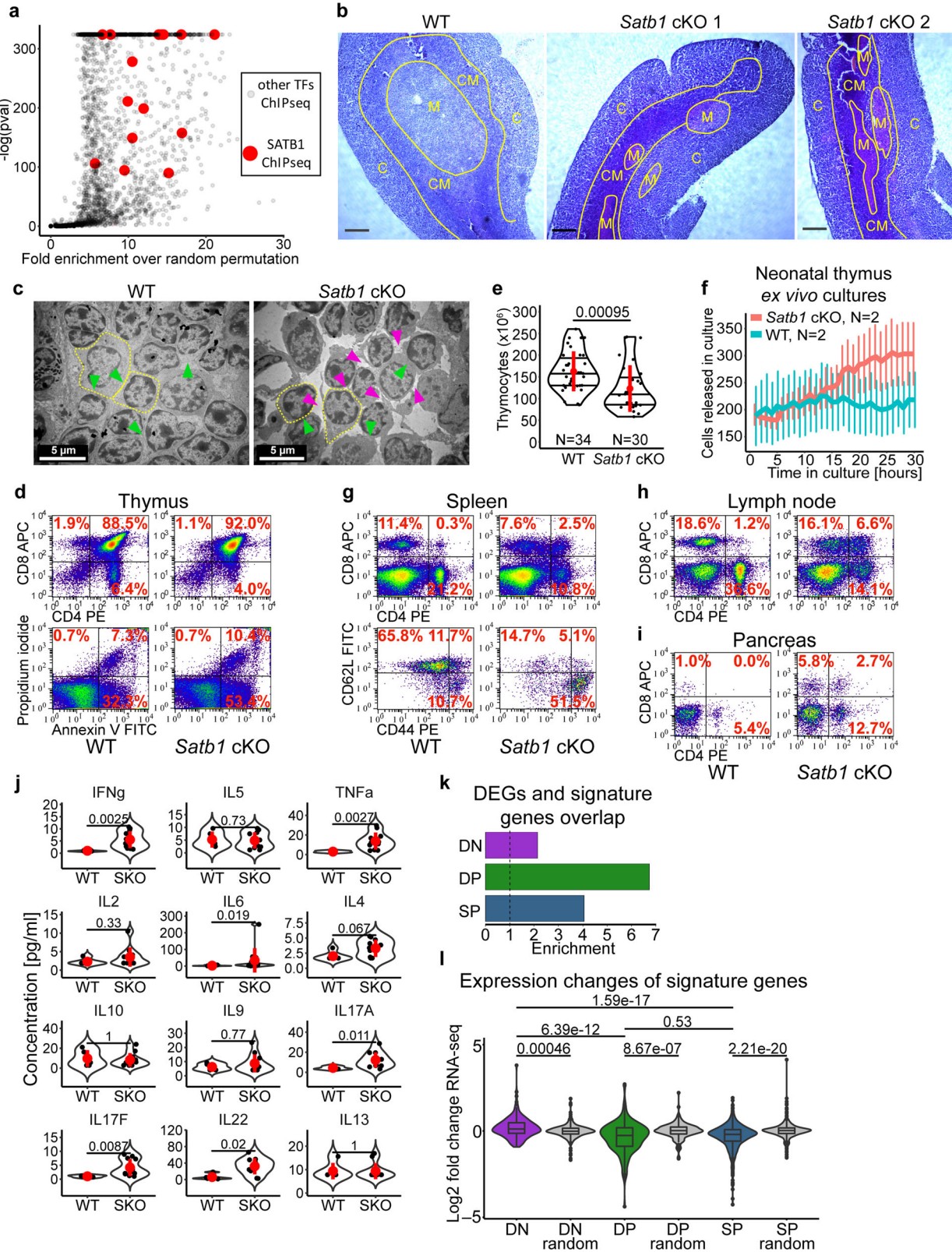

this analysis indicated the highest overlap enrichment between DP signature genes and differentially expressed genes compared to other T cell subset signature genes (Fig. 1k, Supplementary Data 3). Moreover, the DP signature genes displayed the highest drop in RNA levels in *Satb1* cKO animals compared to DN and SP signature genes (Fig. 1l).

Based on these findings, we concluded that SATB1 absence alters transcription programs, leading to impaired thymocyte development

and concomitant deregulation of T cell populations in the secondary lymphoid organs, affecting T cell homeostasis and sustaining an autoimmune-like phenotype.

### Roles of genome organizers in T cell chromatin organization

SATB1 has been previously attributed with genome organizing functions[50,51], therefore we aimed to investigate the potential

**Fig. 1 | Deregulated thymic development in *Satb1* cKO mice. a** Fold-enrichment distribution of peaks from all available blood ChIP-seq datasets based on ChIP-Atlas[42] at WT H3K27ac loop anchors over random permutation and the corresponding p values. Peaks from SATB1 ChIP-seq datasets (highlighted in red) were enriched at the anchors of regulatory chromatin loops. **b** Methylene blue stained thymus sections from WT and *Satb1* cKO mice displaying a disrupted thymic environment in the *Satb1* cKO mouse (C: cortex, M: medulla, CM: corticomedullary region; scale bar 50 μm). The experiment was repeated 4 times in total with similar results. **c** Transmission electron microscopy thymic cryosections (representative of 4 biological replicates) indicating thymocytes with disrupted intercellular contacts in the *Satb1* cKO cells. Cell borders are indicated by yellow dashed lines and green and magenta arrows show examples of intact and disrupted intercellular contacts, respectively. Figures **d, g, h, i**: Flow cytometry analysis to characterize cell populations in the thymus, spleen, lymph node and pancreas of WT and *Satb1* cKO animals, respectively. Gating for CD4/CD8, CD62L/CD44 and Annexin V/PI experiments were G1, G2 and G3, respectively (Supplementary Fig. 3b). Images are representative of the analyzed group and percentages represent the mean. All experiments are summarized in Supplementary Fig. 3c. **e** Number of thymocytes in WT and *Satb1* cKO mice. The horizontal lines inside violins represent the 25th, 50th and 75th percentiles. The red circle represents the mean ± s.d. *P* values by two-sided Wilcoxon rank sum test. **f** Neonatal thymi from WT and *Satb1* cKO mice were cultivated for 30 hours. An image was taken every hour to monitor the rate of T cell exit from the thymus. The final result represents an average from two animals for each genotype. The error bars represent the standard error of the mean. **j** Differences in the cytokine milieu in the blood serum of *n* = 5 WT and n = 11 *Satb1* cKO (SKO) animals measured with bead-based immunoassay. Note the elevated IL17 response and increased inflammatory cytokines. The red circle represents the mean ± s.d. *P* values by two-sided Wilcoxon rank sum test. All data analyzed can be found in the source data file. **k** Overlap enrichment of T cell subset signature genes and differentially expressed genes in *Satb1* cKO. Overlapping genes are depicted in the source data file. **l** Expression changes of DN, DP and SP T cell subset signature genes. DP signature genes displayed the lowest RNA levels in *Satb1* cKO, indicating the specificity of the SATB1 regulatory function at the DP stage. *P* values by two-sided Wilcoxon rank sum test, non-adjusted for multiple comparisons. The box-plots show median with the top and bottom edges of the box representing the 75th and 25th percentiles, respectively. The whiskers represent the most extreme values that are within 1.5 times the interquartile range of the 25th and 75th percentiles. Outliers outside the whiskers are shown as dots. The exact number of features analyzed can be found in the source data file.

deregulation of thymocyte genome organization that is anticipated upon SATB1 depletion and link it to the deregulation of transcription programs and immune physiology that we observed in the *Satb1* cKO mice. To address this, we performed Hi-C experiments[2] in both WT and *Satb1* cKO thymocytes (Supplementary Data 4). Notably, we did not identify any major changes in high-order chromatin organization (Fig. 2a; 500 kbp resolution, balanced normalization). To better understand the role of SATB1 in chromatin organization, we compared the characteristics of WT and *Satb1* cKO Hi-C heatmaps to publicly available Hi-C datasets probing the roles of the conventional genome organizers CTCF (untreated vs CTCF-AID degron system in mESCs[6]) and RAD21 (cohesin subunit; WT vs *Rad21*fl/fl*Cd4*-Cre+ DP thymocytes[56]). As anticipated, the knockout Hi-C datasets for RAD21 and CTCF revealed a perturbation of TADs, which were mostly unaffected in the *Satb1* cKO thymocytes (Fig. 2b, Supplementary Fig. 4b). The modest impact of SATB1 depletion on TADs was also notable from unaltered short- to mid-range contact frequencies (<10 Mbp), both of which were deregulated in CTCF and RAD21 knockouts (Fig. 2c). Similarly, in *Satb1* cKO we did not detect any major changes in chromatin compartmentalization (Fig. 2a, Supplementary Fig. 4c). Saddle plot analysis[57] allows quantification of the degree of homotypic and heterotypic compartment interactions. In line with previous reports[6–8,58–60], differential saddle plot analysis revealed strongly increased homotypic interactions in RAD21-depleted cells (i.e. A/A and B/B; A corresponds to euchromatin and B to heterochromatin regions[2]) indicating reinforced compartmentalization, and slightly deregulated homotypic interactions in the CTCF-depleted cells (Fig. 2d). On the contrary, *Satb1* cKO cells had only a mild deregulation of both hetero- and homo-typic interactions.

Even though broad-scale differences were not observed in the Hi-C maps, one may interrogate more localized conformational changes with HiChIP data, especially given our underlying hypothesis of transcription factor-guided genome organization. Therefore, we next compared the SATB1-dependent and CTCF-dependent chromatin loops by performing HiChIP experiments targeting the respective factors in WT cells. Our HiChIP datasets at 5 kbp resolution (FDR ≤ 0.01) yielded 1,374 and 3,029 loops for SATB1 and CTCF, respectively (Supplementary Data 4–5). It is important to note that in the SATB1 HiChIP experiments we used a custom-made antibody specifically targeting the long SATB1 isoform that we recently characterized[54] (Supplementary Data 6; antibody validated in Supplementary Fig. 1d–g, Supplementary Fig. 2e–f and in manuscript[54]). The validity of this dataset was additionally tested as described in Supplementary Notes and in Supplementary Fig. 4d–e.

We made several comparisons to infer the impact of both SATB1 and CTCF on chromatin organization. First, we performed aggregate peak analysis (APA[1]) to show that SATB1 HiChIP loops had stronger interaction signal in WT Hi-C datasets than in *Satb1* cKO Hi-C. CTCF HiChIP loops retained the same APA score in both Hi-C datasets, indicating that CTCF-based high-order chromatin organization remained unaltered in *Satb1* cKO (Supplementary Fig. 4f). We also performed the aggregate domain analysis (ADA[61]), which similarly indicated diminished interactions within the SATB1-dependent and not within the CTCF-dependent loops in *Satb1* cKO Hi-C datasets (Supplementary Fig. 4g). Furthermore, we compared differentially interacting areas of the HiChIP matrices at 100 kbp and 500 kbp resolution (Fig. 3a). At 100 kbp resolution, 46 interaction pairs were stronger in the SATB1 contact matrix compared to 553 in the CTCF matrix (FDR ≤ 0.05). The analysis at 500 kbp resolution indicated a similar disproportion (7 vs 42), collectively suggesting that CTCF contributes to the high-order chromatin organization in developing T cells to a much higher extent than SATB1. Together with the unaltered RNA (Supplementary Fig. 4h; Supplementary Data 2) and protein (Supplementary Fig. 4i) levels of CTCF in the *Satb1* cKO, all data suggested that the CTCF-cohesin axis was sufficient to maintain the high-order chromatin structure in the *Satb1* cKO cells. An overlap score, calculated by dividing the length of the overlap by the total size of a loop, indicated that most of SATB1-dependent loops were engulfed within CTCF loops (Fig. 3b). CTCF is a well-characterized protein with insulator function[62], therefore, it is reasonable that genes residing within CTCF- and SATB1-dependent loops were transcriptionally insulated from their gene neighbors outside the loops (Supplementary Fig. 4j) and this characteristic was not altered in the *Satb1* cKO. The latter was not surprising since out of 1,374 SATB1-dependent loops (not to be confused with loop anchors), the vast majority (84%) overlapped with at least one CTCF-dependent loop (Fig. 3c). Nevertheless; the binding pattern of these factors was quite different. Similar to previously published results[30], the SATB1 binding sites we have identified, showed a nucleosome preference, unlike CTCF (Fig. 3d). Gene ontology analysis of the genes intersecting with loop anchors uncovered the high propensity of SATB1 to participate in chromatin loops involving immune-related genes, while CTCF-dependent chromatin loops exhibited omnipresent looping patterns resulting in the enrichment of general metabolic and cellular processes (Fig. 3e). Taking these results under consideration we conclude that the high-order chromatin organization of murine thymocytes is primarily maintained via CTCF long-range chromatin interactions with minor input from the SATB1-dependent loops.

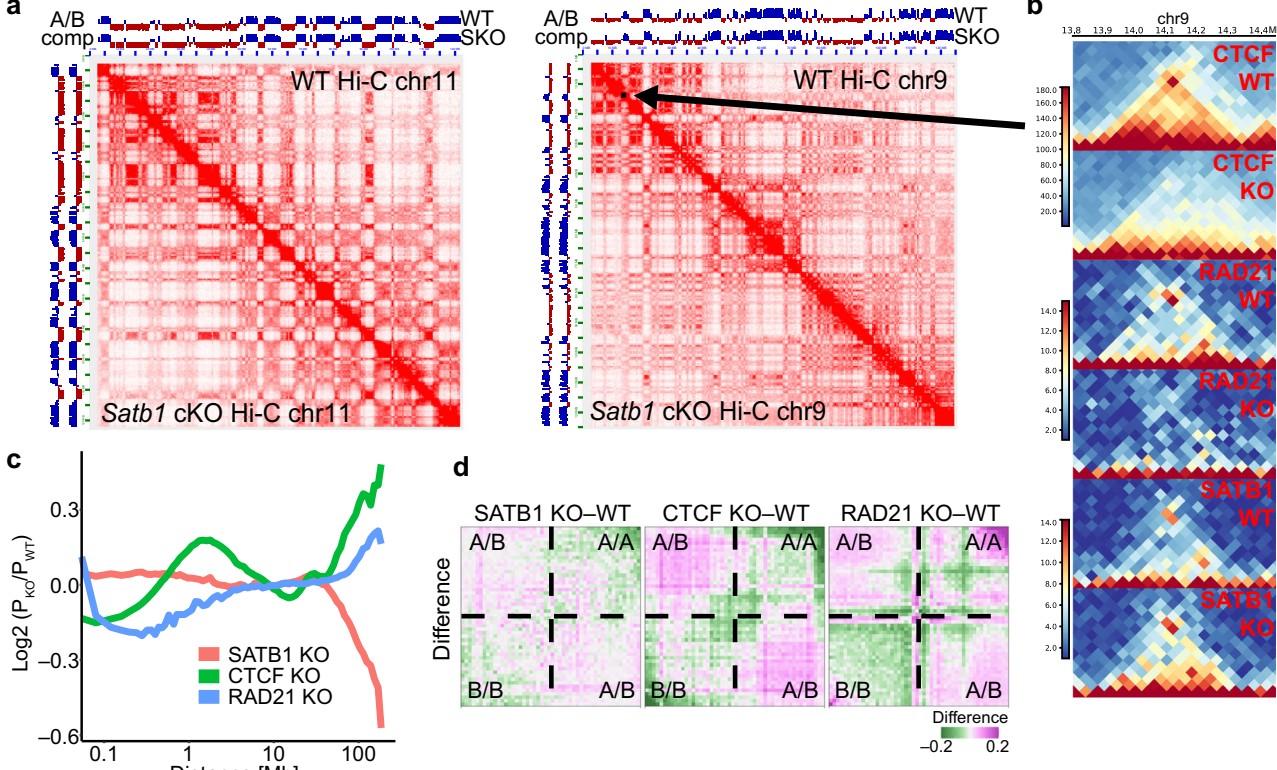

**Fig. 2 | Roles of genome organizers in T cell chromatin organization.**
**a** Comparison of WT and *Satb1* cKO Hi-C heatmaps at 500 kbp resolution (balanced normalization) of chromosomes 9 and 11 indicates no major changes at the high order chromatin level of murine thymocytes. **b** Zoomed-in region of chromosome 9 (indicated by a black arrow in **a**) depicts a topologically associating domain and its disruption in CTCF-depleted mESCs[6] and RAD21-depleted DP T cells[56] but not in SATB1-depleted murine thymocytes (this study). **c** Genome-wide log2 fold change of relative contact probabilities between WT and SATB1-, CTCF- and RAD21-depleted cells, calculated using GENOVA[57]. **d** Saddle plot analysis was computed using GENOVA[57] to illustrate the abundance of inter- vs intra-compartment interactions. The visualization represents the difference between a factor-depleted versus WT observed/expected output of the saddle analysis. In brief, *Rad21* cKO cells displayed the highest compartmentalization, i.e. the strongest gain of homotypic compartment interactions, whereas *Satb1* cKO cells showed only mild compartmental changes.

## Regulatory role of SATB1-dependent chromatin loops in T cells

To unravel the regulatory potential of SATB1-dependent chromatin loops we investigated the impact of SATB1 depletion in *Satb1* cKO thymocytes. These cells displayed deregulated heterochromatin organization (Supplementary Fig. 5a). *Satb1* cKO cells generally appeared to have more compact chromatin compared to wild type counterparts, as demonstrated by the decreased sensitivity of chromatin to DNase I treatment (Supplementary Fig. 5b) as well as by the higher fraction of less accessible regions (6,389 compared to 5,114 more accessible regions; p ≤ 0.01) based on ATAC-seq analysis performed for WT and *Satb1* cKO thymocytes (Supplementary Fig. 5c, Supplementary Data 7). One of the differentially accessible regions based on ATAC-seq was also validated by DNase I treatment followed by qPCR (Supplementary Fig. 5d). Decreased chromatin accessibility in *Satb1* cKO was also apparent from the increased volume of intensively DAPI stained areas of individual nuclei in *Satb1* cKO confocal microscopy experiments, accompanied by the decreased volume of the phosphorylated form (phosphorylated Ser 5 of CTD) of RNA polymerase II regions per cell (Supplementary Fig. 5e). Both this and the chromatin accessibility changes were reflected at the transcriptional level, as demonstrated by the vast transcriptional changes in the *Satb1* cKO compared to WT thymocytes[54] (Supplementary Data 2). Such a strong deregulation of the transcriptional landscape in *Satb1* cKO cells in contrast to the modest transcriptional changes observed upon depletion of conventional genome organizers[6–8] emphasizes the regulatory importance of SATB1-dependent chromatin organization.

SATB1 is known for its functional ambiguity of acting either as a transcriptional activator or a repressor[35,63]. Although the original studies were mainly focused on its repressive roles[64–66], our aforementioned observations supported the increased chromatin compactness and subsequent repressed environment of the SATB1-depleted cells. This rather indicated its positive impact on transcriptional gene regulation. The majority of SATB1 binding sites in WT thymocytes had higher chromatin accessibility than randomly shuffled binding sites (Fig. 4a), with a visible drop in chromatin accessibility in the *Satb1* cKO (Fig. 4b). This drop in chromatin accessibility in *Satb1* cKO cells was especially evident at the transcription start site of genes (TSS; all genes were used in this analysis), suggesting a direct role of SATB1 in gene transcription regulation (Fig. 4c, Supplementary Fig. 5f). Moreover, the expression changes of significantly deregulated genes were positively correlated with the changes of chromatin accessibility at promoters determined by ATAC-seq (Supplementary Fig. 5g). Only about 5% of SATB1 binding sites had low chromatin accessibility in WT (lower than the average accessibility score of ten randomizations depicted in Fig. 4a). These regions had increased chromatin accessibility in the *Satb1* cKO (Supplementary Fig. 5h), which would suggest a repressive function of some SATB1 molecules. However, these regions were not enriched for immune-related genes (not shown) and thus probably not contributing to the observed phenotype.

We have next created a linear regression model, as an unbiased way to identify how gene expression was affected in murine thymocytes upon SATB1 depletion. We utilized SATB1 binding, SATB1- and

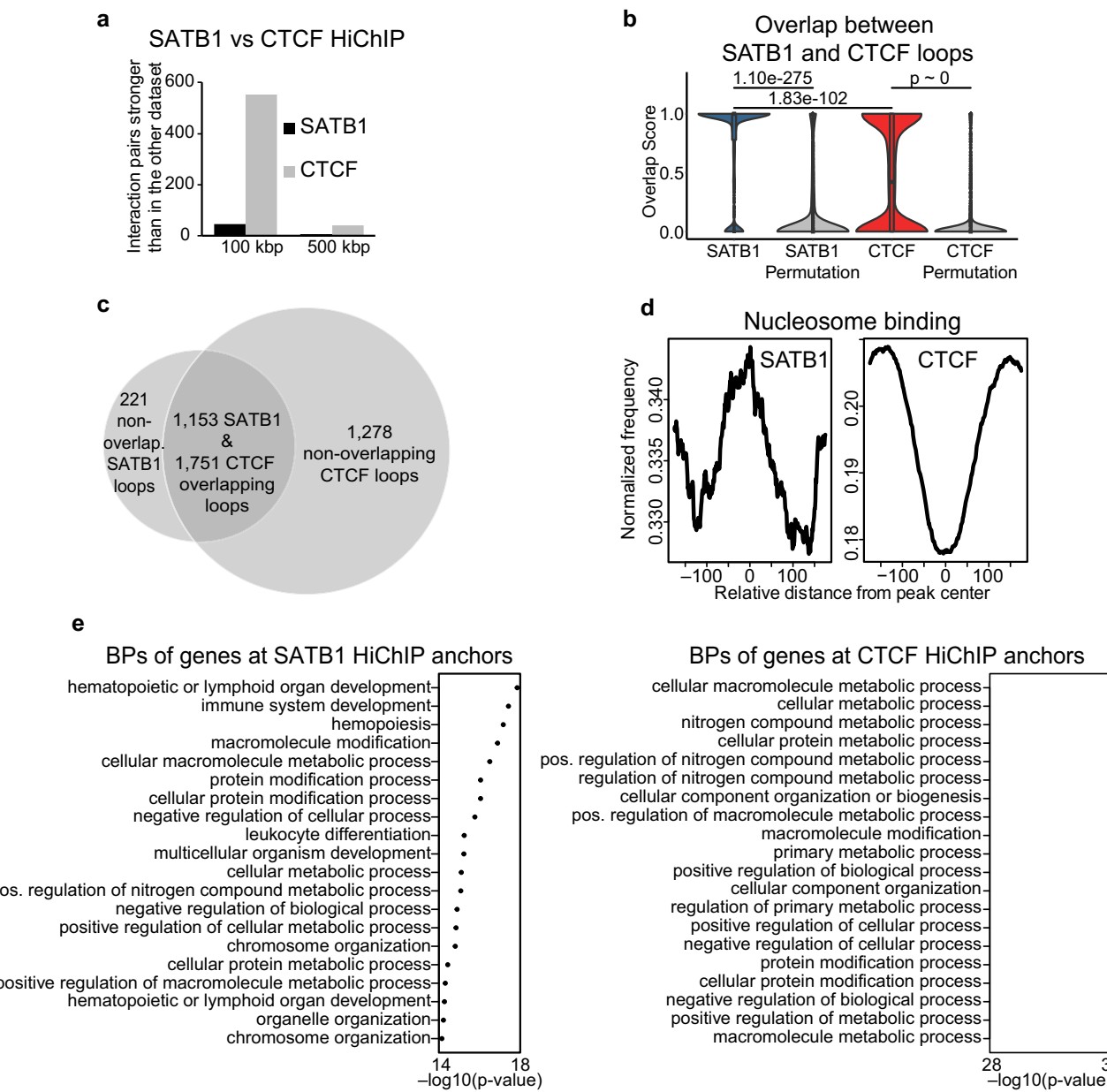

**Fig. 3 | Fine-scale T cell chromatin organization by SATB1. a** diffHic analysis[107] of differentially interacting chromatin areas indicates that CTCF contributes more strongly to the higher order chromatin organization of the murine T cell genome than SATB1. **b** Overlap score between SATB1 and CTCF loops calculated as (number of overlapping bp) / (bp size of a loop). For example, for the SATB1-labeled violin, a score of 1.0 indicates either 100% overlap or engulfment of a SATB1-dependent loop in a loop dependent on CTCF. A score of 0.0 indicates no overlap. The plot indicates that the majority of the SATB1 loops were engulfed in CTCF loops. The same approach was repeated for randomly shuffled loops. *P* values by two-sided Wilcoxon rank sum test, non-adjusted for multiple comparisons. The exact number of features analyzed can be found in the source data file. **c** SATB1-dependent loops highly overlap with CTCF-dependent loops detected by HiChIP. For the overlap, the outer coordinates of left and right loop anchors were used. **d** SATB1 preferentially binds nucleosomes unlike CTCF, as determined from analysis of the ATAC-seq data using NucleoATAC[132]. **e** SATB1 loop anchors overlap with genes enriched for immune system-related biological processes (BPs). CTCF-dependent loops display more widespread coverage of intersecting genes thus the most enriched gene ontology pathways belong mostly to general cellular processes. Cumulative hypergeometric P values calculated by g:Profiler[121] are displayed.

CTCF-dependent chromatin loops, changes in H3K27ac-dependent chromatin loops and changes in chromatin accessibility at different positions of a gene, as predictors of RNA level changes between WT and *Satb1* cKO cells. We found that the majority of predictors exhibited an expected behavior, such as increased chromatin accessibility at gene promoters was associated with increased gene expression (Supplementary Notes, Supplementary Fig. 6). The regression model highlighted SATB1 binding and SATB1-dependent chromatin loops as good predictors associated with decreased RNA levels of influenced genes in the *Satb1* cKO. In this analysis, we applied the model for all

known genes, which resulted in a quite low R-squared value (0.113). For this reason, we verified the activatory role of SATB1 loops with an additional approach. We constructed a conditional inference tree, where we systematically probed the distribution of gene expression for the genes that were or were not found in SATB1-dependent loops (Supplementary Fig. 7a). Genes located in SATB1-dependent loops displayed lower RNA levels in the *Satb1* cKO, an effect that was further intensified when the gene was connected to a thymus-specific enhancer, suggesting a positive role for SATB1 in gene transcription via promoter-enhancer mediated chromatin loops.

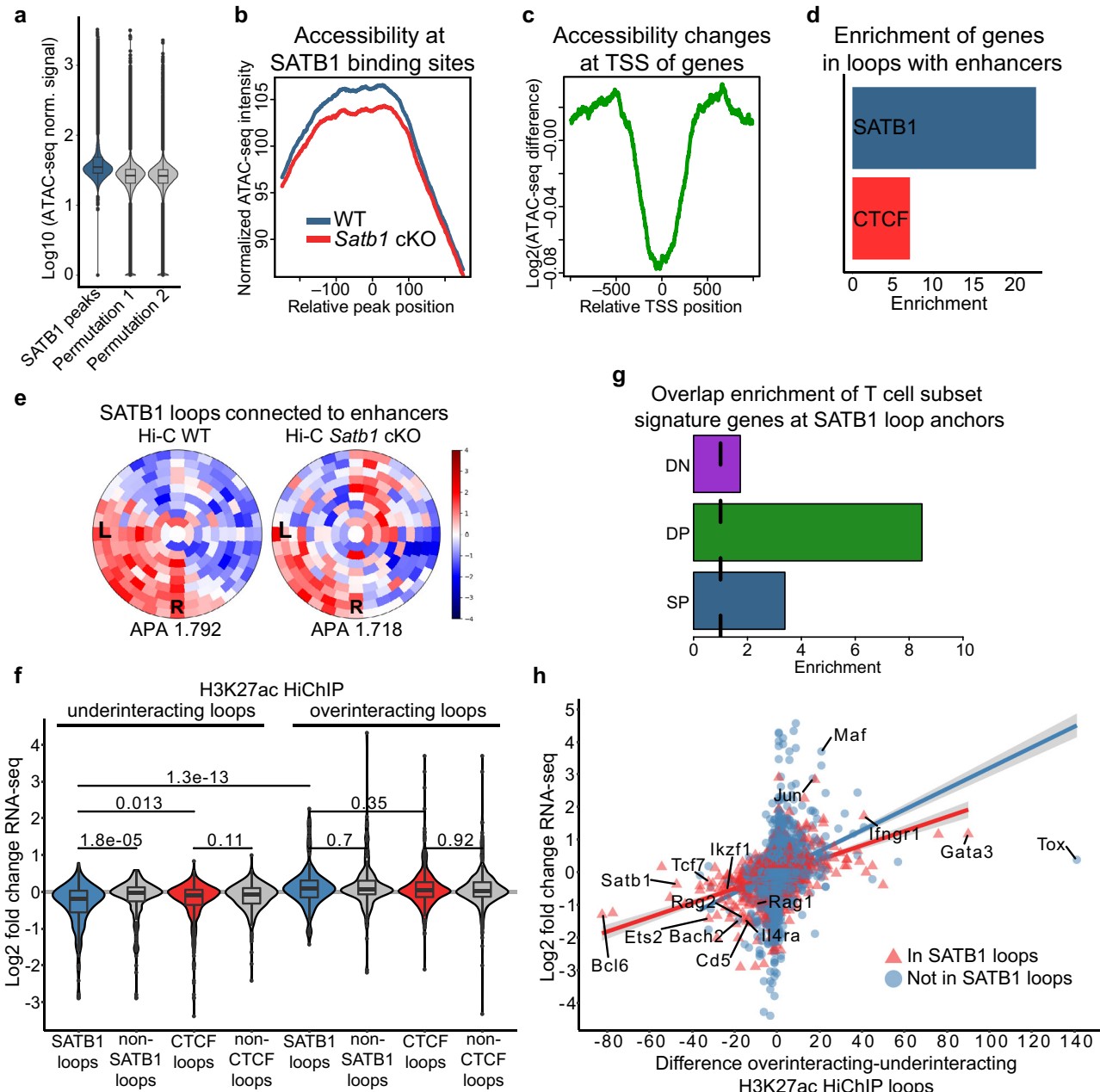

**Fig. 4 | SATB1-dependent promoter-enhancer communication of critical immune-related genes. a** ATAC-seq signal indicates higher chromatin accessibility at WT SATB1 binding sites than expected by chance (i.e, randomly shuffled SATB1 binding sites). 100 randomizations were used for statistical evaluation (p-value - 0). Two representative random distributions are depicted in the figure. **b** Chromatin accessibility at SATB1 binding sites is decreased in *Satb1* cKO. **c** Log2 fold change of chromatin accessibility indicates the highest accessibility drop in *Satb1* cKO being at the TSS of genes. **d** SATB1-dependent loops connecting genes to enhancers are about three-fold enriched compared to CTCF loops. **e** SATB1-dependent loops connected to enhancers display enriched interaction signal between left (L) and right (R) anchor in WT Hi-C data (z-score normalized), which is deregulated in the *Satb1* cKO. Aggregate peak analysis (APA)[1] was calculated and visualized by SIPMeta[61]. See also Supplementary Fig. 4g for more details. **f** Log2 fold change expression values of genes that were present in anchors of H3K27ac HiChIP loops under- and over-interacting in *Satb1* cKO and which also did or did not intersect with SATB1/CTCF-dependent loops. An equal number of underinteracting loops

that did not overlap with SATB1/CTCF-dependent loops were randomly generated. *P* values by two-sided Wilcoxon rank sum test, non-adjusted for multiple comparisons. **g** Overlap enrichment between DN, DP and SP T cell subset signature genes and anchors of SATB1-dependent loops. Overlapping genes are depicted in the source data file. **h** Scatter plot indicating positive correlation between gene expression changes and the difference between H3K27ac overinteracting and underinteracting loops in WT thymocytes compared to *Satb1* cKO. See Supplementary Fig. 7d for complete information. The grey zones indicate 95% confidence level interval for predictions from a linear model (red and blue lines for genes present and absent in SATB1-dependent loops, respectively). In **a** and **f**, the boxplots show median with the top and bottom edges of the box representing the 75th and 25th percentiles, respectively. The whiskers represent the most extreme values that are within 1.5 times the interquartile range of the 25th and 75th percentiles. Outliers outside the whiskers are shown as dots. The exact number of features analyzed can be found in the source data.

Thymic enhancers were previously shown to be occupied by conventional genome organizers such as CTCF and cohesin, suggesting their involvement in gene regulatory loops of thymocytes[56,67]. We utilized a list of predicted thymus-specific enhancers[68] and we found more than 2-fold enrichment of SATB1- over CTCF-dependent loops overlapping with such enhancers (Supplementary Fig. 7b) and more than 3-fold enrichment of genes connected to enhancers by SATB1-dependent chromatin loops over CTCF-dependent loops (Fig. 4d). SATB1-dependent loops connected to thymus-specific enhancers, also displayed a disturbed chromatin interaction pattern in *Satb1* cKO Hi-C data compared to WT (Fig. 4e). Collectively, these findings suggested that CTCF participates in mechanisms responsible for supporting a basal high-order T cell chromatin structure, whereupon SATB1 likely exerts its action in a more refined organization layer consisting of promoter-enhancer chromatin loops.

## Deregulated promoter-enhancer loops in *Satb1* cKO T cells

To further investigate the latter hypothesis, we compared the promoter-enhancer chromatin loops present in WT and *Satb1* cKO thymocytes, utilizing the H3K27ac HiChIP loops. H3K27ac HiChIP in *Satb1* cKO thymocytes yielded 19,498 loops (compared to 16,458 loops detected in WT; Supplementary Data 4–5). Differential analysis of the 3D interactions (independent on the 1D H3K27ac ChIP-seq signal) identified 11,540 and 12,111 H3K27ac loops displaying decreased or increased contact enrichment in the *Satb1* cKO compared to WT cells, respectively (further referred as "underinteracting" and "over-interacting" H3K27ac loops, respectively; Supplementary Data 8). The SATB1-dependent underinteracting H3K27ac loops displayed the highest drop in the RNA levels of the overlapping genes compared to those in non-SATB1 underinteracting H3K27ac loops (Fig. 4f). In contrast, the genes localized in anchors of overinteracting H3K27ac loops did not show any major changes in expression (Fig. 4f). Moreover, the expression of genes located at anchors of SATB1-dependent loops was dramatically decreased supporting the direct involvement of SATB1 in the regulatory chromatin loops (Fig. 4f). Next, we sought to investigate the genes intersecting with anchors of the differential H3K27ac loops. Overlap enrichment for anchors of SATB1-dependent as well as differential H3K27ac loops was the highest for DP T cell signature genes (Fig. 4g, Supplementary Fig. 7c). Expression of genes associated with differential H3K27ac HiChIP loop anchors displayed a positive correlation with H3K27ac differential looping (Fig. 4h, Supplementary Data 8). The expression of DP T cell signature genes was the most positively correlated with H3K27ac differential looping compared to DN and SP T cell signature genes (Supplementary Fig. 7d). Moreover, the functional link between SATB1-dependent chromatin organization and transcription regulation was demonstrated by strong positive correlation between the RNA levels of differentially expressed genes and differential H3K27ac looping (Spearman's $\rho = 0.62$; Supplementary Fig. 7d).

Overall, our data indicate that SATB1 promotes activatory promoter-enhancer chromatin loops in developing thymocytes. However, it should be noted that we also identified a number of *Satb1* cKO overinteracting H3K27ac loops. The overinteracting H3K27ac with the highest score even contained some factors important for proper T cell development and differentiation such as *Tox*, *Gata3*, *Ifngr1*, *Maf* and/or *Jun* (Fig. 4h, Supplementary Data 8), which all correspondingly displayed increased gene expression in the *Satb1* cKO thymocytes. Certain genes present in the overinteracting H3K27ac loops were bound by SATB1 and a fraction of them was also found in SATB1-dependent loops. Thus, we cannot exclude the possibility that some SATB1 molecules mediate a repressive role for these targets. Nevertheless, our unbiased approaches have primarily suggested an activatory role for the SATB1-dependent loops, hence in this work we focused on this. Moreover, the underinteracting H3K27ac loops with the highest score included more genes encoding for master regulators

and T cell signature genes such as *Bcl6*, *Ets2*, *Tcf7*, *Cd8b1*, *Ikzf1*, *Bach2*, *Cd6*, *Rag2*, *Il4ra*, *Rag1*, *Lef1* and others (in descending order; Fig. 4h, Supplementary Data 8), which all displayed decreased RNA levels in the *Satb1* cKO. Therefore, these observations are all in line with the existence of functionally important, SATB1-dependent promoter-enhancer chromatin loops.

## SATB1 positively regulates *Bcl6* and other master regulator genes

In line with previously published results[27], our data showed the SATB1-dependent transcriptional regulation of several genes with important immune regulatory function (Fig. 4h, Supplementary Data 8). Genomic tracks and SATB1-dependent HiChIP loops for selected genes (*Tcf7*, *Lef1*, *Cd8*, *Satb1*, *Ikzf1*, *Socs1*) are presented in Supplementary Fig. 8.

The most highly affected candidate gene, in terms of H3K27ac underinteracting loops in the *Satb1* cKO compared to WT, was *Bcl6*. The expression of *Bcl6* gene in B cells is regulated by a set of super-enhancers; one spanning the promoter and 5' UTR region, and additional three distal upstream enhancer stretches at 150–250 kbp, -350 kbp and at -500 kbp[69–72]. Apart from H3K27ac differential loops, we identified increased SATB1-dependent interactions in the locus with two significant SATB1-dependent loops connecting *Bcl6* and the super-enhancer regions at -250 kbp and -500 kbp upstream of the gene (Fig. 5a; here referred to as SE1 and SE2, respectively). These enriched chromatin interactions observed in WT were absent in the *Satb1* cKO thymocytes as deduced by Hi-C experiments (red arrows; validated by 3 C in Supplementary Figs. 9 & 10a, b). Next, we utilized this gene locus as an example for 3D modeling experiments. 3D modeling based on Hi-C data allowed us to better visualize the differences in the proximity between *Bcl6* and its super-enhancers in WT and *Satb1* cKO cells (Fig. 5b). As a result of this deregulation, the *Bcl6* gene displayed significantly lower RNA levels in the *Satb1* cKO thymocytes (log2FC = −1.290, FDR = 5.6E−10; Supplementary Data 2; validated in Supplementary Fig. 10c), which was also reflected at the protein level (Fig. 5c and Supplementary Fig. 10d). Next, we utilized our CTCF and SATB1 HiChIP data and performed computational modeling which supported the idea that SATB1-dependent chromatin landscape represented a regulatory layer, built on a generic scaffold dependent on other factors, at least partly by CTCF (Supplementary Fig. 11a). To further support the functional significance of interactions between *Bcl6* and its super-enhancers, we overlaid the 3D models with ChIP-seq data for the histone modifications H3K27ac, H3K4me1 and H3K4me3 from WT cells (Supplementary Fig. 11b). These models showed that active enhancers decorated by H3K27ac and H3K4me1 were located in spatial proximity to *Bcl6* gene in WT and not in *Satb1* cKO cells. It is worth noting that in line with a previous study[17], 1D H3K27ac ChIP-seq peaks derived from HiChIP experiments available for WT and *Satb1* cKO did not reveal any major differences between the genotypes, which further reinforces the importance of SATB1-dependent 3D chromatin organization regulating *Bcl6* expression. 3D modeling, based on thymocyte Hi-C datasets, allowed us to untangle the potential presence of discrete cell subpopulations. In WT animals, we identified two subpopulations of cells differing in the proximity between *Bcl6* and its super-enhancers (Supplementary Fig. 11c); yet no significant subpopulation formation was detectable in the *Satb1* cKO. BCL6 is the master regulator of Tfh cell lineage specification during the differentiation of naive CD4⁺ cells into Tfh cells[73–75]. However, it is also expressed in developing thymocytes[76,77] where it was shown to form a complex with E3 ubiquitin ligase CUL3 and exert a negative feedback loop on the Tfh program by repressing *Batf* and *Bcl6*[78]. Thus, we speculate that the different subpopulations identified in our 3D modeling may be linked to the different developmental T cell fates, where Tfh precursor cells would differ from other cell type precursors by the distance between Bcl6 and its super-enhancers. In support to

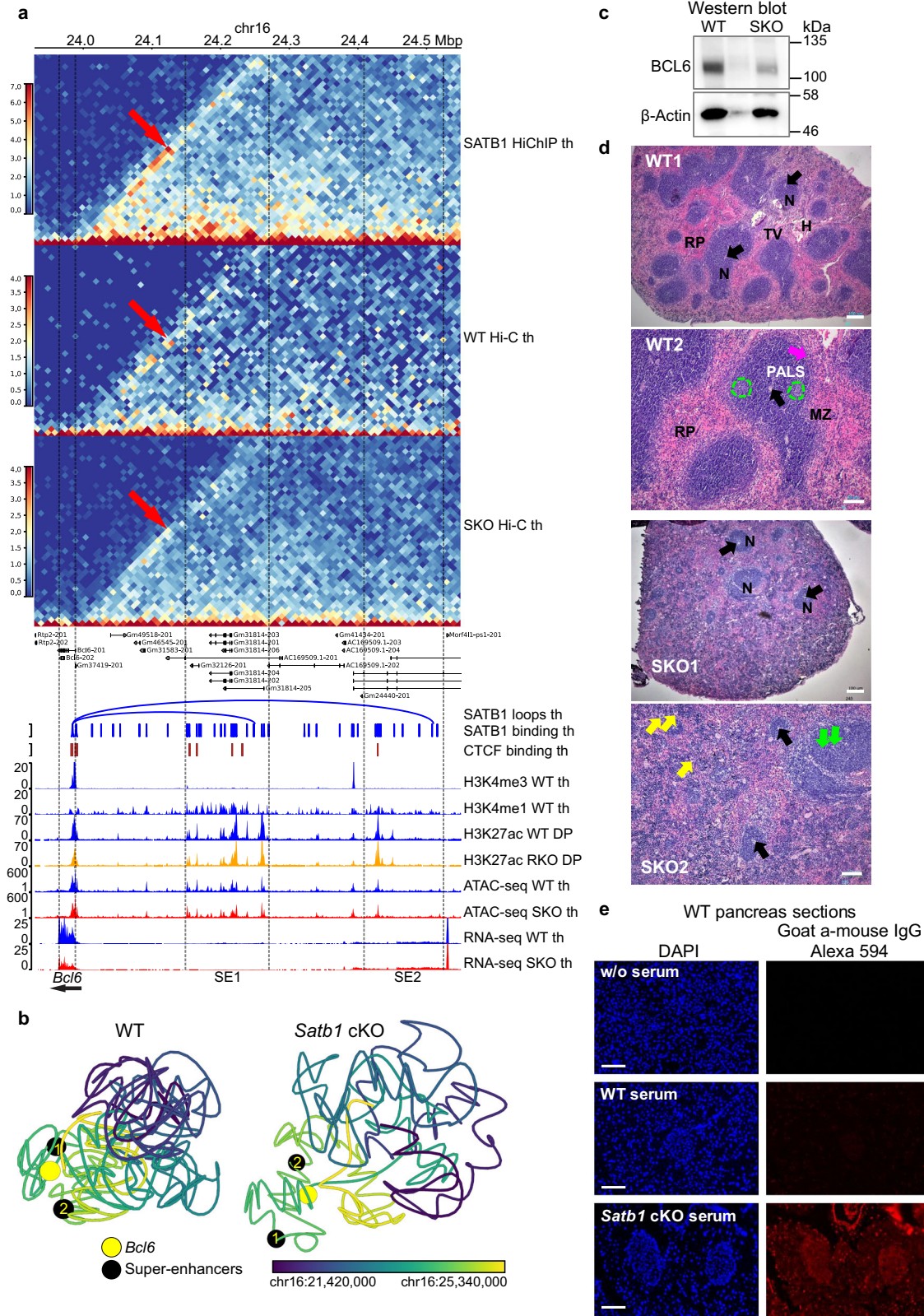

the connection between SATB1-dependent regulation and Tfh lineage specification, we demonstrated that Satb1 cKO animals had disturbed spleen structure, containing fewer lymphoid nodules harboring impaired B cell follicular regions with accumulated phagocytic cells and increased apoptosis (Fig. 5d). This translated into the production of autoantibodies (Fig. 5e, Supplementary Fig. 11d) as also previously demonstrated[26].

## Cell adhesion in *Satb1* cKO and deregulation of T cell receptor locus

Gene ontology analysis for both underexpressed genes by RNA-seq and genes associated with less accessible regions as deduced by ATAC-seq (together with GO pathways for other genomic datasets presented in Supplementary Fig. 12) showed enrichment for the "cell adhesion pathway". Indeed, genes encoding key molecules for intrathymic

**Fig. 5 | SATB1-dependent transcriptional regulation of *Bcl6* impacts Tfh development. a** Two significant SATB1-dependent loops connect *Bcl6* (same loop anchor for both loops; chr16:23985000–23990000; mm10) with its super-enhancer regions (SE1 – chr16:24245000–24250000, SE2 – chr16:24505000–24510000). In *Satb1* cKO, these interactions were diminished as seen in the Hi-C heatmap. Legend: th – thymocytes, DP – CD4⁺CD8⁺ T cells, RKO – *Rad21*^fl/fl^*Cd4*-Cre⁺ and SKO – *Satb1*^fl/fl^*Cd4*-Cre⁺. **b** Computational 3D modeling utilizing the WT and *Satb1* cKO Hi-C data, to visualize the proximity between the *Bcl6* gene body and its two super-enhancer regions. The black beads represent the edge of super-enhancer regions demarcated by the SATB1 loop anchors (SE1 and SE2 as in **a**). Color gradient represents linear genomic position along the locus. **c** Western blot analysis for BCL6 expression of whole cell protein extracts prepared from WT and *Satb1* cKO thymocytes. Actin serves as a loading control (upon pixel analysis of the images a 45% reduction of BCL6 expression in *Satb1* cKO thymocytes was calculated). Two biological replicates were performed with similar results. The Western blot analysis was performed three times with similar results. **d** H&E staining of mouse spleen sections. In WT sections, Red Pulp (RP), lymphoid Nodules

of the white pulp (N), the Hilus (H) and Trabecular Vein (TV) are labeled. Arteries (black arrows) are present in each nodule. The nodule structure is clear with extensive periarteriolar lymphocyte sheath (PALS), rich in round, dark stained T lymphocytes surrounding the arterioles (WT2). The distinct marginal zone surrounding the marginal sinus (MZ and magenta arrow, respectively) is shown. Dotted lines mark lighter stained B-lymphocyte rich follicle regions within the nodule. In *Satb1* cKO (SKO), there is disturbed spleen structure with few apparent small sized lymphoid nodules with the periarteriolar region depleted of T lymphocytes. The follicular region has accumulated large phagocytic cells and displays many foci of phagocytosis of apoptotic cells (green arrows). The red pulp contained higher numbers of haemopoietic cell clusters and megakaryocytes (yellow arrows) suggesting increased haemopoietic activity. Scale bar WT1 & SKO1 100 μm and WT2 & SKO2 50 μm. The experiment was repeated four times with similar results. **e** WT pancreas sections were incubated with serum from either WT or *Satb1* cKO animals to detect the presence of autoantibodies in the *Satb1* cKO serum. Scale bar 100 μm. The experiment was repeated 2 times with similar results.

crosstalk[79] were all underexpressed in the *Satb1* cKO (Fig. 6a). Expression of these adhesion molecule gene loci such as *Cd28*, *Lta*, *Ltb* and *Ccr7* was affected by SATB1-dependent regulatory looping (Supplementary Fig. 13).

We should note that in the differential analysis of H3K27ac loops, short genes could be underestimated, hence we further considered gene length in our analysis. Indeed, upon taking this into account, the most affected genes in both categories of overinteracting and underinteracting loops were enriched for gene segments of the T cell receptor (TCR) locus (Fig. 6b; Supplementary Data 8). TCR is the most important cell surface receptor expressed in thymocytes which defines multiple developmental decisions. The generation of a functional TCR involves recombination of the variable (V), diversity (D) and joining (J) gene segments via a process called V(D)J recombination. This process is based upon the action of protein complexes that include RAG1 and RAG2 recombinases[45]. Recruitment of RAG proteins is highly correlated with active promoters labeled with H3K4me3[44,80] and also with enhancers and regions decorated by the H3K27ac mark[43,44]. Moreover, recombination is regulated by the 3D organization of the locus driven by the architectural proteins CTCF and cohesin, mainly via the arrangement of specific TCR enhancers[81,83].

Here, we revealed a number of SATB1-dependent loops connecting the TCRα enhancer with inner regions of the locus (Fig. 6c). Moreover, a number of highly significant SATB1 loops split the region of joining gene segments into two parts – one part containing gene segments that were overexpressed and the other half containing gene segments that were underexpressed in the *Satb1* cKO. This resulted in the defective usage of the TCRα joining segments (Fig. 6c), coupled with the overall *Tcra* rearrangement in *Satb1* cKO animals as previously reported[84,85]. A previous study ascribed this deregulation to the lost SATB1-dependent regulatory loops, positively controlling the expression of both *Rag1*/*Rag2* genes resulting in lower levels of RAG proteins[85]. We validated the presence of these regulatory loops (Fig. 6d) as well as the resulting 2-fold and 2.8-fold decrease in thymic RNA levels of the *Rag1* and *Rag2* genes (Fig. 6e; Supplementary Data 2; validated in Supplementary Fig. 10c), respectively (which was less profound in sorted DP cells: 1.34-fold and 1.61-fold decrease for *Rag1* and *Rag2* genes, respectively; data not shown). However, given the long turnover of RAG proteins, their thymic protein levels were not significantly affected (Fig. 6f). Moreover, the representation of *Traj* fragments was correlated with the presence of overinteracting and underinteracting H3K27ac loops (Fig. 6g), indicating the importance of TCR locus 3D chromatin organization in its rearrangements.

The decline in thymic adhesion molecules was accompanied by decreased RNA levels of receptors specific for the medullary thymic epithelial cells (Fig. 6a), as detected by total thymocyte RNA

sequencing. Such changes were anticipated, given the overall disruption of the thymic structure in *Satb1* cKO animals (Fig. 1b). The deregulation of T cell signaling was also in line with the reduction of intrathymic cell-to-cell contacts (Fig. 1c), ultimately representing a link between the 3D chromatin organization and impairment of the T cell development in *Satb1* cKO animals.

## Discussion

The adaptive immune response relies on the accurate developmental coordination of several alternative cell lineage fates. 3D genome organization in T cells represents a crucial denominator for this coordination[86,87]. In this work, we described the regulatory chromatin network of developing T cells and identified SATB1 protein being enriched at the anchors of regulatory chromatin loops. We performed a systematic genome-wide analysis of SATB1 roles in T cells. First, we compared the chromatin organization role of SATB1, to that of the conventional genome organizer CTCF. Utilizing a plethora of research approaches, we demonstrated that SATB1 establishes a finer-scale organizational layer, built upon the pre-existing scaffold dependent on other architectural proteins. Depletion of conventional genome organizers such as CTCF[6] or cohesin[7,8] resulted in vast deregulation of TADs, however it did not show dramatic changes in gene expression as one would expect. On the contrary, SATB1 depletion did not result in any major changes in high order chromatin organization. However, the long-range promoter-enhancer interactions, within the unaltered TADs, were highly deregulated as well as the underlying transcriptional programs.

It was not clear so far whether SATB1 should be primarily assigned a role as an activator or a repressor. This characteristic makes SATB1 markedly similar to the ubiquitously expressed factor YY1. YY1 was also found enriched at promoters and enhancers, mediating their spatial contacts[88]; however, depending on the cellular context it has also been found in association with Polycomb repressive complexes[89]. It would be interesting to further investigate the determinants of such functional ambiguity for such factors. In our experimental setup, we have demonstrated the activatory function of SATB1, in specifically mediating promoter-enhancer long-range chromatin interactions. Although the repressed nuclear environment of *Satb1* cKO cells was in agreement with those findings, we should note that in our study we were mostly focused on functions of the long SATB1 protein isoform. The presence of two SATB1 protein isoforms was recently described by our group[54] and it could be another reason, aside from various post-translational SATB1 variants[22,35], supporting its functional ambiguity. We showed that the long SATB1 protein isoform had a higher propensity to undergo phase transitions compared to the short isoform[54]. Considering the proposed model of transcriptional regulation via liquid-liquid phase separated transcriptional condensates[90,91], we

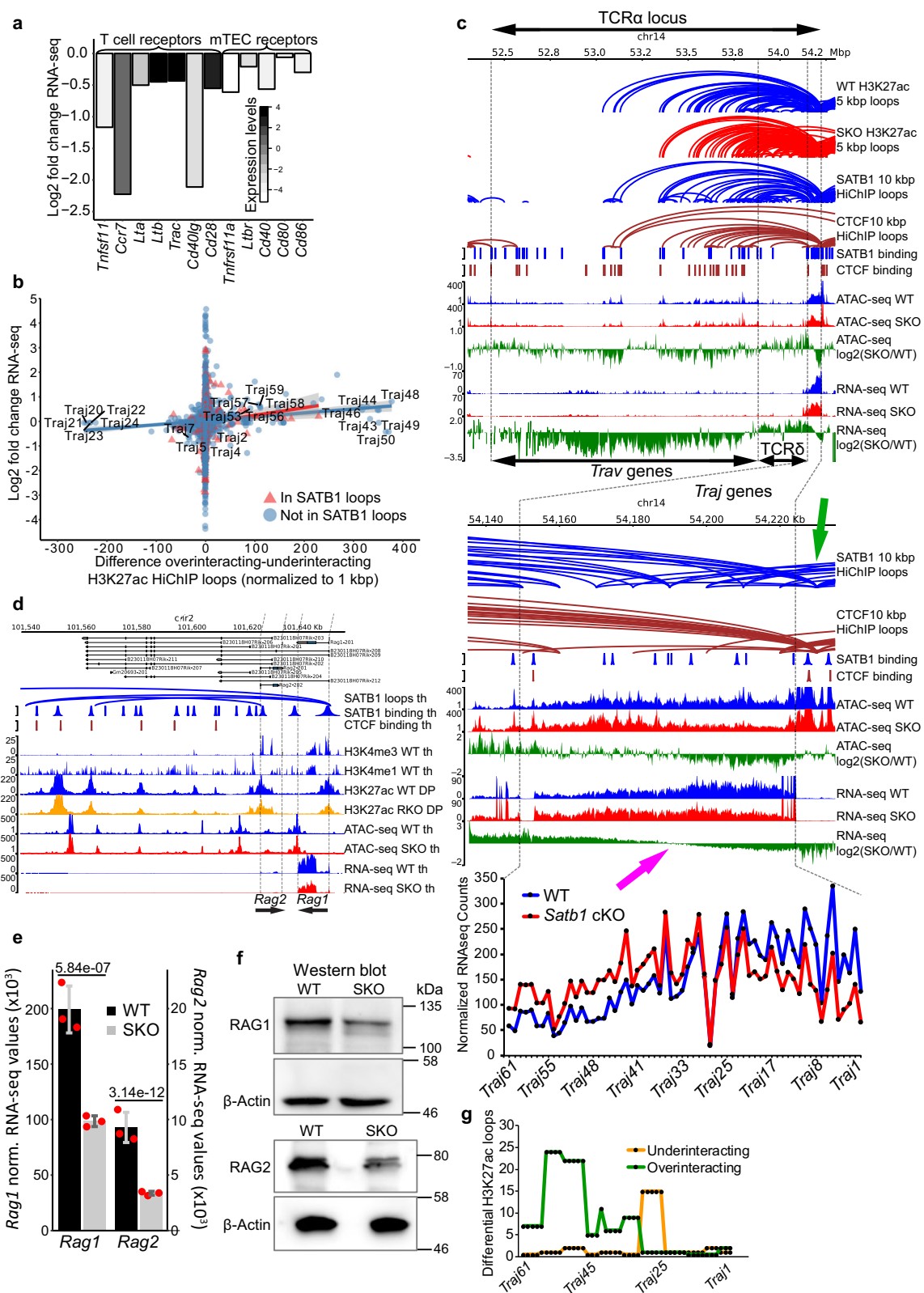

reasoned that even the subtle differences in biophysical properties between the two SATB1 isoforms may play an important regulatory role. Following these observations, we hypothesize a model in which SATB1-dependent contacts between promoters and enhancers are attracted to the transcriptional condensates depending on the type and concentration of the SATB1 variants present. Post-translational modifications and/or different SATB1 isoforms would therefore

regulate the whole process and under certain conditions, SATB1 could even function as a repressor. Thus, in comparison to other transcription factors capable of mediating long-range chromatin interactions, the SATB1's mode of action may include one or more mechanisms that were proposed[19–21]; i.e. via direct[33,34] or indirect[36–40] oligomerization or via phase separation[54]. Overall, the presence of proteins like SATB1 with tissue-restricted expression profile, may represent the missing

**Fig. 6 | SATB1 controls expression of TCR and other adhesion molecule genes.**
**a** Important thymic receptors were underexpressed in *Satb1* cKO based on RNA-seq
data. **b** Scatter plot indicating positive correlation between gene expression
changes and the difference between H3K27ac overinteracting and underinteracting
loops in WT compared to *Satb1* cKO thymocytes that was normalized to the 1 kbp
gene length. The grey zones indicate 95% confidence level interval for predictions
from a linear model (red and blue lines for genes present and absent in SATB1-
dependent loops, respectively). **c** Genomic tracks as well as SATB1 and CTCF HiChIP
loops at the T cell receptor alpha locus (TCRα). The bottom green genomic tracks
of log2 fold change RNA-seq values summarize the overall deregulation of the
TCRα locus, with variable regions (*Trav* genes) being mostly underexpressed, TCRδ
locus overexpressed and TCRα joining regions (*Traj* genes) displaying geome-
trically symmetric deregulation splitting the region into over- and under-expressed
in *Satb1* cKO cells (magenta arrow). This deregulation was markedly correlated with
the presence of SATB1-dependent loops. Note especially the region of joining genes

which manifests a deregulation similar to previous reports from *Satb1* depleted
animals[84,85]. Both SATB1 and CTCF loops displayed a tendency to connect the TCR
enhancer (green arrow) to the regions inside the locus. The presented loops were
called with low stringency parameters and with a different set of binding sites
compared to the rest of our study due to technical details explained in the methods
section. All tracks refer to murine thymocytes. **d** SATB1-dependent promoter-
enhancer regulatory loops control the expression of both *Rag1* and *Rag2* genes.
Legend: th – thymocytes, DP – CD4⁺CD8⁺ T cells, RKO – *Rad21*ᶠˡ/ᶠˡ*Cd4*-Cre⁺ and SKO –
*Satb1*ᶠˡ/ᶠˡ*Cd4*-Cre⁺. **e** RNA levels of *Rag1* and *Rag2* genes in WT and *Satb1* cKO (SKO)
thymocytes based on three biological replicates of RNA-seq experiments. Data are
presented as mean values ± s.d. FDR scores from DESeq2 analysis are provided.
**f** Protein levels of RAG1 and RAG2. The Western blot analysis was performed three
times for each protein. **g** Differential H3K27ac HiChIP loops reflected the dereg-
ulation of *Traj* regions.

link between chromatin organization and tissue-specific transcrip-
tional regulation.

One of our goals was to provide molecular mechanisms such as
the 3D chromatin organization of T cells in order to explain the phe-
notypical malformations observed in the *Satb1* cKO mice. *Satb1*-defi-
cient mice suffer from multiple health problems, markedly resembling
autoimmunity. Previous research suggested a cell-extrinsic mechan-
ism of autoimmunity based on the deregulation of regulatory T cells[17].
Here we presented that SATB1 is a regulator of several genes involved
in T cell development, such as *Bcl6*, *Tcf7*, *Lef1*, *Cd6*, *Cd8*, *Lta*, *Ltb* and
others. The individual deregulation of most of these genes would also
result in deregulated immune responses. A great example for a cell-
intrinsic mechanism of autoimmunity in the *Satb1* cKO is the SATB1-
dependent spatial rearrangement of the TCRα enhancer and the TCR
locus per se, controlling TCR recombination. A previous study linked
the deregulation of the TCRα locus identified in the *Satb1* cKO to the
downregulation of the *Rag1* and *Rag2* genes[85]. In addition to this
deregulation and the SATB1-dependent regulatory loops at the *Rag*
locus, we also revealed the deregulation of chromatin accessibility and
H3K27ac looping at the TCR locus. This was correlated with the SATB1-
dependent chromatin loops and ultimately the defective usage of the
different TCR segments. Since the recruitment of RAG proteins is
based on the epigenetic status of a gene locus[43,44], we hypothesized
that the altered TCR organization, due to missing SATB1-dependent
loops, together with the disrupted recruitment of chromatin modify-
ing complexes interacting with SATB1[36–40], would be critical con-
tributors for the defective TCR arrangement in the *Satb1* cKO.
Moreover, the impact of the 3D organization of TCR and BCR, neces-
sary for proper VDJ recombination was recently highlighted[92,93].

Apart from the TCRα locus, the most affected gene regarding the
differential interaction analysis of regulatory loops between WT and
*Satb1* cKO thymocytes was *Bcl6*. BCL6 represents a master regulator of
Tfh[73–75] and innate-like T cells[94]. We have demonstrated that the Tfh
program is deregulated in the *Satb1* cKO utilizing several research
approaches. Moreover, the blockade at stage 0 (ST0) of iNKT devel-
opment in *Satb1*-deficient mice was previously reported[27], collectively
suggesting a potential link between SATB1-dependent regulation of
*Bcl6* and these developmental programs. BCL6 is known to function as
an antagonist of factors specifying other cell lineage fates[95,96], espe-
cially of PRDM1 and underlying Th17 lineage specification[73]. Indeed,
the increased IL-17 response we observed in the cytokine milieu of
*Satb1* cKO mouse sera (Fig. 1j), suggests a favored Th17 specification
due to downregulation of BCL6. However, since the depletion of SATB1
and its underlying regulome took place already during the intra-thymic
development (*Satb1*ᶠˡ/ᶠˡ*Cd4*-Cre⁺), we hypothesize that the increased IL-
17 cytokine levels could be primarily due to elevated γδT17 cells. Based
on the transcriptomic analysis depicted in Fig. 6c (RNA-seq, *Satb1* cKO/
WT) we have detected the overexpression of the TCRδ locus and
correspondingly also of the Vγ4⁺ and Vγ6⁺ chains, which are expressed

in γδT17 cells[97–100]. Moreover, the most overexpressed gene in *Satb1*
cKO thymocytes, as indicated by our RNA-seq experiments, was *Maf*
(encoding c-MAF; log2FC = 3.696 in female thymus and log2FC = 7.349
in male DP cells), which was shown to be essential for the commitment
of γδT17 cells[101]. Note that we used a *Cd4*-Cre⁺ mouse and our findings
cannot be directly related to the former report on dispensability of
SATB1 for the differentiation of peripheral Th17 cells, which were
based on different mouse models targeting later developmental stages
of T cells[102].

Our findings point to the regulatory overlap between TCR
recombination and transcriptional activity of master regulator genes in
T cells, collectively orchestrated via spatial chromatin arrangements
dependent on SATB1 and ultimately leading to the control of devel-
opmental decisions in the thymus. We provide a unique report on the
functional intersection between CTCF and a tissue-specific genome
organizer such as SATB1. Using the link between the altered 3D
enhancer network and the physiological deregulation of *Satb1* cKO
animals, we demonstrate the importance of the functional layer of
chromatin organization provided by transcription factors such as
SATB1. Our goal is to stir up a discussion about the existence of other
tissue- or cell-type-restricted factors, potentially contributing to the
higher complexity and direct regulatory potential of the 3D chromatin
architecture.

## Methods

### Animals and isolation of thymocytes

All experiments were conducted in accordance with the Laboratory
Animal Care and Ethics Committee of IMBB-FORTH. Animal work was
approved by the IMBB Institutional Animal Care and Ethics Committee.
All the experiments were performed on mice with C57BL/6 back-
ground. The generation of *Satb1*ᶠˡ/ᶠˡ mice was previously described[24].
The *Satb1* cKO (conditional knockout) mouse under study was created
by crossing the *Satb1*ᶠˡ/ᶠˡ mouse with a *Cd4*-Cre transgenic animal,
according to the scheme in Supplementary Fig. 1a, b. Validation of
*Satb1* cKO generation is provided in Supplementary Figs. 1c–g & 2a–d.
The oligonucleotide primers sequence used for genotyping were R10
(5′-CAG GCC ACA TTG TCC TAA CT-3′) and F9 (5′-TGC TCA TGT GGA
ATG TCG AG-3′). The animals used for the experiments were 4–8 weeks
old, unless otherwise specified. Primary thymocytes were resuspended
by rubbing and passing the thymus through a 40 μm cell strainer
(Falcon, 352340) into 1× PBS buffer. Cells were washed twice with 1×
PBS: cells were centrifuged at 500 g, 4 °C for 5 min, resuspended in
10 ml of 1× PBS and both steps were repeated. Prepared thymocytes
were either used directly for experiments or fixed with 1% methanol-
free formaldehyde (Pierce, 28908) at room temperature (RT) for
10 minutes while rocking. To quench the reaction, glycine was added
to 0.125 M final concentration and incubated at RT for 5 min, while
rocking. Cells were centrifuged at 1000 g, 4 °C for 5 minutes and
washed twice with ice cold 1× PBS.

## Flow cytometry

**Characterization of T cell populations in the *Satb1* cKO.** Depending on the experiment, we used either thymocytes or splenocytes. Splenocytes were isolated in the same way as thymocytes, but they were further resuspended in plain water for 3 seconds to lyse erythrocytes, with immediate dilution by HBSS (Gibco, 14180) to a final 1× concentration. One million cell aliquots were distributed into 5 ml polystyrene tubes (BD Falcon, 352052). For the experiments probing the percentage of apoptotic cells in tissues, we followed the PI/Annexin protocol (Biolegend, 640914). For staining with antibodies, we washed the cells once with Staining Buffer (1× PBS, 2% FBS, 0.1% NaN3) and then stained in 100 µl of Staining Buffer with 1 µl of antibodies at 4 °C for 30 minutes. The stained cells were washed with excess of Wash Buffer (1× PBS, 0.5% FBS) and then analyzed on FACSCalibur flow cytometer. Experiments probing cell populations with CD4/CD8/ CD62L/CD44 markers were FSC/SSC gated for the expected size of lymphocytes (gate G1; Supplementary Fig. 3b). The CD62L/CD44 populations were additionally gated for CD4+ cells (gate G2; Supplementary Fig. 3b). For the experiments with Annexin V and PI staining, the gating was set to exclude cellular debris (gate G3; Supplementary Fig. 3b). The antibodies used in flow cytometry experiments were anti-DNA PI-conjugated (Biolegend, 79997), anti-Phosphatidylserine FITC conjugated (Biolegend, 640906, 1/100), anti-CD4 PE-conjugated (Pharmingen, 553730, 1/100), anti-CD8a APC conjugated (Biolegend, 100712, 1/100), anti-CD4 PerCP conjugated (Biolegend, 100432, 1/100), anti-CD44 PE-conjugated (Pharmingen, 553134, 1/100) and anti-CD62L FITC conjugated (Biolegend, 104406, 1/100). Information on how the used antibodies were validated is provided in Supplementary Data 9. Data were collected using FlowJo and FCSalyzer was used for data analysis.

**SATB1 intracellular staining and *Satb1* cKO validation.** After extracellular staining of CD4 & CD8 antigens, intracellular staining for the SATB1 protein was performed as follows. Cells were firstly fixed for 10 min using 2% PFA (Pierce, 28908) in 1× PBS on ice. Fixed cells were washed with 1× PBS and then permeabilized using 0.3% Triton-X in 1× PBS for 5 minutes on ice. After three 5-minute washes in 1× PBS, cells were blocked for 30 min at room temperature with Blocking Buffer [0.4% acetylated BSA (Ambion, AM2614), 10% NGS and 0.1% Tween-20 in 4× SSC]. Staining for the primary antibody in question was performed using Detection Buffer (0.1% acetylated BSA and 0.1% Tween-20 in 4× SSC) using an antibody concentration of 0.5 µg per million of cells stained. Primary antibody staining was performed for 1 hour at room temperature. Following this, cells were washed thrice in Wash Buffer (0.1% Tween-20 in 4× SSC). Then, cells were incubated for 45 minutes at RT with a goat anti-mouse antibody conjugated with Alexa Fluor 647 (0.5 µg per million cells) in Detection Buffer. Finally, cells were washed thrice in Wash Buffer and following the final wash, cells were re-suspended in 1× PBS and analyzed on FACS Calibur flow cytometer. Antibodies used for these experiments: anti-SATB1 Antibody (Santa Cruz Biotechnology, sc-376096, 1/200), anti-CD4 FITC conjugated (Biolegend, 100406, 1/100), anti-CD8a PE-conjugated (Biolegend, 100708, 1/100), Goat anti-Mouse IgG (H + L) Cross-Adsorbed Secondary Antibody Alexa Fluor 647 (Invitrogen, A-21235, 1/250). Data were collected using FlowJo and FCSalyzer was used for data analysis.

## Infiltration of CD4+ cells in pancreas

Pancreas was isolated from three WT and three *Satb1* cKO mice of 120–136 days of age. Pancreas was cut in pieces and digested in 5 ml of 1 mg/ml collagenase (SIGMA, C2674) in PBS solution at 37 °C for 30 minutes. Samples were washed twice with 5% FBS in PBS and filtered through a polypropylene mesh. After centrifugation, cell pellets were resuspended in 1 ml of 0.05% Trypsin solution and incubated for 5 minutes at 37 °C. Cells were washed twice with ice-cold PBS and

eventually filtered through a 40 µm cell strainer and blocked in 5 ml of 5% FBS in PBS for 30 minutes at 4 °C. Cells were stained with 1:200 CD4-PE and CD8-APC for 30 minutes at 4 °C and then washed twice with 0.5% FBS in 1× PBS. Lastly, cells were resuspended in 2% FBS in 1× PBS and analyzed by flow cytometry.

## Characterization of the cytokine milieu

Cytokines were characterized and quantified from serum of 16 female mice (5 WT, 11 *Satb1* cKO) of varying age 1–7 months by the LEGENDplex (13-plex) Mouse Th Cytokine Panel V02 (Biolegend, 740741; Lot B289245) according to the manufacturer's instructions. Data were analyzed by the provided software LEGENDplex 8.0.

## Intraperitoneal glucose tolerance test

Groups of four WT and five *Satb1* cKO animals of 85 days (±8) of age were fasted for 6 hours. Weight and blood glucose levels were measured before and after the fasting period. 10% dextrose solution was injected intraperitoneally – the volumes were adjusted individually for each animal in order to inject 2 g of dextrose per kg of body mass. Blood (taken from tail) glucose levels were measured at given time points using Bayer Contour XT machine with Bayer Ascensia Contour Microfill Blood Glucose Test Strips. Animals were sacrificed and their pancreas was used for histology sections to demonstrate the disturbance of the islets of Langerhans.

## Histology and tissue sectioning

Samples were fixed in 4% formaldehyde in 1× PBS (pH 7.4) for 12–15 h at 4 °C. Tissues were rinsed in PBS and stored in PBS at 4 °C until embedding. For embedding, samples were dehydrated for 30 min in 70% ethanol, 2 × 30 minutes in 90% ethanol and 3 × 30 minutes in 100% ethanol – all at RT while stirring. Specimens were cleared for 2×60 min in xylol and then impregnated for 1–2 hours at 58 °C with paraffin. Samples were positioned in embedding moulds and left overnight to harden. Samples were sectioned on a sliding microtome to achieve 5–10 µm thin sections. The prepared sections on glass slides were deparaffinized for 30 min at 65 °C and then for 2 × 30 minutes in Neo-Clear (Merck Millipore, 109843). Samples were rehydrated for 2 × 10 min in 100% ethanol, 1 × 5 minutes in 90% ethanol, 1 × 5 minutes in 70% ethanol, 1 × 5 minutes in 50% ethanol, 1 × 5 min in 30% ethanol and 1 × 5 minutes in 1 × PBS. Samples were immersed in Haematoxylin bath for 5 minutes in dark and then washed by running water for 5–10 minutes and in 1 × PBS for 1 minute. Next, samples were immersed in Eosin bath for 30 seconds in dark and then washed by running water for 5–10 minutes and in 1× PBS for 1 min. Samples were dehydrated again by dipping ten times in 30%, 50% and 70% ethanol and then incubated for 2×2 minutes in 100% ethanol. Eventually, samples were incubated for 2×10 minutes in xylene and then mounted using Entellan® new (Merck Millipore, 107961).

## Transmission electron microscopy

For scanning electron microscopy (SEM), fresh thymi were cut into small blocks. Briefly, tissue was fixed for 2 hours with 2% paraformaldehyde – 2% glutaraldehyde in 0.1 M sodium cacodylate buffer. Samples were postfixed overnight in 1% osmium tetroxide (OTO method) and dehydrated in a graded series of ethanol. Specimens were coated in gold, mounted on aluminum stubs and examined with a JEOL JSM6390 LV scanning electron microscope (Peabody, MA) using an accelerating voltage of 15 kV.

## Micrococcal nuclease assay

Isolated thymocytes were crosslinked by adding 1/10th volume of Fixation Buffer [11% methanol-free formaldehyde (Pierce, 28908), 100 mM NaCl, 1 mM EDTA, 0.5 mM EGTA, 50 mM Hepes pH 8.0] for 10 minutes at room temperature while rocking. Crosslinked cells were quenched by adding glycine to 0.2 M final concentration (870 µl 2.5 M

 

stock), and incubating at room temperature for 5 minutes. Cells were centrifuged at 1000×g at 4 °C for 5 minutes and washed twice with 1× PBS. Pellets were resuspended in 60 µl of Lysis Buffer 3 (1% SDS, 50 mM Tris-HCl pH 8.0, 10 mM EDTA, 1× protease and phosphatase inhibitors), and incubated for 20 minutes at room temperature. Samples were diluted to 600 µl using cold TE Buffer + 1× protease inhibitors and centrifuged at 1000× g for 5 minutes at 4 °C. Nuclei pellets were resuspended in MNase Reaction Buffer (10 mM Tris-HCl pH 7.5, 10 mM NaCl, 3 mM MgCl$_2$, 1 mM CaCl$_2$, 4% NP40, 1 mM PMSF) and different concentrations of MNase enzyme (Sigma Aldrich, N3755) were added. Samples were incubated for 5 minutes at 37 °C and the reaction was stopped by adding Stop Reaction Buffer (10 mM EDTA, 20 mM EGTA). Samples were centrifuged and resuspended in 125 µl Elution Buffer (10 mM Tris-HCl pH 8.0, 5 mM EDTA, 300 mM NaCl, 1% SDS) and incubated at 65 °C overnight. Samples were treated with RNase A, Proteinase K and DNA was isolated by phenol extraction and ethanol precipitation.

### Detection of autoantibodies

The WT pancreas sample was prepared as previously described. 5 µm thick sections were deparaffinized at 55 °C for 8 min and then processed in the following solutions: 2 × 3 min in Neo-Clear, 2 × 3 min in 100% ethanol, 1 × 3 min in 95% ethanol, 1 × 3 minutes in 70% ethanol, 1 × 3 min in 50% ethanol. Samples were then rinsed with water and carefully dried with a paper towel. Tissue was circled with a PAP pen, let dry for 1 minute and then dipped in PBS for 2 min. Antigens were retrieved by incubation with 130 µl of TE-Triton-PK solution (2 ml TE buffer, 10 µl 0.5% Triton X-100, 40 µg Proteinase K) in a humidified chamber at 37 °C, for 12 min. Samples were then washed twice with TBST buffer (10 mM Tris-HCl pH 8.0, 150 mM NaCl, 0.05% Tween-20) for 3 minutes each. Samples were blocked by incubation with 5% normal goat serum (NGS) in TBST buffer at RT for 30 min in a humidified chamber. Samples were incubated at 4 °C overnight with blood serum collected from two WT and four *Satb1* cKO animals of 4–7 months of age. The serum was diluted 1:10 in 5% NGS-TBST and 5% NGS-TBST was used as a negative control. Samples were washed twice with TBST, 5 minutes each. Samples were incubated with a goat anti-mouse IgG antibody conjugated with Alexa Fluor 594 (H + L; Invitrogen, A-11005, 1/500) in TBST at RT for 1 h. Samples were washed three times, 5 min each, with TBST and incubated with 1 µM DAPI solution in 5% NGS-TBST at RT for 10 minutes. Samples were washed three times, 5 min each, with TBST and mounted with Mowiol on glass slides.

### Hi-C and HiChIP experiments

**Generation of proximity-ligated contacts.** A detailed step-by-step protocol used in this study was recently published[103]. A biological duplicate was used for each sample. Both Hi-C and HiChIP experiments were performed identically until the chromatin immunoprecipitation step. Aliquots of 10 million isolated thymocytes resuspended in 1× PBS were fixed by adding 1/10th volume of fixation butter [11% methanol-free formaldehyde (Pierce, 28908), 100 mM NaCl, 1 mM EDTA, 0.5 mM EGTA, 50 mM Hepes pH 8.0] with rocking at RT for 10 minutes. To quench the reaction, glycine was added to 0.125 M final concentration and incubated at RT for 5 minutes, while rocking. After two washes with 1× PBS, cell pellet was resuspended in 500 µl of ice-cold Hi-C Lysis Buffer (10 mM Tris-HCl pH 8.0, 10 mM NaCl, 0.2% NP40, 0.5 mM PMSF) and rotated at 4 °C for 1.5 hours. Cells were centrifuged at 2500 g, at 4 °C for 5 min and the supernatant was discarded. The cell pellet was washed once with 500 µl of ice-cold Hi-C Lysis Buffer and then resuspended in 100 µl of 0.5% SDS. Cells were incubated at 62 °C for 10 minutes and then combined with 296 µl of H2O and 50 µl of 20% Triton X-100. Samples were incubated at 37 °C for 15 minutes and then combined with 50 µl of 10× DpnII Buffer and 200 U of DpnII restriction enzyme (NEB, R0543M) and digested for additional 16 hours at 37 °C while shaking (160 rpm). The restriction enzyme was inactivated at 62 °C for 20 min and the nuclei were centrifuged at 2500 g, at 4 °C for 6 minutes. The supernatant was discarded and the nuclei were resuspended in 300 µl Fill-in Buffer containing 30 µl Klenow Buffer 10× (NEB, M0210L), 15 µl 1 mM Biotin-16-dCTP (Jena Bioscience, NU-809-BIO16-L), 1.5 µl 10 mM dATP (Promega, U1240), 1.5 µl 10 mM dGTP (Promega, U1240), 1.5 µl 10 mM dTTP (Promega, U1240), 12 µl 5 U/µl DNA Polymerase I, Klenow Fragment (NEB, M0210L) and 238.5 µl water. The biotinylation mixture was incubated at 37 °C for 30 minutes with rotation. SDS was added to a final concentration of 0.5% to inactivate the Klenow enzyme. Triton X-100 was added to 1% final concentration and samples were incubated at 37 °C for 5 minutes. Samples were centrifuged at 2500 g, at 4 °C for 10 minutes and the supernatant was discarded. The nuclei pellet was resuspended in the Ligation Buffer containing 120 µl 10× NEB T4 DNA Ligase Buffer supplemented with 10 mM ATP (NEB, B0202), 60 µl 20% Triton X-100 (1% final), 6 µl 2% (20 mg/ml) BSA, 40 µl 30% PEG 6,000 (1% final), 5 µl 400 U/µl T4 DNA Ligase (NEB, M0202L) and 969 µl water. The samples were incubated for 6 hours at RT with mild rotation. Nuclei were centrifuged at 2,400 g, at RT for 15 minutes and the supernatant was discarded. The pellet was resuspended in 60 µl Lysis Buffer (1% SDS, 50 mM Tris-HCl pH 8.0, 20 mM EDTA, 1× protease inhibitors) and incubated at RT for 15 minutes. The lysate was diluted to 600 µl using ice-cold TE Buffer + 1× protease inhibitors and then sonicated with a Labsonic M – Tip sonicator for 3 cycles (30 seconds ON/OFF, 40% power). The sonicated material was centrifuged at 16,000 g, at RT for 15 minutes and the supernatant was collected into a new tube. Samples from the same genotype were merged and then split again: separately 100 µl for Hi-C and 450 µl for HiChIP.

Hi-C samples were combined with two volumes of Hi-C Elution Buffer (10 mM Tris-HCl pH 8.0, 5 mM EDTA, 300 mM NaCl, 1% SDS) and incubated at 65 °C overnight. Decrosslinked material was diluted to 500 µl with TE Buffer and treated with RNase A and Proteinase K as previously described. DNA was purified using a ChIP DNA Clean & Concentrator kit following the manufacturer's instructions (Zymo Research, D5205). Purified DNA was quantified using Qubit dsDNA BS Assay Kit (Invitrogen, Q32853) checked on an agarose gel for shearing efficiency and 100 µg were used for library construction.

HiChIP samples were combined with Triton X-100 to 1% final concentration and samples were incubated at 37 °C for 15 minutes. Samples were combined with an equal volume of 2× ChIP Binding Buffer (20 mM Tris-HCl pH 8.0, 2 mM EDTA, 0.2 % sodium deoxycholate, 2× protease inhibitors). Chromatin preclearing and antibody binding to magnetic beads were performed as described in the ChIP protocol. The following antibodies were used for the immunoprecipitation step: 8 µg of custom-made David's Biotechnologies SATB1 long isoform (targeting only the extra peptide present in the long isoform[54]), 7 µg anti-CTCF (Abcam, ab70303) and 2 µg anti-H3K27ac (Abcam, ab4729).

Antibody-coupled beads were incubated with chromatin at 4 °C with rotation for 16 hours. Beads were washed five times with ice cold RIPA Buffer (50 mM Hepes pH 8.0, 1% NP-40, 0.70% Na-Deoxycholate, 0.5 M LiCl, 1 mM EDTA, 1× protease inhibitors) and then twice with TE Buffer. After the first wash with TE Buffer, resuspended beads were transferred into a new tube. Immune complexes bound to beads were eluted in 125 µl Hi-C Elution Buffer (10 mM Tris-HCl pH 8.0, 5 mM EDTA, 300 mM NaCl, 1% SDS) at 65 °C for 16 hours. Decrosslinked material was diluted to 250 µl with TE Buffer and treated with RNase A and Proteinase K as previously described. DNA was purified using a ChIP DNA Clean & Concentrator kit following the manufacturer's instructions (Zymo Research, D5205). Purified DNA was quantified using Qubit dsDNA HS Assay Kit (Invitrogen, Q32854) and used for library construction.

**Biotin pull-down and library construction.** Samples were brought to 25 µl with water. 5 µl and 20 µl for HiChIP / Hi-C samples, respectively, of Dynabeads MyOne Streptavidin C1 beads (Invitrogen, 65001) were

washed with 500 µl Tween Wash Buffer (5 mM Tris-HCl pH 7.5, 0.5 mM EDTA, 1 M NaCl, 0.05% Tween-20). Beads were resuspended in 25 µl of 2× Biotin Binding Buffer (10 mM Tris-HCl pH 7.5, 1 mM EDTA, 2 M NaCl) and combined with samples. Samples were incubated at RT for 20 minutes. Beads were separated on a magnet and washed twice with 400 µl of Tween Wash Buffer. Beads were washed by 100 µl of 1× TD Buffer (10 mM Tris-HCl pH 7.5, 5 mM MgCl₂, 10% Dimethylformamide). Beads were resuspended in 25 µl of 2× TD Buffer and combined with Tn5 enzyme from the Nextera DNA Sample Preparation Kit (Illumina, FC-121-1030) and water to final volume 50 µl. The amount of Tn5 enzyme was adjusted according to the input DNA amount: 4.5 µl for Hi-C libraries, 1.5 µl for SATB1 HiChIP and 1 µl for other HiChIP libraries. The reaction was incubated at 55 °C for 10 minutes. The beads were collected with a magnet and the supernatant was discarded. Beads were resuspended in 300 µl of Strip Buffer (0.15% SDS, 10 mM Tris-HCl pH 8.0, 50 mM EDTA) and incubated for 5 minutes at RT to strip off and deactivate Tn5. Beads were washed once with 400 µl of Tween Wash Buffer and once with 500 µl of 10 mM Tris-HCl (pH 8.0). The beads were resuspended in 50 µl of the following PCR master mix with indexed primers from the Nextera DNA Sample Preparation Index Kit (Illumina, FC-121-1011): Phusion HF 2× (NEB, M0531L) 25 µl, Nextera Index 1 (N7XX 5.5 µM) 1 µl (1.5 for Hi-C), Nextera Index 2 (N5XX 5.5 µM) 1 µl (1.5 for Hi-C) and water 23 µl (22 for Hi-C). The PCR reaction was performed following the program 72 °C for 5 minutes and repeated cycles of 98 °C for 15 seconds, 63 °C for 35 seconds, 72 °C for 1 minute.

The number of PCR cycles was estimated based on post-ChIP quantification and amplification was 6 cycles for Hi-C libraries, 11 cycles for SATB1 HiChIP and 13 cycles for the other HiChIP libraries. DNA libraries were purified and size-selected using AMPure XP beads, quantified by Qubit and analyzed on a Bioanalyzer, as previously described. The DNA Libraries were sequenced on an Illumina® HiSeq 4000 2× 75 bp platform by the sequencing facility at IKMB, Kiel University, Germany.

**Data processing.** Raw reads were mapped with bowtie2[104] to the mm10 genome and fully processed using the HiC-Pro pipeline (version 2.11.1)[105] with default parameters. All biological replicates were processed individually to assess their quality and then combined for downstream analyses and visualization. The same parameters were used for the re-analysis of publicly available Hi-C datasets from untreated and CTCF-depleted mESCs[6] and WT and *Rad21*^fl/fl*Cd4*-Cre⁺ DP murine thymocytes[56]. Note that the genetically induced depletion of RAD21 is not absolute and thus the observed phenotype does not properly reflect the perturbation. In a degron system, the complete loss of cohesin resulted in a more profound effect and the vast majority of loops and TADs were eliminated[7], yet for the sake of our comparison we appreciate the *Rad21*^fl/fl*Cd4*-Cre⁺ knockout, which specifically impacts DP T cells. Note also that to our knowledge there are no Hi-C data available for T cells with depleted CTCF; however, since TADs are conserved among different cell types[1], the comparison between T cells and mESCs could be performed. The HiChIP datasets were additionally processed by FitHiChIP[106]. Unless stated otherwise, the following parameters were used to call HiChIP loops: 5,000 kbp resolution, 20000–2000000 distance threshold, FDR 0.01, coverage-specific bias correction, merged nearby peak to all interactions.

To identify differentially interacting regions between SATB1 and CTCF HiChIP experiments, we utilized raw matrices at 100 kbp and 500 kbp resolutions. Matrices were analyzed using diffHic[107] and differentially interacting regions were determined as differential interactions with FDR ≤ 0.05. For the differential analysis of H3K27ac WT and *Satb1* cKO HiChIP datasets, we utilized the differential analysis pipeline from FitHiChIP[106] using default parameters and only utilized loops that were classified as differential in 3D but not in 1D. These were loops that showed differences in interaction counts but no significant differences in H3K27ac occupancy at loop anchors. Since some regions contained

H3K27ac loops from both under- and over-interacting categories, we furthermore calculated a difference between the number of under- and overinteracting loops.

Binding site datasets, needed to call HiChIP loops, were either derived from HiChIP data or an external ChIP-seq dataset. The SATB1 binding sites were extracted from another HiChIP experiment with >60% Dangling End Pairs (~280 million reads), using the PeakInferHiChIP.sh script from FitHiChIP (--nomodel --extsize 147; filter peaks with >2.5 enrichment). SATB1 binding sites were compared to a published SATB1 ChIP-seq dataset (GSM1617950)[85]. The H3K27ac peaks were derived similarly from the HiChIP datasets (separately for two biological replicates and then merged). The CTCF peaks were also derived the same way and motif analysis with MEME[108] validated the high enrichment of the CTCF binding motif in the HiChIP-derived peaks, confirming its specificity. However, due to a relatively low number of HiChIP-derived CTCF binding sites, for FitHiChIP loop calling and other computational analyses, we employed a CTCF ChIP-seq dataset from the ENCODE project (ENCFF714WDP)[109]. Only for the visualization purposes, we used combined datasets (mergeBed command of bedtools)[110] of our HiChIP-derived binding sites and publicly available ChIP-seq datasets for both SATB1 (GSM1617950)[85] and CTCF (ENCFF714WDP)[109]. Since our antibodies were specific for the long SATB1 isoform, we used the dataset combined with the public ChIP-seq to ensure that all types of SATB1 peaks were shown.

HiCHIP loops were classified according to the presence of a thymus-specific enhancer[68] and/or gene promoter (TSS + 5 kbp) in the loop anchors. Each category of loops was characterized by an overlap with other factors HiChIP loops. For each overlap, the Fisher's exact test Odd ratios and P-values are provided in Supplementary Data 5. The Odd-ratio values were calculated as: (Number of factor2 loops in a category / Number of total factor2 loops) / (Number of factor1 loops in a category / Number of total factor1 loops). Additionally, the Fisher's exact test Odd ratios and P-values are also provided for overlaps between loop anchors and differentially expressed genes.

At the TCR locus, whole DNA segments are missing in some cells due to V(D)J recombination. Thus, in the cell population some regions of TCR are underrepresented compared to other genomic loci and this fact penalizes loop or binding site calling at the TCR locus. For this reason, for the TCR analysis we used different datasets with adjusted and more relaxed parameters to compensate for this. For the purpose of loop calling in HiChIP experiments, we utilized the combined binding site datasets of our HiChIP-derived binding sites and publicly available ChIP-seq datasets for both SATB1 (GSM1617950)[85] and CTCF (ENCFF714WDP)[109]. Furthermore, the following parameters were used: 10,000 kbp resolution, 20000–2000000 distance threshold, FDR 0.05, coverage-specific bias correction, merged nearby peak to all interactions.

For operations with matrices, the datasets were processed using hicexplorer[111]. Hi-C and HiChIP matrices were normalized (*hicNormalize* command) to the smallest dataset from each compared pair, i.e. WT vs SKO Hi-C, SATB1 vs CTCF HiChIP and WT vs SKO H3K27ac HiChIP. Moreover, matrices were corrected using a KR balancing method[112] (*hicCorrectMatrix* command). The analysis of high-order chromatin organization was performed using hicexplorer[111], HOMER[113] and GENOVA[57]. The datasets used for analysis by GENOVA were at 40 kbp resolution and ICE normalized[114]. Differential analysis of TADs was performed using TADCompare[115] at 100 kbp resolution. Visualization of matrices was done by hicexplorer[111], pyGenomeTracks and/or by Juicebox[116]. APA and ADA scores[1] were calculated and visualized by SIPMeta using the z-score parameter[61] and/or by Juicer Tools[116].

**Chromosome conformation capture**
Isolated thymocytes were crosslinked by adding 1/10ᵗʰ volume of Fixation Buffer [11% methanol-free formaldehyde (Pierce, 28908), 100 mM NaCl, 1 mM EDTA, 0.5 mM EGTA, 50 mM Hepes pH 8.0] for

10 min at room temperature while rocking. Crosslinked cells were quenched by adding glycine to 0.2 M final concentration (870 µl 2.5 M stock), and incubating at room temperature for 5 minutes. Cells were centrifuged at 1000×g at 4 °C for 5 min and washed twice with 1× PBS. The cell pellet was resuspended in 500 µl ice-cold Hi-C Lysis Buffer (10 mM Tris-HCl pH 8.0, 10 mM NaCl, 0.2% NP-40, 1× protease inhibitors) and rotated at 4 °C for 1.5 hours. After centrifugation at 2500× $g$ at 4 °C for 5 min the supernatant was discarded and the pellets were resuspended in 100 µl of 0.5% SDS and incubated at 62 °C for 10 minutes. 300 µl of ddH$_2$O was added and SDS was quenched by adding 50 µl of 20% Triton-X 100. Afterwards, 50 µl of 10× BglII Buffer was added together with 200U of BglII restriction enzyme (Enzyquest, REc008S). Nuclei were incubated for 16 hours at 37 °C, while shaking (300 rpm). BglII was inactivated by adding SDS (to 1% final) and BglII was heat inactivated at 62 °C, for 20 min and let cool down to room temperature. SDS was quenched by the addition of 60 µl 20% Triton (final 2%) and incubated at 37 °C for 15 min. In the same tube Ligation Reaction Buffer was added [120 µl 10× T4 Ligase Buffer, 12 µl 10 mM ATP, 6 µl 2% BSA, 40 µl 30% PEG 6000, 5 µl 400U/ µl T4 Ligase (Enzyquest PD006S), 417 µl ddH$_2$O] and ligation reaction was performed at room temperature for 6 hours. Nuclei were centrifuged and resuspended in 125 µl Elution Buffer (10 mM Tris-HCl pH 8.0, 5 mM EDTA, 300 mM NaCl, 1% SDS) and incubated at 65 °C overnight. Samples were treated with RNase A, Proteinase K and DNA was isolated by phenol extraction and ethanol precipitation.

**Stranded-total-RNA sequencing**
**Experimental protocol.** A biological triplicate was used for each genotype. Freshly isolated thymocytes from female animals were resuspended in 1 ml of TRIzol Reagent (Invitrogen, 15596026) and RNA was isolated according to manufacturer's protocol. The aqueous phase with RNA was transferred into a tube and combined with 10 µg of Linear Acrylamide (Ambion, AM9520), 1/10 of sample volume of 3 M CH3COONa (pH 5.2), 2.5 volumes of 100% Ethanol and tubes were mixed by flipping. Samples were incubated at –80 °C for 40 min. Samples were centrifuged at 16,000 g, at 4 °C for 30 minutes. The supernatant was removed and the pellet was washed twice with 75% Ethanol. The air-dried pellets were resuspended in 40 µl RNase-free water and incubated at 55 °C for 15 min to dissolve RNA. To remove any residual DNA contamination, RNase-free DNase Buffer was added to the samples until 1× final concentration together with 20 units of DNase I (NEB, M0303L) and incubated at 37 °C for 20 minutes. Samples were then purified using RNeasy Mini Kit (Qiagen, 74104) according to the manufacture's protocol. RNA quality was evaluated using Agilent 2100 Bioanalyzer with Agilent RNA 6000 Nano Kit (Agilent Technologies, 5067–1511). Libraries were prepared using an Illumina® TruSeq® Stranded Total RNA kit with ribosomal depletion by Ribo-Zero Gold solution from Illumina® according to the manufacturer's protocol and sequenced on an Illumina® HiSeq 4000 (2× 75 bp).

**Data processing.** Raw reads were mapped to the mm10 mouse genome using HISAT2[117]. Only mapped, paired reads with a map quality >20 were retained. Transcripts were assembled with StringTie[118] using an evidence-based Ensembl-Havana annotation file. Transcripts and genes were summarized using featureCounts[119] and statistically evaluated for differential expression using DESeq2[120]. When application required an intra-sample transcript comparison, DESeq2 values were further normalized to the gene length. The functional analyses were performed by g:Profiler[121]. The plots depicting enriched BP terms and KEGG pathways were generated by presenting the top 20 pathways/ terms with the lowest p-values.

**ATAC-seq**
**Experimental protocol.** A biological triplicate was used for each genotype. The ATAC-seq experiment was performed according to the

Omni-ATAC protocol previously published[122], with modifications. Murine thymocytes were isolated as previously described, without fixation. To ensure the presence of only viable cells, cells were separated using Lympholyte®-M (Cedarlane, CL5030) according to the manufacturer's protocol. Isolated cells were washed twice with 1× PBS and aliquots of 10,000 cells were used for analysis. The cell pellet was gently resuspended by pipetting up and down three times in 50 µl of ice cold ATAC-RSB-NTD Buffer (10 mM Tris-HCl pH 7.5, 10 mM NaCl, 3 mM MgCl$_2$, 0.1% NP40, 0.1% Tween-20, 0.01% Digitonin) and incubated on ice for 3 min. Cell lysis was stopped by adding 1 ml of cold ATAC-RSB-T Buffer (10 mM Tris-HCl pH 7.5, 10 mM NaCl, 3 mM MgCl$_2$, 0.1% Tween-20) and inverting the tube three times to mix. Nuclei were centrifuged at 1000 g, at 4 °C for 10 mins. The pellet was resuspended in 50 µl of Transposition Mix [25 µl 2× TD buffer (20 mM Tris-HCl pH 7.6, 10 mM MgCl2, 20% Dimethyl Formamide – before adding DMF, the pH was adjusted to 7.6 with 100% acetic acid), 2.5 µl transposase (100 nM final), 16.5 µl PBS, 0.5 µl 1% digitonin, 1 µl 5% Tween-20, 4.5 µl H2O] by pipetting up and down six times. The reaction was incubated at 37 °C for 30 min with occasional pipetting. DNA was purified with a DNA Clean & Concentrator-5 Kit (Zymo Research, D4013) according to the manufacturer's protocol. DNA was eluted in 20 µl of Elution Buffer and all the material was used in a PCR reaction, together with 25 µl Phusion HF 2× Master Mix (NEB, M0531L) and 2.5 µl of each Nextera Index 1 (N7XX) and Nextera Index 2 (N5XX) primers from a Nextera DNA Sample Preparation Index Kit (Illumina, FC-121-1011). PCR was performed according to the following protocol: 72 °C for 5 min, 98 °C for 1 min and 5 cycles of 98 °C for 15 s, 63 °C for 35 s, 72 °C for 1 minute. The samples were put on ice and 5 µl of the preamplified mixture was combined with 15 µl of qPCR Master Mix using the following set-up: water 3.25 µl, primer ad1 0.5 µl, primer ad2 0.5 µl, 20× SYBR Green 0.75 µl, Phusion HF 2× Master mix 5 µl and pre-amplified sample 5 µl. The qPCR reaction was run for 20 additional cycles following the program 98 °C for 1 min and 5 cycles of 98 °C for 15 s, 63 °C for 35 s, 72 °C for 1 min.

Based on the Rn (Fluorescence) vs Cycle linear plot, a cycle with 1/4 up to 1/3 of the maximum fluorescence level was determined. This was the number of additional PCR cycles to run on the pre-amplified libraries which were stored on ice until this point. The final amplified libraries were purified using a DNA Clean & Concentrator-5 Kit (Zymo Research, D4013) according to the manufacturer's protocol and eluted in 20 µl of Elution Buffer. Libraries were quantified by Qubit and analyzed on a Bioanalyzer as previously described, followed by two-sided size selection using AMPure Beads. Libraries were sequenced on an Illumina® HiSeq 4000 2× 75 bp platform by the sequencing facility at IKMB, Kiel University, Germany.

**Data processing.** To acquire BAM files and ATAC-seq peaks, raw data were fully processed by the esATAC pipeline[123]. To identify the regions with differential accessibility between genotypes, all esATAC-called peaks across samples and replicates were pooled and tested for differences in accessibility levels. ATAC-seq counts were calculated for each peak, each replicate and each condition using FeatureCounts[119] and used as an input for edgeR[124] with the standard parameters. A cutoff of |logFC| ≥ 1 and p-value ≤ 0.01 was used to determine the differentially accessible regions.

To assess the accessibility around SATB1 binding sites BigWig files were generated. Mitochondrial reads and PCR duplicates were removed from all bam files using samtools[125] and Picard MarkDuplicates (http://broadinstitute.github.io/picard/), respectively. Low quality and unmapped reads were removed using the following samtools command: samtools view h -b -F 1804 -f 2 -q 30. The final bam files were merged for the two conditions using samtools and RPKM normalized BigWig files were generated using deeptools[126] with the following parameters: -of bigwig --effectiveGenomeSize 2652783500 --normalizeUsing RPKM -bl mm10.blacklist.bed -bs 1. mm10 blacklisted regions

were downloaded from ENCODE. Moreover, using the merged bam files, BigWig files with the log2 ratio between the normalized reads of *Satb1* cKO and WT thymocytes were generated using deeptools bamCompare. The parameters were the same as above. SATB1 binding sites were first centered and then extended by 250 bp upstream and downstream. For each bp position, ATAC-seq signal was calculated using the generated BigWig files. Similarly, each TSS of each gene was centered and extended by 1 kbp upstream and downstream. For each base of each gene, an average log2 fold change of normalized accessibility score (*Satb1* cKO vs WT) was plotted. To calculate the accessibility changes along the entire genes, genes and upstream and downstream regions were divided into bins and average log2 fold change of normalized accessibility score for each bin was plotted.

## Immunofluorescence experiments

Glass coverslips were coated by dipping in 0.1 mg/ml poly-D-lysine solution (Sigma Aldrich, P6407). Freshly isolated thymocytes were attached to the coated coverslips. Attached cells were washed once with 1× PBS and then fixed for 10 minutes on ice with 4% formaldehyde (Pierce, 28908) in 1× PBS. Fixed cells were permeabilized with 0.5% Triton-X in 1× PBS for 5 minutes on ice. Cells were washed three times with 1× PBS for 5 minutes each and blocked for 30 minutes at RT with Blocking Buffer [0.4% acetylated BSA (Ambion, AM2614) in 4× SSC] in a humidified chamber. Cells were incubated for 1.5 hours at RT with an antibody in Detection Buffer (0.1% acetylated BSA, 4× SSC, 0.1% Tween 20) in a humidified chamber. An antibody against the phosphorylated form of RNA polymerase II (phosphorylated Ser 5 of CTD, CTD4H8-Alexa Fluor labeled; Covance, A488-128L, 1/500) and anti-BCL6 antibody (Santa Cruz Biotechnology, sc-7388, 1/50) were used. The excess of antibodies was washed away by three washes for 5 minutes each with Washing Buffer (4× SSC, 0.1% Tween 20). For the BCL6 IF experiment, cells were incubated for 60 minutes at RT with a secondary goat anti-mouse antibody conjugated to Alexa Fluor 594 (Invitrogen, A-11005, 1/250) in Detection Buffer (0.1% acetylated BSA, 4× SSC, 0.1% Tween 20) in a humidified chamber. The excess was washed away by three washes for 5 minutes each with Washing Buffer (4× SSC, 0.1% Tween 20). The cells were mounted in a hardening ProLong Gold medium with DAPI (Invitrogen, P36935). Images were taken using an inverted microscope DMI6000 CS with laser scanning confocal head Leica TCS SP8, equipped with a 63×/1.40 oil immersion objective. Images were analyzed using the Fiji software[127]. Cells were manually selected and signal was measured as an integrated signal density from summed z-stacks.

## Western blot

For Western blot experiments we used anti-BCL6 antibody (Santa Cruz Biotechnology, sc-7388 & sc-550543, 1:250), anti-RAG1 (Abcam, ab172637, 1:2500), anti-RAG2 (Proteintech, 11825-1-AP, 1/1000) and β-Actin as a loading control (ORIGENE, TA811000, 1:500).

## Cultivation of neonatal thymi

Thymi of neonatal WT and *Satb1* cKO mice were collected and embedded in collagen. Thymi were cultivated in a medium (10% FBS, RPMI, Hepes, Pen-Strep, Glutamine, β-mercaptoethanol) for 30 hours and monitored with the Operetta high content screening microscope (PerkinElmer) to detect cells exiting the thymus. Images were pre-processed with the in-built analysis software Harmony 4.1 and then analyzed with custom-made macros in Fiji software[127]. Random shifts were corrected by a StackReg plugin (Rigid Body transformation). Areas without any distortion were selected and cells outside the thymus were counted using Find Maxima function of Fiji. Each selection was normalized to the area size and for each animal the selections were averaged. The final result represents an average from two animals for each genotype.

## Linear regression model

A linear regression model was built in R and used to identify the impact of individual variables from our datasets on log2FC RNA-seq values. The predictors used are described below: Each gene was binned into 3 bins. Moreover, two extra bins upstream of the TSS of each gene (upstream region: −4 kbp to −2 kbp and promoter region: −2kb to TSS) and two extra bins downstream of the transcription termination site (TTS to +2 kbp and +2 kbp to +4 kbp) were used. SATB1 binding occupancy in WT cells was determined via a binary score for each bin of each gene. "1" indicated the presence of a SATB1 peak and "0" the absence of a SATB1 peak overlapping with the corresponding bin.

To quantify ATAC-seq signal differences between the *Satb1* cKO and WT thymocyte samples, the following calculation for each bin of each gene was used: log10[(*Satb1* cKO Normalized Reads + 0.01) / (WT Normalized Reads + 0.01)] * log2(Total Normalized Reads + 1)

To utilize SATB1 and CTCF loops as predictors, we assigned to each gene the number of times it overlapped with the anchors of a SATB1 or CTCF loop. In case that both anchors of the same SATB1/CTCF loop overlapped a gene, only one was counted. The number of times, the anchors of a SATB1 loop were found to connect an enhancer with a gene, was also used as a predictor. The same metrics were calculated for overinteracting and underinteracting H3K27ac loops.

## Complementary bioinformatics analyses

**Identification of SATB1 binding in H3K27ac HiChIP anchors.** Loop anchors from WT H3K27ac HiChIP loops were pre-processed by extracting the anchors from both sides of loops and then merging them into non-overlapping unique regions (mergeBed command of bedtools)[110]. The resulting regions were converted from mm10 to mm9 using CrossMap[128] and were analyzed using the enrichment analysis ChIP-Atlas[42] against all the available murine ChIP-seq datasets from the blood cell type class and compared to 100× random permutations.

**Publicly available ChIP-seq datasets.** To validate the reliability of our SATB1 binding sites dataset, we compared it to two SATB1 ChIP-seq datasets that were prepared using standard SATB1 antibodies non-selectively targeting all SATB1 isoforms (Abcam, ab109122[85] & ab70004[17]). The first SATB1 ChIP-seq (GSM1617950[85]) is already provided with the processed SATB1 peaks, hence these were only converted from mm9 to mm10 mouse genome assembly using CrossMap[128] to match our dataset. The second dataset (DRR061108[17]) was completely re-analyzed: sequencing reads were mapped to mm10 genome using Bowtie2[104] and unmapped reads and reads with Q < 20 were filtered out using Samtools[125]. Peaks were called using MACS2[129] with default parameters and providing the available input dataset. Only sequences unambiguously mapped to chromosomes (chr1-chr19 & chrX; i.e. excluding chrUn and Random assemblies) were kept for the analysis. Characteristics of each dataset and their mutual similarity was assessed using Jaccard statistic and Relative Distance metric of Bedtools[110,130]. The genomic feature association analysis (Genome Ontology) was performed using the AnnotatePeak function of Homer[113].

Additionally, we utilized the CTCF ChIP-seq dataset from the ENCODE project (ENCFF714WDP)[109]. For the analysis of enhancers, in relation to SATB1 and CTCF loops, we utilized the thymus-specific list of enhancers previously generated within the ENCODE project (GSE29184)[68] – after extending the center of each enhancer by 50 bp upstream and downstream. Moreover, for visualization purposes we also utilized the H3K4me3 (ENCFF200ISF) and H3K4me1 (ENCFF085AXD) ChIP-seq datasets for thymus, from the ENCODE project[131]. The H3K27ac ChIP-seq datasets were from WT and *Rad21*^fl/fl^*Cd4*-Cre^+^ DP cells[67] and they were first converted from mm9 to mm10 using CrossMap[128]. An average file based on two biological replicates (GSM1504384 + 5 for WT and GSM1504386 + 7 for *Rad21* cKO) was created and used for visualization.

**Overlap score calculation for HiCHIP loops**. The overlap score between SATB1 and CTCF loops was calculated as (number of overlapping bp)/(bp size of a loop). Overlaps between the two types of loops were found using bedtools[110] and the above score was calculated using R. A score of one indicates either 100% overlap or engulfment of a loop dependent on one factor in a loop dependent on another factor. A score of zero indicates no overlap. In cases where a loop dependent on one factor intersected with multiple loops dependent on the other factor, the maximum score was used.

**Functional analyses of gene lists**. All gene ontology pathway analyses were performed with the R package gProfileR[121]. The top twenty GO biological processes terms with the lowest p-values were plotted.

**Nucleosome binding**. Nucleosome positions and factor occupancy was analyzed with NucleoATAC[132]. NucleoATAC was run with standard parameters on merged ATAC-seq bam files, separately for each genotype. SATB1 and CTCF binding sites were used as the input for the occupancy analysis. Peaks were centered and extended 250 bp upstream and downstream prior to the analysis.

**Transcriptional insulation scores**. We investigated whether the expression of genes inside SATB1 and CTCF loops was different from the neighboring genes outside the loops. To test this, we established an insulation score by calculating the absolute difference between the mean expression of genes inside loops [log10(counts / gene length + 0.01)] and the mean expression of genes found in the same size regions upstream and downstream from each loop. To compare the isolated values against a null distribution, loop coordinates were randomly shuffled across the genome and differences between the expression of genes residing inside the randomized loops with their neighbors were calculated as described above. If a shuffled loop was not containing any genes it was re-shuffled.

**Permutation analyses**. In order to construct a null statistical model (e.g. determination of common genes between two gene subsets, estimation of expected peak overlaps), permutation analyses were performed. In cases of overlaps between two files with genomic coordinates, the coordinates of one file were shuffled 1,000 times with the bedtools shuffle command[110]. The new overlap for each iteration was calculated. The mean overlap count was used to determine enrichment. Finally, a p-value was calculated as follows: p-value = (X times the permutation overlap was higher than the actual overlap/1000). The same was applied for overlaps between gene lists, with the exception of drawing out random genes instead of shuffling coordinates. Difference in mean values were assessed with a Mann Whitney U test (Wilcoxon rank sum test) except where otherwise stated.

To statistically evaluate accessibility of SATB1 peaks (Fig. 4a), each SATB1 HiChIP peak was centered and extended 250 bp upstream and downstream. An average read-depth normalized ATAC-seq score was calculated for each centered peak using the UCSC executable, bigWigSummary. Scores were log10 transformed. The process was repeated 100 times after random shuffling of the centered and extended peaks across the mm10 genome. The shuffleBed command from the bedtools suite[110] was used to shuffle the peaks. The average accessibility scores were calculated for the permuted and the "real" value distributions. A bootstrap p-value was calculated as: Number of randomly permuted mean values bigger than the "real" mean value / 100. In order to identify SATB1 peaks that occupied regions with reduced accessibility levels (Supplementary Fig. 5h), the aforementioned analysis was performed with 10 permutations. The average log10 transformed read-normalized accessibility scores of the ten random permutations was used as a cutoff for picking SATB1 peaks with low accessibility levels.

**Identification and analysis of T cell subset-specific genes**. RNA-seq raw counts of sorted DN1, DN2b, DN3, DN4, ISP, DP, CD4SP, CD8SP T cell subset populations (GSE109125[55]) were normalized using DESeq2[120]. DESeq2 likelihood ratio test was used to identify genes that transcriptionally differed between cell subsets (p < 0.05). Genes were clustered using a heatmap (Supplementary Fig. 4a, Supplementary Data 3) created by the pheatmap[133] R package using Ward's D-distance criterion for the formation of clusters. Eight clusters were isolated using the cutree R function using k = 8. Of those, three clusters were manually selected as DN, DP and SP signature genes (arrows in Supplementary Fig. 4a, Supplementary Data 3). Moreover, the DN (DN1, DN2b, DN3), DP and SP (CD4SP, CD8SP) signature genes were further filtered to be expressed at least 2-fold higher than in the other T cell subsets analyzed.

Expression changes of T cell signature genes were compared between WT and *Satb1* cKO cells and random permutations. Hypergeometric distribution tests were used to assess the statistical significance of set enrichments between T cell signature genes and different categories of genes (e.g. genes within SATB1 loop anchors, differentially expressed genes, etc.), using the R phyper function. Actual enrichments were calculated using the average value of the theoretical hypergeometric background distribution corresponding to each comparison. In scatter plots depicting the correlation between gene expression changes and changes in H3K27ac looping, the genes with |linking score| > 40 (overinteracting – underinteracting H3K27ac loops) were removed as influential outliers.

**3D computational modeling**. The WT and *Satb1* cKO thymocyte Hi-C contact matrices, binned at a resolution of 20 kbp, were pre-processed by vanilla coverage normalization[1]. An area delimited by two TADs (called by hicexplorer[111] at 150 kbp resolution and spanning chr16: 21420000–25340000) encompassing the *Bcl6* locus (chr16: 23965052–23988612) was used for 3D modeling of chromatin interactions, using TADbit Python library[134]. Each contact matrix was modeled as a coarse-grained "beads-on-a-string" polymer model at an equilibrium scale of 0.01 nm/bp. First, we identified the optimal parameters to establish Z-score thresholds for attractive (*upfreq*) and repulsive (*lowfreq*) restraints, maximum inter-locus distance (*maxdist*) and maximum model physical distance, below which two model loci could be considered in contact with each other. Out of 500 models in total, we selected 100 models with the lowest amount of violated contact restraints. Next, we generated an ensemble based on the selected 100 models. Parameters were selected using a grid search approach aimed to optimize the correlation between the experimental input and the ensemble, as a model-derived contact map. The selected parameters were for WT: upfreq −0.3, lowfreq −1.0, maxdist: 550, cutoff dist: 400.0 and for *Satb1* cKO: upfreq 0.0, lowfreq −2.0, maxdist: 1100, cutoff dist: 800.0. We used the optimized set of parameters and based on correlation between the model and the experimental contact maps, we selected the top 5,000 models (out of 25,000 models) to generate a production ensemble. Production ensembles were well correlated with the input Hi-C contact matrices (Spearman's r = 0.716 for WT and r = 0.7877 for *Satb1* cKO).

The production ensemble models were clustered using TADbit's built-in objective function and the MCL Markov clustering[135]. The centroid model of the first cluster for each genotype was visualized using the UCSF Chimera molecular visualization software[136] at a scale of 1:4. The models were colored according to the distance from the 5′-end. The highlighted beads represent an approximation of SATB1 loop anchors derived from SATB1 HiChIP data at *Bcl6* (yellow) and its super-enhancers (black). The WT models were rotated to obtain a clear view of all relevant structures and then aligned to the *Satb1* cKO-derived models using the match command.

The three-dimensional structures in the production ensemble were used to calculate the distance distribution between the beads

corresponding to the loci closest to the *Bcl6* locus and its proximal super-enhancer 1 (corresponding to chr16:23980000–24000000 and chr16:24240000–24260000, respectively). The resulting distributions of the WT and *Satb1* cKO data were compared using the Mann-Whitney U test and assessed for multimodality with the skinny-dip test[137]. The distances between *Bcl6* and its super-enhancer based on the ensemble of the 5,000 sampled models were: mean rank WT: 2616.3162; mean rank *Satb1* cKO: 7384.6838 ($p = 0.0$; two-sided Mann-Whitney U test). Dip test results (distribution modes): WT: [171.205, 200.087], [200.248, 298.606] ($p < 0.001$); *Satb1* cKO: [198.063, 441.868].

### Reporting summary

Further information on research design is available in the Nature Research Reporting Summary linked to this article.

## Data availability

The data that support this study are available from the corresponding author upon reasonable request. Raw and processed data for all genomics experiments are deposited in Gene Expression Omnibus database under accession number GSE173476. We re-analyzed publicly available Hi-C datasets from untreated and CTCF-depleted mESCs (GSE98671[6]) and WT and RAD21-depleted DP murine thymocytes (GSE48763[56]). We re-analyzed publicly available RNA-seq of sorted DN1, DN2b, DN3, DN4, ISP, DP, CD4SP, CD8SP immune cell populations (GSE109125[55]). Publicly available ChIP-seq datasets were: SATB1 ChIP-seq (GSM1617950[85]), (DRR061108[17]), CTCF ChIP-seq (ENCFF714WDP[109]), H3K4me3 ChIP-seq (ENCFF200ISF[131]), H3K4me1 ChIP-seq (ENCFF085AXD[131]), H3K27ac ChIP-seq in WT (GSM1504384 + 5 [https://www.ncbi.nlm.nih.gov/geo/query/acc.cgi?acc=GSM1504384]) and *Rad21* cKO (GSM1504386 + 7 [https://www.ncbi.nlm.nih.gov/geo/query/acc.cgi?acc=GSM1504386]) DP cells[67], thymus-specific enhancers (GSE29184[68]). Additionally, ChIP-Atlas[42] was used in the pilot analysis. Source data are provided with this paper.

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

## Acknowledgements

We would like to thank Elena Deligianni for the neonatal thymi image acquisition, Sevasti Papadogiorgaki and George Chalepakis for transmission electron microscopy and George Garinis for fruitful discussions. Some of the molecular graphics and analyses were performed with UCSF Chimera, developed by the Resource for Biocomputing, Visualization, and Informatics at the University of California, San Francisco, with support from NIH P41-GM103311. This work was supported by the European Union (European Social Fund ESF) and Greek national funds through the Operational Program 'Education and Lifelong Learning' of the National Strategic Reference Framework (NSRF) Research Funding Program ARISTEIA [MIRACLE 42], by FONDATION SANTE (X-COAT) and by Chromatin3D-H2020-MSCA-ITN (GA642934). D.P. was supported by the Polish National Science Centre (2019/35/O/ST6/02484 and 2020/37/B/NZ2/03757), Foundation for Polish Science co-financed by the European Union under the European Regional Development Fund (TEAM to DP), and by Warsaw University of Technology within the Excellence Initiative: Research University (IDUB) programme. The funders had no role in study design, data collection and analysis, decision to publish, or preparation of the manuscript.

## Author contributions

T.Z. and C.S. designed the study. T.Z. performed the genomics and immunofluorescence experiments. T.Z. and A.K. performed the computational analyses. S.F. consulted for library construction and performed sequencing experiments. P.T. and D.T. created the *Satb1* cKO mouse. T.Z. and D.T. performed animal, histology and flow cytometry experiments. D.A.P. performed the western blot, 3 C and MNase experiments. I.R.T. performed computational 3D modeling experiments. D.P. and C.N. consulted the computational analyses. G.P. performed FACS analysis and IF experiments. M.K. performed tissue section staining. T.Z. wrote the original manuscript. C.S. supervised the work, obtained funding and corrected the manuscript. All authors read, discussed and approved the manuscript.

## Competing interests

The authors declare no competing interests.
