## [Peer Review File · Nature Communications]

REVIEWER COMMENTS

Reviewer #1 (Remarks to the Author):

This paper describes a genome-wide chromatin organization formed by a cell-type restricted nuclear protein, SATB1. By employing Hi-C, HiChIP (for SATB1 and H3K27ac) and comparing wild-type and *Satb1^{fl/fl}* CD4-Cre⁺ mice, the authors found that SATB1 mediates chromatin looping connecting enhancers and promoters in thymocytes. Importantly, expression of some of the critical genes (e.g. *Bcl6*) for T cell lineages is affected by SATB1 ablation in *Satb1^{fl/fl}* CD4-Cre⁺, which is accompanied by loss of promoter-enhancer loops for these gene loci. The authors compared the results with CTCF and claim that SATB1 builds a more “refined layer of genome organization” upon a CTCF scaffold. This statement is primarily based on a major overlap (84% of 1374 loops) of SATB1-mediated loops with CTCF-mediated loop (3029 loops) (Fig. 2c). SATB1-mediated loops appear to be engulfed within CTCF loops. Unlike CTCF, which is a ubiquitous protein, it is significant that SATB1 anchors loops overlapping with genes enriched for immune-system categories in thymocytes.

The SATB1-mediated chromatin organization as described in this manuscript is very similar to YY1 which is also engaged in enhancer-promoter loop formation, except that YY1 is a ubiquitous protein. Because SATB1 is known to be a global regulator for gene expression, especially in T cells, and it is already known that expression of critical genes for T cell lineages is under the control of SATB1. An important question addressed is whether and how SATB1 establishes specific 3D chromatin organization to endorse such gene expression. This remain ambiguous for YY1 loops, as the loss of looping was associated with either up- or down-regulation of genes associated with YY1 loops. Thus, changes in gene expression could be due to an indirect effect of looping or looping itself has a repressive effect. This remains the case for the effect of SATB1-mediated looping on gene expression as well. Furthermore, there are many genes associated with SATB1 loops, displaying no change in expression. In fact, all possibilities (up-, down and no change in expression) are found for associated genes when SATB1 loops were either decreased or even increased upon SATB1 deletion. Therefore, unfortunately, these results do not necessarily advance our understanding about the direct causal effect of chromatin looping on gene expression.

Having said this, chromatin looping by the cell-type specific SATB1 protein is interesting. A thorough analysis of genome-wide SATB1-mediated chromatin landscape would be meaningful for future chromatin research. Also, because the authors give a strong impression by stating that in line 245-246 “CTCF participates in mechanisms responsible for supporting a basal high-order T cell chromatin structure, whereupon SATB1 likely exerts its action in a more refined organization layers consisting of promoter-enhancer chromatin loops”, a burning question comes up whether SATB1-mediated structure depends on CTCF or not. This hypothesis has not been directly addressed. Some insight with regards to the functional relationship between the two factors would be nice if provided, besides a large overlap in looping.

There are multiple points that need to be addressed as below.

1. The results from their characterization of SATB1 loops are not described sufficiently in the main text. To support the main text, Supplementary File 6 need to be revised, adding more sheets with specific information.

Describe loop distribution (H3K27ac HiChIP) in WT and cKO (number of loops for promoter-enhancer, promoter- others, enhancer-others, promoter-promoter, enhancer-enhancer, others-others). Provide number of SATB1 anchored loops for each type of looping in the

parenthesis.

Describe loop information from SATB1 HiChIP, and compare with H3K27ac HiChIP to show overlapped and non-overlapped loops and for the latter loops, describe characteristics.

Any overlap between cKO and WT?

List of genes associated with loops (indicate criteria for assignment of genes that are associated with loops and their expression changes in cKO). Especially, 922 down-regulated and 719 up-regulated genes have been identified in cKO. What types of looping do these gene loci exhibit?

Relevant to these questions, Figure 3e is unclearly explained.

2. Figure 1a shows a list of genes frequently found at H3K27ac-mediated loop anchors, and RAG1/2 proteins are identified. In light of RAG1/2 function in the recombination of B and T cell receptor loci, one would expect that RAG1/2 associate preferentially with receptor loci. Is this the case or their association is genome-wide? A potential explanation is needed if the association is genome-wide.

3. It is not convincing why SATB1-mediated looping overlap significantly (84%) with CTCF-mediated looping and yet, the authors find SATB1 binding primarily to nucleosome regions while CTCF is not (Fig. 2E). How genes were assigned to be linked to loop anchors? Mention criteria employed in the main text. Also, Fig. 2E could be removed as so many SATB1 peaks coincide with CTCF peaks. Relevant to this point, the definition of SATB1 binding specificity as stated in Line 88 (based on ref 26) should be restated based on original earlier papers for its true binding specificity.

4. A more detail description is required about the long SATB1 isoform in this manuscript. This is necessary because the paper referred is not published. Also, the efficiency of pull-down and target specificity by the custom antibody must be shown. Also describe any difference in ChIP-seq data generated by this new antibody versus ChIP-seq generated by other antibody or antibodies (references 81 and 73).

5. Based on several publications, SATB1 recruits histone modifying enzymes and change histone modification status. However, in this paper, no major change in H3K27ac marks was detected (line 295). This is surprising. For T cell development, Kitagawa et al paper using the same cKO mice showed major difference in H3K27ac marks at key enhancers. How to explain this? In this context, in Line 192, switch from A to B compartment is mentioned. Even though gene ontology pathway did not show anything, at the individual gene level, this point needs to be addressed (histone modification, relevant genes, and their expression).

6. Line 187, chromatin compaction in cKO is based on HP1alpha, but this information alone may not be sufficient, as it could reflect increased in its expression. Some other method needs to be added to verify chromatin compaction, if indeed chromatin compaction occurs in cKO.

7. Quality of 3D interaction in Figure 4D (HiC) is not fully convincing. Arguing with one image that contains one small yellow box is not sufficient. This is an important point making the case for Bcl6, and therefore, show a larger scale including SE2 for HiC. Also, this reviewer suggests to show similar HiC image for H3K27ac HiChIP and SATB1 HiChIP (if Fig. 4D is simple HiC and not H3K27acHiChIP--unclear).

8. There are multiple places where references are missing, or not placed earlier in the main manuscript. This could result in not providing an accurate background information.

1) Line 72-72) Modify the sentence slightly to accommodate Kitagawa et al., paper and Cruz-Molina, Cell Stem Cell, 2017 paper. Both represent data supporting that chromatin structure precedes (or prime) correct future transcription.

2) Line 125-140). References 73, 81 and 85, (especially 81 and 85) must be mentioned much earlier in the right context with clear explanation (preferentially in Introduction). Line 84 under "during T cell development (20), can accommodate these. Some explanation is required and this could continue in the result section (e.g. Line 127).

3) SATB1 being both activator or repressor has been published long back (for instance, Han

et al., 2006, nature).

4) Although it is not a genome-wide study, for the cytokine cluster 200kb region, SATB1 has been published to form chromatin loop domains including regulatory regions. Induction of cytokine genes and looping were tightly linked. This earlier paper (ref 45) needs to be explained in the main text as it is highly relevant background paper to this current manuscript.

5) Several critical T cell developmental genes have been published to be the targets of SATB1 (81, 85) and not all of them are found deregulated here. It is strange why Foxp3 was not found in this present study at all. Maf being SATB1 dependent is described in ref 45, as its induction requires SATB1.

6) Since SATB1 is very similar to YY1 in linking enhancer to promoter, it is better to introduce YY1 in Introduction.

7) Line 119 onward). It sounds like the production of Satb1^{fl/fl} CD4-Cre⁺ were unique in this paper, but at least two other papers have already described these mice. Refer to them and highlight what is new.

8) Line 138) Again, earlier work must be mentioned about the T cell phenotypes upon SATB1 ablation, and then highlight new results found in this paper (or say, consistent with...).

Additional points

1) Line 148) State what is meant by 77% "unchanged" TADs. Unchanged in what aspects?

2) Line 150) State resolution of Hi-C here after "broad scale differences"

3) Line 162-163) The explanation is unclear in the main text. The corresponding Figure legend was clearer.

4) Line 166) Regarding Extended Data Fig. 1e, what is the significance of this analysis?

6) Line 82-83) Change sentence to "We have identified SATB1 to be enriched at gene promoters ----".

Reviewer #2 (Remarks to the Author):

This manuscript presents a detailed analysis of the genomic topology features that depend on the transcription factor SATB1 in developing T cells, and a survey of their impacts on gene expression in the cells. As SATB1 is important in multiple unrelated tissues, the authors begin by establishing a highly specific T-lineage-restricted conditional knockout SATB1 mouse line in which the Satb1 gene is deleted selectively at the beginning of the DP stage in the thymus, which is a common source of all mature alpha beta lineage T cells. This mouse proves to replicate the autoimmunity phenotype of previously described germline Satb1-knockouts, confirming the functional abnormality of the Satb1-deficient T cells in vivo. The rest of the paper focuses on identifying the origins of these abnormalities by study of total thymocytes (>80% DP cells) from these Satb1 conditional knockout mice. The analysis combines Hi-C, SATB1 Hi-ChIP (compared with CTCF Hi-ChIP), and RNA-seq analyses of these thymocytes with previous data sources identifying "predicted thymic enhancers" and ChIP-seq analyses of H3K27ac, H3K4me1 and H3K4me3, similarly in unfractionated thymocytes, but from wildtype cells.

The results show that SATB1 protein is needed for particular local loops to form, especially affecting genes involved in T cell function (a marked difference from the loops requiring CTCF), and that when these loops connect predicted enhancers with promoters, the corresponding genes are often altered in expression, usually although not always losing expression in the absence of SATB1. Of particular interest, the loss of SATB1 in DP stage thymocytes has particularly strong effects on TCRA gene rearrangement success. First, it causes a marked depression of TCRA-V segment transcription and bisects the Traj region

into a segment that shows increased accessibility and transcription (encoding distal J segments) and a segment that shows decreased accessibility and transcription (proximal J segments). Second, it also inhibits Rag1 expression substantially. Finally, the authors show that

There is a great deal of value in this paper, and the statistical and modeling analyses particularly are described with unusual care and lucidity. The results are intrinsically very interesting. However, the effects of the SATB1 loss on local formation of loops seem surprisingly weak, and the KO also shows only mild effects on ATAC accessibility. It is not completely clear, then, whether the authors conclude that their meticulous looping analysis has revealed a distinctive role of SATB1 different from other sequence-specific transcription factors. In addition, there are several questions about how basic aspects of the study were characterized that the authors should address, and some additional questions about the order of the presentation.

1. While the pathology of the *Satb1* cKO mice is clear evidence that the KO makes a difference, the actual thymocytes used for the analysis are not well defined. The problem is that most, but not all of the cells in the thymus are DP cells, but there is a considerable body of evidence (ranging from Scollay, 1990's to Hogquist, recently) that the average dwell time of normal thymocytes in the DP stage is no more than 3-5 days at most. With the CD4-Cre knockout only deleting *Satb1* in this stage, it is possible that some cells in the majority population of the thymus at a given time have lost *Satb1* too recently to deplete the protein from the chromatin. This would yield an underestimate of the importance of SATB1 for loop formation.

(a) How well is the SATB1 protein depleted in actual SATB1 cKO thymocyte samples? Any evidence, e.g. from Western blots or intracellular protein staining of these thymocyte populations, would be very helpful. If feasible, intracellular staining could also be helpful to indicate whether the depletion is uniform across the population, also relevant as in the next comment.

(b) The genes that are downregulated in *Satb1* cKO cells compared to WT include genes that normally increase sharply from DP to SP (*Ccr7*, *Cd40lg*) as well as genes that decrease sharply from DP to SP (*Rag1*, *Bcl6*). Are these gene expression changes happening in the same cells? Conceivably subsets of DP cells could be separated based on these other surface molecules, or based on cell-surface TCRab expression, CD5, and/or CD69 staining, sufficiently to use qRT-PCR to check which cells express these genes. Extended data table 6, which is extremely informative, suggests that the two late-expressed genes might be affected indirectly as a result of a general block of differentiation, in which case they could be downregulated without direct promoter-enhancer loop interactions with SATB1 itself.

2. In this paper, the authors use a novel home-made antibody against the long isoform of SATB1. However, the report of the characteristics of this antibody has not been published, and it is not clear whether all the thymocyte populations or only a minority express this long isoform. If the characterization paper still is not available, then some details about the properties of the new antibody should be described. It is especially important to know whether this antibody is recognizing the same SATB1 sites that are reported using other reagents, and how well its specificity for SATB1 is established.

3. It is important to explain how many independently derived biological replicates were involved in all the new data, Hi-C, Hi-ChIP, ATAC-seq, and RNA-seq. Some of the effect sizes are small enough (e.g. Ext. Fig. 1e,g; Fig. 3c) so that their reproducibility is a strong concern.

4. The organization of the paper is a little strange, making it hard for the reader to follow some aspects of the story.

(a) A reader would expect to see the full information about the organ that is being studied in the cKO, namely the thymus, presented all together at the beginning, rather than having cell percentages presented in Fig. 1 and Fig. S1 but cell numbers and viabilities reserved for Fig. 6.

(b) Similarly, an obvious early question to ask about removing a transcription factor is what genes are transcriptionally affected. It is surprising that this whole subject is deferred till Figs. 5 and 6, and Ext. Fig. 4, leading to questions about whether the chromatin-level changes that are seen have any impact on function – and whether they are specifically occurring at loci that change expression.

(c) In general, the reader wishes that the authors would make a more direct statement about how important they judge the SATB1-mediated looping to be in mediating gene expression control in these thymocytes. Are they being reticent because the effects on ATAC-seq and even RNA-seq tracks shown in Ext. Fig. 4 are modest? It would be valuable in any case to make a clear statement of the relationship that is seen.

5. In considering the results in Extended Data 6, it seems worth considering some additional criteria that might sharpen the relationship between SATB1-mediated enhancer-promoter looping and gene regulation. First, is it possible, that *Tcra* and *Bcl6* show a stronger effect than the other targets precisely because they are the only genes in this set that turn on for the first time in the DP stage, when the SATB1 deletion is being induced? *Tcf7*, *Lef1*, *Ikzf1* and even *Cd8a* would all normally be activated before the CD4-Cre could be strongly enough expressed to delete *Satb1*, and this could contribute to a high background of expression. Second, as noted above, some of the genes that are affected without showing clear evidence of SATB1-dependent loops (overlinked – underlinked >1) normally are not expressed at all during the DP stage, when most *Satb1* cKO thymocytes are arrested. Would removal of these maturation-dependent genes yield a higher correlation between looping and transcriptional responsiveness? Thus, it might be useful to employ ImmGen RNA-seq data to filter the target genes on their normal developmental patterns of expression and then check for correlation with SATB1-dependent looping.

6. Minor: Extended Fig. 3 shows results of a very interesting model, but the figure is extremely difficult to see. First, the legend does not do a good enough job to explain panel (a). Presumably this is showing striking compaction of the chromosome predicted in the presence of SATB1, but one needs to be able to visualize the DNA strand better against the dark background even when it is printed out. How realistic is it to assume that the *Bcl6* locus in the absence of SATB1 would exist in such a stretched-out, linearly extended form? Second, panel (b) must be enlarged substantially, as it is really hard to see the viridis color coded regions around the *Bcl6* promoter and SE1 enhancer. (I could only see them by zooming in on the pdf on the monitor; the printout was not at all clear.) It could be worthwhile to add a set of zoom-in panels for the lower left sections of each of these panels in Extended Fig. 3b, so that these key features of the model predictions can be appreciated unambiguously.

Reviewer #3 (Remarks to the Author):

In the current study, Zelenka et. al. investigated the role of genome organizer *Satb1* in thymocytes and how the 'loopscape' mediated by *Satb1* affects the developing T cells. The authors delve into the global chromatin architecture by using various assays including HiC, HiChIP and ATAC-seq and correlated with the transcriptomic modulations upon ablation of *Satb1* in CD4+ T cells. Almost at every level of analysis, they have drawn parallels with the organizer CTCF to demonstrate *Satb1* mediated changes. A positive aspect of the work described in the manuscript is the authors have presented myriad of analyses to demonstrate how *Satb1* regulates the nuances in thymic gene regulation (e.g. using the

Bcl6 and Tcr loci) primarily through differential enhancer-promoter contacts. The authors have also extended previous autoimmune related findings to other organs such as pancreas. The methods are written well and extensive. However, the conclusions drawn from this data are not always consistent within the present analysis and also do not correlate with what has been shown by others. The primary conclusion is that the SATB1 mediates promoter-enhancer loops affecting master regulator genes, collectively being critical for cell lineage specification and immune system homeostasis. This key point is related to a first weakness of the manuscript, which is that the authors have chosen to perform this study using one specific isoform of SATB1 that is not the abundant form. The rationale for using this specific isoform is not clear. The functional importance of this isoform is mentioned in the discussion section but the reference cited for the same is a submitted manuscript hence it can't be cross-checked. Secondly, the cKO model used eliminates all isoforms of Satb1 hence the functional data can't be correlated with the assays such as HiChIP using antibody against the long isoform. Although the data shown provides additional information on the role of Satb1 as a genome organizer, the manuscript suffers from major pitfalls and requires substantial revisions to unequivocally support their conclusions.

Comments

1. What is Satb1 'loopscape'?

2. Line 115. The authors mention "We drew our attention to SATB1, which displayed significant enrichment at the H3K27ac 115 loop anchors (Fig. 1b) and also represents a known genome organizer." The authors have excluded key publications which have looked at SATB1 binding alongside the H3K27ac mark (eg Kitagawa 2017). The authors should introduce these findings in order to build the premise for their work.

3. The authors have used pre-analyzed datasets from ChIP-Atlas for generating Figure 1a. The conditions under which the ChIP seq for the factors in question are not stated anywhere. The authors should perform their own ChIP seq (at least for SATB1 is much essential) replicating the conditions which were used to make H3K27ac HiChIP samples.

4. While expounding the autoimmune phenotype of Satb1 cKO mice, the authors again fail to cite the original and subsequent findings which showed that SATB1 deletion in CD4 leads to autoimmunity. The authors should include and discuss their findings in the text to support their observations. Furthermore, the authors mention in line 125 that there is a reduction of SP cells in the thymus.

5. Although in Figure 1c, there is no percentage indicated. Just by looking at Fig 1c, it seems that there is an increase in CD4 and CD8SP upon cKO of Satb1. This should be clarified and gatings should be carefully checked. Additionally, showing levels of intracellular Satb1 in these validation figures upon gating is essential.

6. The authors have used a conditional knockout (cKO) model of Satb1 for all studies reported here. However, the phenotypes described here do not correlate with the other published models of Satb1 cKO. Furthermore, the phenotype shown in Extended figure 1a., the phenotype of CD4cre+Satb1fl mice shown has not been discussed in any of the earlier studies (eg. Kakugawa 2017). The authors should provide the data regarding the KO regions (fl and cre), the genotyping figures and validations for a successful KO.

7. Lines 130-131: The authors state "... prevailing IL-17 cell responses and increased levels of pro-inflammatory cytokines such as IFN γ and TNF α detected in Satb1 cKO sera..." Yet again, there is no reference to the article Yasuda, 2019 which showed Th17 mediated autoimmunity in the absence of Satb1. The authors should thoroughly revise the text to incorporate all the important information.

8. The authors show in Fig 1e 'disturbance of the islets of Langerhans' upon Satb1 KO. Authors should show more sections and point the differences in WT vs cKO of Satb1 for better clarity. Furthermore, there is no discussion/inference from the glucose tolerance test

as to why only one time point differ which also seems to recover later?

9. The authors indicate an increase in autoantibodies upon Satb1 cKO (Fig 1F). The imaging is poorly done and is unclear. This is also a crude and indirect method. Instead, the authors should use more robust and quantitative methods such as ELISA or LIA.

10. In line 146, the authors state "We did not identify any major changes in high-order chromatin organization...." In the related figure 2a, the authors should describe the parameters for this analysis, and multiple regions should be shown to arrive at such conclusion. Same for the TAD analysis in the supplementary information. The authors should also compare this data with HiC analysis for the same regions under CTCF null condition (the data is publicly available).

11. Importantly, the authors used a custom antibody specific for only SATB1 long isoform for their HiChIP experiments. Since this is the central focus of their findings, it necessitates the validation of the efficacy of the antibody. As almost 90% of Satb1 expressed in thymocytes is not the long isoform, it warrants that either the HiChIP experiments to be repeated with Satb1 antibody or show that the effect is specific only to the Satb1 long form.

12. How was the strength of the interaction pair calculated (fig2b)? This should be explained more in the text to support the conclusion of the figure. It is important that the authors plot for higher resolution contacts to better conclude the reduced contact strength of Satb1 loops.

13. In lines 168-170, the authors state "...genes residing in both CTCF- and SATB1-mediated loops were transcriptionally insulated from their gene neighbors (Extended Data Fig. 1f) and this characteristic was not altered in the Satb1 cKO. ..." Does this mean if the contacts by Satb1 are lost (as described by the APA analysis), there is no change in these genes' expression?

14. Figure 2e, The authors have used 'Satb1 binding site' for their nucleosome binding analysis from ATAC-seq data. This becomes dicey as Satb1 binding sequence is not perfect but dynamic. Hence, Satb1 ChIP-seq peaks are necessarily to be used instead of a singular input sequence, which is applicable to Ctf only, here. The authors should also indicate the motif for Satb1 that they have used for their analyses.

15. Figure 3a. The authors show immunofluorescence of HP1 and PolII to demonstrate 'more chromatin compactness' as stated in the text. Firstly, the images are of very poor resolution with a lot of background noise. More importantly, relative increase in HP1a does not necessarily demonstrate heterochromatinization, since PolII remains the same. The authors need to show nuclear stainings alongside these stains. The authors should also perform quantitative Western blotting to ascertain these observations.

16. The immunostaining the authors have used total thymocytes from CD4+ cKO mouse. Therefore, it is unclear whether the cells shown/quantitated are of which thymic subset. A straightforward way would be to use sorted thymocytes.

17. Line 195: The authors wrote "This observation further reinforced our hypothesis that SATB1 acts at a finer-scale level of genome organization' the conclusions need to be re-written according to the results in a specific manner.

18. Lines 213-214: The authors mention "especially evident at the transcription start site of genes (TSS), suggesting a direct role of SATB1 in gene transcription regulation." Which genes are the used for overlapping ATAC peaks; all or up or down upon Satb1 cKO? The accessibility (shown in fig 3 b and 1k) should be validated at the transcriptome level via qRT-PCRs to demonstrate the repressive role of Satb1 for the genes as mentioned in line 220.

19. Figure 4a: -The authors show a correlation of RNA seq to over-vs-under-interacting K27 loops in the KO condition. Given its low correlation (although positive), it seems that there is only a subset of genes expression that correspond to change in the looping. The authors should discuss this. Further, it would be more informative to plot upgenes vs over-interacting and downgenes vs under-interacting.

20. For Figure 4b and 4c, the authors show expression change for genes at the loops of Satb1, CTCF and the rest of the under-interacting and over-interacting loops. For the

conclusion (lines 261-263) "...Moreover, the expression of genes located at anchors of SATB1-mediated loops was decreased more dramatically than genes located at CTCF loopssupporting direct involvement of SATB1 in the regulatory chromatin loops", it is essentially required to include CTCF KO thymic RNAseq data or CTCF KO HiChIP data of K27ac to infer this correctly.

21. In Figure 4F, what 'disturbance' are the authors stating here? The differences should be labeled and clearly shown. Cell quantitation from germinal centers from both WT and cKO mice is needed to support the data. Same statements were also used for Fig 3F. The authors need to revise their results and conclusions in the text thoroughly.

22. For Fig 4e, It is imperative to validate some of these contacts (like Bcl6) via 3C-qPCRs on WT vs SATB1 cKO conditions. BCL6 expression should be shown at the protein level.

23. Figure 6D. The panels shown for WT and cKO do not show any difference to support the conclusion "we also identified the disrupted thymic structure and impaired cell-to-cell communication".

24. In the Satb1 cKO thymus, the H&E staining shows no difference. Methylene Blue staining only shows the relatively low number of cells and the TEM images are set to different contrasts, hence are not reliable measure of differences. The authors need to include higher quality and properly labeled pictographs which show the claimed difference unambiguously.

25. Lines 378-382: The authors state "The presence of two SATB1 protein isoforms was recently described by our group..... We showed that the long SATB1 protein isoform had a higher propensity to undergo phase transitions compared to the short isoform". What is the significance here? It is not appropriate to cite papers that are submitted (not to any preprint server) and the data can't be verified.

Response to Reviewers' Comments

Nature Communications manuscript NCOMMS-21-25391-T

"The 3D enhancer network of the developing T cell genome is controlled by SATB1"

We have now completed the revision process of our manuscript and we submit the revised version for your consideration. We would like to thank the Reviewers for their time in reviewing our manuscript but most importantly for their constructive comments that helped us improve our work. We hope that our responses, based on the experiments we have now completed, will adequately address the concerns raised.

Below please find a point-by-point response to the Reviewers' comments. (Original comments are in **black** and responses in **blue**).

Reviewer #1

This paper describes a genome-wide chromatin organization formed by a cell-type restricted nuclear protein, SATB1. By employing Hi-C, HiChIP (for SATB1 and H3K27ac) and comparing wild-type and *Satb1^{fl/fl}* CD4-Cre⁺ mice, the authors found that SATB1 mediates chromatin looping connecting enhancers and promoters in thymocytes. Importantly, expression of some of the critical genes (e.g. *Bcl6*) for T cell lineages is affected by SATB1 ablation in *Satb1^{fl/fl}* CD4-Cre⁺, which is accompanied by loss of promoter-enhancer loops for these gene loci. The authors compared the results with CTCF and claim that SATB1 builds a more "refined layer of genome organization" upon a CTCF scaffold. This statement is primarily based on a major overlap (84% of 1374 loops) of SATB1-mediated loops with CTCF-mediated loops (3029 loops) (Fig. 2c). SATB1-mediated loops appear to be engulfed within CTCF loops. Unlike CTCF, which is a ubiquitous protein, it is significant that SATB1 anchors loops overlapping with genes enriched for immune-system categories in thymocytes.

The SATB1-mediated chromatin organization as described in this manuscript is very similar to YY1 which is also engaged in enhancer-promoter loop formation, except that YY1 is a ubiquitous protein. Because SATB1 is known to be a global regulator for gene expression, especially in T cells, and it is already known that expression of critical genes for T cell lineages is under the control of SATB1. An important question addressed is whether and how SATB1 establishes specific 3D chromatin organization to endorse such gene expression. This remain ambiguous for YY1 loops, as the loss of looping was associated with either up- or down-regulation of genes associated with YY1 loops. Thus, changes in gene expression could be due to an indirect effect of looping or looping itself has a repressive effect. This remains the case for the effect of SATB1-mediated looping on gene expression as well. Furthermore, there are many genes associated with SATB1 loops, displaying no change in expression. In fact, all possibilities (up-, down and no change in expression) are found for associated genes when SATB1 loops were either decreased or even increased upon SATB1 deletion. Therefore,

unfortunately, these results do not necessarily advance our understanding about the direct causal effect of chromatin looping on gene expression.

Having said this, chromatin looping by the cell-type specific SATB1 protein is interesting. A thorough analysis of genome-wide SATB1-mediated chromatin landscape would be meaningful for future chromatin research. Also, because the authors give a strong impression by stating that in line 245246 “CTCF participates in mechanisms responsible for supporting a basal high-order T cell chromatin structure, whereupon SATB1 likely exerts its action in a more refined organization layers consisting of promoter-enhancer chromatin loops”, a burning question comes up whether SATB1-mediated structure depends on CTCF or not. This hypothesis has not been directly addressed. Some insight with regards to the functional relationship between the two factors would be nice if provided, besides a large overlap in looping.

There are multiple points that need to be addressed as below.

1. The results from their characterization of SATB1 loops are not described sufficiently in the main text. To support the main text, Supplementary File 6 need to be revised, adding more sheets with specific information.

Describe loop distribution (H3K27ac HiChIP) in WT and cKO (number of loops for promoter-enhancer, promoter- others, enhancer-others, promoter-promoter, enhancer-enhancer, others-others). Provide number of SATB1 anchored loops for each type of looping in the parenthesis. Describe loop information from SATB1 HiChIP, and compare with H3K27ac HiChIP to show overlapped and non-overlapped loops and for the latter loops, describe characteristics.

Any overlap between cKO and WT? List of genes associated with loops (indicate criteria for assignment of genes that are associated with loops and their expression changes in cKO). Especially, 922 down-regulated and 719 up-regulated genes have been identified in cKO. What types of looping do these gene loci exhibit?

Relevant to these questions, Figure 3e is unclearly explained.

We thank the Reviewer for the comment. Please note, that in most cases, the H3K27ac loops could be detected in both WT and *Satb1* cKO at the same genomic positions. Thus, we would be unable to show the comparison of WT/cKO cells that is proposed. As a result, the simple comparison between proportions of loops overlapping a feature would not change between the genotypes. See for example the two figures below indicating a similar enrichment for WT SATB1-mediated loops overlapping with WT H3K27ac loops and *Satb1* cKO H3K27ac loops. As such, we would like not to include these figures in the revised version of the manuscript. We nonetheless provide a complete characterization of all HiChIP loops, their mutual overlaps, and their overlaps with differentially expressed genes as a supplementary excel file (**Supplementary file 5**).

Despite no major changes in the presence/absence of loops, the looping interaction strength is very different between the genotypes. That is why in the original version of this manuscript we applied a differential looping analysis by FitHiChIP to identify these differential H3K27ac loops. We then subtracted the under- and over-interacting H3K27ac loops to emphasize the positive correlation between changes in H3K27ac interaction frequency and gene expression (**revised Figure 4h**). In the revised version of the manuscript, we extended and improved this analysis as a response to Reviewers' comments. More specifically, we identified sets of signature genes for DN, DP and SP T cells and plotted the correlation plots for each set of signature genes. Please see also materials and methods for more details on how these sets have been characterized. This analysis revealed that the regulatory SATB1 loops are highly specific to DP T cells. Additionally, a correlation plot for differentially expressed genes evinced a strong positive correlation (Spearman's $\rho = 0.62$) between the looping changes and transcriptional deregulation of *Satb1* cKO animals. An indication of causality for this effect is demonstrated in the **revised Figure 4f**, where we specifically compared the differential *Satb1* cKO under- and over-interacting H3K27ac loops that are or are not mediated by SATB1/CTCF, and the expression changes of genes affected by these loops.

These analyses clearly indicate that genes at the anchors of SATB1-mediated H3K27ac loops that were diminished in *Satb1* cKO, were significantly underexpressed, unlike the genes in either the CTCF-mediated loops or any factor-mediated overinteracting H3K27ac loops. Additionally, more unbiased analyses have been provided, indicating enrichment of enhancers at anchors of SATB1-mediated loops (**revised Figure 4d**). Note that in this analysis, we used the thymocyte-specific set of enhancers retrieved from ENCODE <http://chromosome.sdsc.edu/mouse/download/tissue-specific-enhancers.zip> from <https://doi.org/10.1038/nature11243>. Even though this was specified in the methods, it was not highlighted in the main text and in the figure legends. We emphasized it in the revised version.

CTCF-mediated loops displayed similar enrichment to SATB1-mediated loops when all thymic enhancers were used (data not shown). However, the thymus-specific enhancers that were highly enriched in SATB1 loops were not as enriched in CTCF loops, supporting the tissue-specific regulatory function of SATB1. This set of enhancers is based upon several ChIP-seq datasets (H3K4me1, H3K4me3, p300) and it was identified by a trained Hidden Markov model. Unfortunately, we do not have such a dataset available for *Satb1* cKO cells. A good approximation of regulatory chromatin loops is provided by the H3K27ac HiChIP, which we performed both in WT and *Satb1* cKO cells. Nevertheless, this chromatin mark decorates both active promoters and enhancers and thus H3K27ac peaks should not be synonymized with enhancers. Even in the aforementioned ENCODE study they mention: "Consistent with this finding, we found H3K27ac at only a portion (between 15 and 40%) of enhancers identified in this work (Supplementary Fig. 13c)", suggesting that one should not fully rely on H3K27ac when investigating enhancers. Having said that, in all our experiments and in the newly generated characterization of HiChIP loops, we utilized the WT thymus-specific set of enhancers. Thus, the comparison between WT and *Satb1* cKO reflects the deregulation of WT enhancer loops rather than describing the novel regulatory network present in the knockout, simply because this is the main question posed and because we do not have datasets to accurately infer the knockout enhancers.

Figure 3e is **Supplementary Figure 7a** in the revised version. We have now provided additional information in the figure legend to explain it in a better way.

2. Figure 1a shows a list of genes frequently found at H3K27ac-mediated loop anchors, and RAG1/2 proteins are identified. In light of RAG1/2 function in the recombination of B and T cell receptor loci, one would expect that RAG1/2 associate preferentially with receptor loci. Is this the case or their association is genome-wide? A potential explanation is needed if the association is genome-wide.

Discussing the genome-wide distribution of RAG1/2 proteins is beyond the scope of this paper. However, it is well established in the literature and we properly cited the respective *Cell* paper, investigating the association between RAG1/2 and active genes and enhancers genome-wide (<https://doi.org/10.1016/j.cell.2015.07.009>).

3. It is not convincing why SATB1-mediated looping overlap significantly (84%) with CTCF-mediated looping and yet, the authors find SATB1 binding primarily to nucleosome regions while CTCF is not (Fig. 2E). How genes were assigned to be linked to loop anchors? Mention criteria employed in the main text. Also, Fig. 2E could be removed as so many SATB1 peaks coincide with CTCF peaks. Relevant to this point, the definition of SATB1 binding specificity as stated in Line 88 (based on ref 26) should be restated based on original earlier papers for its true binding specificity.

A critical point that we should have made clear is that overlaps refer to loops and not binding sites. The fact that SATB1-mediated loops are frequently found embedded inside CTCF loops (as demonstrated in the original Fig. 2d) is quite logical given our conclusion that CTCF establishes the high-order chromatin structure whereas SATB1 mediates the finer-scale promoter-enhancer looping. Nevertheless, the binding preferences differ between the two proteins and this is highlighted by the differences in nucleosome binding. The discussion on SATB1 binding specificity was based on recent findings utilizing state-of-the-art approaches, while it is also becoming evident that TAATA (the original SATB1 binding motif) is not the exclusive SATB1 binding preference. Yet, we were happy to properly cite the references regarding the TAATA SATB1 binding motif in line 72: “Originally described as a Special AT-rich Binding protein²⁵, it is known for its binding of TAATA DNA motif^{26–28} and generally regions with more negative torsional stress²⁶.”

4. A more detail description is required about the long SATB1 isoform in this manuscript. This is necessary because the paper referred is not published. Also, the efficiency of pull-down and target specificity by the custom antibody must be shown. Also describe any difference in ChIP-seq data generated by this new antibody versus ChIP-seq generated by other antibody or antibodies (references 81 and 73).

We have corrected the reference to the manuscript describing the long SATB1 isoform which is now deposited in bioRxiv (*T. Zelenka, P. Tzerpos, G. Panagopoulos, K-C. Tsolis, D-A. Papamatheakis, V.M. Papadakis, D. Stanek, C. Spilianakis, SATB1 undergoes isoform-specific phase transitions in T cells*, doi: <https://doi.org/10.1101/2021.08.11.455932>). In the *bioRxiv* manuscript (**Figures 1e-f, Figure 2a, Figure 3e, Supplementary Figure 1a,b, Supplementary Figure 3**) we described the specificity of the long SATB1 isoform-specific antibody and we also provided pixel-based colocalization analysis of super-resolution microscopy experiments based on the long SATB1 isoform-specific antibody and general SATB1 antibodies non-selectively targeting all SATB1 protein isoforms. Western blot experiments comparing the two antibodies have also been provided.

Additionally, we have included into our revised manuscript a comparison of the SATB1 long isoform-specific binding sites (identified in this manuscript) with two publicly available SATB1 ChIP-seq experiments (*Hao et al., 2015; Kitagawa et al., 2017*). Based on the Relative Distance metric of Bedtools (see **Supplementary Notes**), the long SATB1 isoform-specific binding sites were non-randomly spatially correlated with both publicly available SATB1 ChIP-seq datasets (which utilized antibodies targeting all SATB1 isoforms; **revised Supplementary Figure 4d**). Moreover, the genomic feature association analysis (Genome Ontology) for the three datasets of SATB1 binding sites, analyzed by the AnnotatePeak function of Homer showed highly comparable binding preference for all SATB1 antibodies (**revised Supplementary Figure 4e**).

5. Based on several publications, SATB1 recruits histone modifying enzymes and change histone modification status. However, in this paper, no major change in H3K27ac marks was detected (line

295). This is surprising. For T cell development, Kitagawa et al paper using the same cKO mice showed major difference in H3K27ac marks at key enhancers. How to explain this? In this context, in Line 192, switch from A to B compartment is mentioned. Even though gene ontology pathway did not show anything, at the individual gene level, this point needs to be addressed (histone modification, relevant genes, and their expression).

In our manuscript, we claim that the major changes in *Satb1* cKO are related to H3K27ac looping (i.e. 3D chromatin arrangements) and not to 1D H3K27ac signal. In the *Kitagawa et al., 2017*, the authors did not identify a major difference in H3K27ac ChIP-seq signal in *Satb1* cKO cells, which is consistent to our data. First, we should emphasize that the authors of *Kitagawa et al.* paper were mostly focused on the roles of SATB1 in the development of regulatory T cells, unlike our manuscript. Regarding the association with H3K27ac we would like to point to the following quote by *Kitagawa et al.*, as well as to the accompanying Figures:

“Wild-type and *Satb1*-deficient DP cells and immature CD4SP cells showed similar intensities of H3K4me1, whereas *Satb1*-deficient *tTreg* precursor cells showed much lower H3K27ac intensity than their wild-type counterparts (Fig. 8a,b and Supplementary Fig. 7d). The reduction occurred ‘preferentially’ at Treg-SE regions to which *Satb1* bound before chromatin opening (Fig. 8c and Supplementary Fig. 7e,f). For example, activation of Treg-SE at the *Foxp3* locus did not increase beyond the level observed at the DP stage in *Satb1*-deficient thymocytes, most notably around CNS0, where *Satb1* initially bound when chromatin was closed. In contrast, the effects of *Satb1* deletion on common-SEs and Tconv-SEs were less pronounced, making *Satb1*-deficient *tTreg* precursor cells closer to immature CD4SP thymocytes in the principal component analysis of global H3K27ac pattern (Fig. 8d and Supplementary Fig. 7g,h). These results collectively indicate that stage-specific *Satb1* deletion ‘preferentially’ impairs activation of Treg-SEs at the *tTreg* precursor stage.”

As seen from all three images extracted from *Kitagawa et al.* (Fig. 8d and Supplementary Fig. 7g,h), the H3K27ac levels in DP T cells are extremely similar in WT and *Satb1* cKO cells.

Regarding the second point, the following genes were differentially expressed in *Satb1* cKO and at the same time affected by the A->B compartmental change: *Car2*, *Hecw2*, *Inpp4b*, *Khdc1a*, *Klra13-ps*, *Klra3*, *Nudt16*, *Ppfibp2*, *Rab23*, *Rptn*, *Sgk3*, *Slc5a9*, *Spats2l*, *Zbtb16*, *Zfp518b*. We would like not to mention them in the revised manuscript since the A->B compartment switch is not relevant to the objective of this work. Moreover, based on the example of *Zbtb16* (encoding PLZF), these genes may just be artifacts due to the population-based nature of our data. Note that *Zbtb16* is not expressed in DP cells, mildly expressed in stage 0 (ST0) and more expressed in ST1 cells – different stages of iNKT development (*Gioulbasani et al., 2020; Kovalovsky et al., 2008; Savage et al., 2008*).

Since iNKT cell population is completely missing in *Satb1* cKO animals (Kakugawa et al., 2017; <http://dx.doi.org/10.1016/j.celrep.2017.04.038>), it could confound this analysis. The minimal number of genes affected by a compartment switch further supports our decision to focus on the chromatin loop scale structure where SATB1 seems to exert its role.

6. Line 187, chromatin compaction in cKO is based on HP1alpha, but this information alone may not be sufficient, as it could reflect increased in its expression. Some other method needs to be added to verify chromatin compaction, if indeed chromatin compaction occurs in cKO.

We do agree with the comment; however, this was one piece of information supporting the other approaches. We did show in our manuscript the decrease in chromatin accessibility as supported by ATAC-seq experiments (**revised Figure 4a/b/c, Supplementary Figure 5c**) and we could also see the compaction from multiple images from electron microscopy (**revised Supplementary Figure 5a**). This is also in line with the decreased MNase digestion of the *Satb1* cKO chromatin (**revised Supplementary Figure 5b**), the increased volume of heterochromatin regions as deduced by quantitative measurements of DAPI staining in *Satb1* cKO nuclei versus WT thymocyte nuclei (**revised Supplementary Figure 5d**, left graph) and the reduced volume of pRNA Pol II in the *Satb1* cKO nuclei versus WT thymocyte nuclei (**revised Supplementary Figure 5d**, right graph). All this is also supported by the higher number of genes with decreased expression seen from RNA-seq (**revised Supplementary File 2**). In the revised version of the manuscript, we omitted the HP1a staining as it did not provide additional value to our conclusions.

7. Quality of 3D interaction in Figure 4D (HiC) is not fully convincing. Arguing with one image that contains one small yellow box is not sufficient. This is an important point making the case for *Bcl6*, and therefore, show a larger scale including SE2 for HiC. Also, this reviewer suggests to show similar HiC image for H3K27ac HiChIP and SATB1 HiChIP (if Fig. 4D is simple HiC and not H3K27acHiChIP--unclear).

We should emphasize that in the **revised Figure 5a** (Hi-C heatmap) and its legend, it is clearly stated that the heatmaps come from Hi-C datasets. It should also be noted that the Hi-C heatmaps only supplement the key finding that *Bcl6* was a gene with the most dramatic changes in H3K27ac looping between WT and *Satb1* cKO. In the revised version of this manuscript, we replaced the original Figure 4a with a more refined correlation plot (**revised Figure 4h**) that also highlights the *Bcl6* gene, in respect to both H3K27ac looping and transcriptional deregulation. The complete differential analysis of H3K27ac loops is provided in the **Supplementary file 8**. *Bcl6* was also detected in SATB1-mediated loops as shown in the figure by the arcs (**revised Figure 5a**) and these loops were identified as statistically significant interactions based on the SATB1 HiChIP (i.e. the third dataset supporting the same conclusion, complementing the Hi-C and the H3K27ac HiChIP). In the revised version of this figure (**revised Figure 5a**), we also added the SATB1 HiChIP heatmap (top heatmap in Figure 5a) which indicates much stronger interactions compared to the WT Hi-C data. Following the recommendation of this Reviewer, we also extended the heatmap to include the SE2 in the revised Figure 5a.

8. There are multiple places where references are missing, or not placed earlier in the main manuscript. This could result in not providing an accurate background information.
 - 1) Line 72-72) Modify the sentence slightly to accommodate Kitagawa et al., paper and Cruz-Molina, Cell Stem Cell, 2017 paper. Both represent data supporting that chromatin structure precedes (or prime) correct future transcription.

We have modified the text in the revised version of the manuscript to “*Additionally, establishment of regulatory 3D chromatin arrangements often precedes changes in transcription during development and differentiation*^{15–18}”, (**page 3, lines 55-56**).

- 2) Line 125-140). References 73, 81 and 85, (especially 81 and 85) must be mentioned much earlier in the right context with clear explanation (preferentially in Introduction). Line 84 under “during T cell development (20), can accommodate these. Some explanation is required and this could continue in the result section (e.g. Line 127).

The following paragraph was added to the introduction: “*SATB1 has been attributed to many biological roles, mostly related to T cell biology²², but it also regulates the function of cell types such as the epidermis²³ and neurons^{24,25}. Mice with SATB1 being conditionally depleted in T cells display impaired T cell development accompanied with autoimmune-like phenotype^{17,26,27}.” (Page 4, lines 6670).*

In the results section, the entire paragraph was modified with the following changes in the suggested part: “*In line with previous studies^{26,27}, mice with SATB1 depleted from T cells had increased number of DP cells and decreased numbers of CD4⁺ and CD8⁺ single positive (SP) cells in the thymus, indicating a developmental blockade at the DP stage^{26,27,52}.” (Page 6, lines 111-113).*

- 3) SATB1 being both activator or repressor has been published long back (for instance, Han et al., 2006, nature).

It is not absolutely clear to which study of *Han et al., 2006* the Reviewer refers to, in the absence of either doi or a specific link. Probably the Reviewer refers to *Han et al., 2008* (<https://doi.org/10.1038/nature06781>), where the authors showed both up- and down-regulated genes upon SATB1-downregulation. However, we think that these data are not that strong in order to support that SATB1 is both an activator and a repressor. Though, we have included both *Han, 2008* and *Kumar, 2006* in the text as follows: “*SATB1 is known for its functional ambiguity of acting either as a transcriptional activator or a repressor.*^{35,66}” (Page 10, lines 221-222).

- 4) Although it is not a genome-wide study, for the cytokine cluster 200kb region, SATB1 has been published to form chromatin loop domains including regulatory regions. Induction of cytokine genes and looping were tightly linked. This earlier paper (ref 45) needs to be explained in the main text as it is highly relevant background paper to this current manuscript.

We appreciate this comment, however, we need to point out that the focus of our study has been on the roles of SATB1 in primary developing thymocytes (~90% DP T cells) from C57BL/6J mice, whereas *Cai et al.* investigated a differentiated Th2 cell clone (murine D10G4.1 conalbumin-specific type 2 helper T-cell clone, derived from the AKR/J mouse). There are many differences between the two settings, as there are extensive differences between primary T cells and immortalized cell clones, but mainly because there are extreme differences between the transcriptional programs and chromatin organization of developing T cells, naive T cells and especially differentiated T cells (see for example <https://doi.org/10.1038/emboj.2010.314>). We thus find that reference to *Cai et al.* is appropriate in the context of SATB1 as a genome organizer and it is properly cited in the manuscript (page 5, line 100, page 7, line 145).

- 5) Several critical T cell developmental genes have been published to be the targets of SATB1 (81, 85) and not all of them are found deregulated here. It is strange why Foxp3 was not found in this present study at all. Maf being SATB1 dependent is described in ref 45, as its induction requires SATB1.

The study by *Kitagawa et al., 2017*, where Foxp3 was found deregulated in *Satb1* cKO, was performed on regulatory T cells, unlike our study which focused on developing thymocytes, mostly DP T cells. First, it should be noted that even in *Kitagawa et al., 2017* the levels of FOXP3⁺ cells were normal at 4 weeks of age (the same age of animals that we used), due to peripheral Tregs present in

the thymus. Secondly, the CD4SP cells represent only ~7-8% of thymic population. From these 7%, the CD25⁺FOXP3⁺ T cells represent only ~2-3% of CD4SP cells which is a negligible population compared to the whole thymus that we have investigated. Therefore, no changes in *Foxp3* levels were anticipated and no changes were actually detected in our experiments (note that we did not detect any changes in *Foxp3* expression either in sorted DP T cells analyzed with RNA-seq).

Regarding *Maf* expression and the reference *Cai et al., 2006*, we explained in the previous point (point 4) that there are significant differences between *Cai et al.* and our study. Therefore, we cannot compare the expression levels of *Maf* between different cellular types used in these two studies.

- 6) Since SATB1 is very similar to YY1 in linking enhancer to promoter, it is better to introduce YY1 in Introduction.

The present study is focused on describing the roles of SATB1 and CTCF in developing T cells. Since there are similarities between YY1 and SATB1, we thought it is a great example for a discussion of factors that may function in a similar manner to SATB1. (**Page 17, lines 399-405**).

- 7) Line 119 onward). It sounds like the production of *Satb1*^{fl/fl} CD4-Cre⁺ were unique in this paper, but at least two other papers have already described these mice. Refer to them and highlight what is new.

Indeed, *Satb1*^{fl/fl} CD4-Cre⁺ mice have been previously utilized and uncovered deregulated T cell development and an autoimmune phenotype. In the revised manuscript these publications are included in the references as:

17. Kitagawa, Y. et al. Guidance of regulatory T cell development by *Satb1*-dependent superenhancer establishment. *Nat. Immunol.* 18, 173–183 (2017).
27. Kakugawa, K. et al. Essential roles of SATB1 in specifying T lymphocyte subsets. *Cell Rep.* 19, 1176–1188 (2017).

We make clear what is new to the prior studies. The work described in the latter publications is discussed and cited in the revised manuscript in:

Introduction

Lines 68-70: “Mice with conditionally depleted SATB1 from T cells display impaired T cell development accompanied with an autoimmune-like phenotype^{17,26,27}.”

Results

Lines 100-102: “Moreover, it is a known genome organizer^{50,51}, which was already found to be associated with enhancers in double positive CD4⁺ CD8⁺ (DP)^{17,27}, single positive CD4⁺ and developing thymic regulatory T cells¹⁷.”

Lines 111-113: “In line with previous studies^{26,27}, mice with SATB1 depleted from T cells had increased number of DP cells and decreased numbers of CD4⁺ and CD8⁺ single positive (SP) cells in the thymus, indicating a developmental blockade at the DP stage^{26,27,52}. Additionally, here we showed that...”

Lines 122-124: “The absence of naïve CD4⁺ T cells together with increased levels of DP T cells in the spleen (Fig. 1g) were suggestive of an autoimmune-like phenotype⁵³, consistently with other studies^{17,26,27}.”

Lines 130-132: “Impaired T cell development in *Satb1* cKO animals is associated with changes to the transcription programs as previously documented^{17,27}. Here and in our accompanying publication⁵⁴, we have described the vast transcriptional deregulation in the *Satb1*^{fl/fl}CD4-Cre⁺ animals.”

Lines 327-330: “It is worth noting that in line with a previous study¹⁷, 1D H3K27ac ChIP-seq peaks derived from HiChIP experiments available for WT and *Satb1* cKO did not reveal any major differences between the genotypes, which further reinforces the importance of SATB1-mediated 3D chromatin organization regulating *Bcl6* expression.”

Discussion

Lines 425-427: “Previous research suggested a cell-extrinsic mechanism of autoimmunity based on the deregulation of regulatory T cells¹⁷.”

Lines 444-446: “Moreover, the blockade at stage 0 (ST0) of iNKT development in *Satb1*-deficient mice was previously reported²⁷, collectively suggesting a potential link between SATB1-mediated regulation of *Bcl6* and these developmental programs.”

Methods

Lines 907-913: “Publicly available ChIP-seq datasets: To validate the reliability of our SATB1 binding sites dataset, we compared it to two SATB1 ChIP-seq datasets that were prepared using standard SATB1 antibodies non-selectively targeting all SATB1 isoforms (Abcam, ab109122⁸⁵ & ab70004¹⁷). The first SATB1 ChIP-seq (GSM1617950⁸⁵) is already provided with the processed SATB1 peaks, hence these were only converted from mm9 to mm10 mouse genome assembly using CrossMap¹²⁷ to match our dataset. The second dataset (DRR061108¹⁷) was completely re-analyzed.”

- 8) Line 138) Again, earlier work must be mentioned about the T cell phenotypes upon SATB1 ablation, and then highlight new results found in this paper (or say, consistent with...).

We have revised all relevant text and it should now be clear which findings are consistent with others and which are novel in this work. Please also see answer in previous Point #7.

Additional points

- 1) Line 148) State what is meant by 77% “unchanged” TADs. Unchanged in what aspects?

Please note that in response to comments from the other Reviewers, we modified this analysis to also include information about TADs in CTCF- and RAD21-depleted cells.

The revised text that now refers to TADs is: (**lines 154-156** of the revised manuscript) “As anticipated, the knockout Hi-C datasets for RAD21 and CTCF revealed a perturbation of TADs, which were mostly unaffected in the *Satb1* cKO thymocytes (Fig. 2b, Supplementary Fig. 4b).”

The legend of **Supplementary Figure 4b** now reads: “Differential analysis of topologically associating domains using TADCompare (see the official manual for explanation of differential categories⁵) at 100 kbp resolution between WT and factor-depleted cells (combined biological replicates). SKO represents *Satb1*^{fl/fl} Cd4-Cre⁺ murine thymocytes, RKO –*Rad21*^{fl/fl} Cd4-Cre⁺ DP murine thymocytes⁶ and CKO – CTCF-depleted (AID degron system) mESCs⁷. Black dots represent different TAD categories for individual chromosomes. The red circles represent the mean \pm s.d. P values by Wilcoxon rank sum test.”

- 2) Line 150) State resolution of Hi-C here after “broad scale differences”

We have modified the text on the first call of the **revised Figure 2a** to include the information about resolution and normalization and we also included this information in the figure legend, as follows:

Lines 148-150: “To address this, we performed Hi-C experiments² in both WT and *Satb1* cKO thymocytes (Supplementary File 4). Notably, we did not identify any major changes in high-order chromatin organization (Fig. 2a; 500 kbp resolution, balanced normalization).”

Legend Figure 2a: “Comparison of WT and *Satb1* cKO Hi-C heatmaps at 500 kbp resolution (balanced normalization) of chromosomes 9 and 11 indicates no major changes at the high order chromatin level of murine thymocytes.”

3) Line 162-163) The explanation is unclear in the main text. The corresponding Figure legend was clearer.

This entire paragraph (revised manuscript, lines 175-203) was extensively modified in response to other Reviewers' comments. We have also modified this particular part to make it more clear as follows:

Lines 175-181: “We made several comparisons to infer the impact of both SATB1 and CTCF on chromatin organization. First, we performed aggregate peak analysis (APA¹) to show that SATB1 HiChIP loops had stronger interaction signal in WT Hi-C datasets than in *Satb1* cKO Hi-C. CTCF HiChIP loops retained the same APA score in both Hi-C datasets, indicating that CTCF-based high order chromatin organization remained unaltered in *Satb1* cKO (Supplementary Fig. 4f).”

Legend of Supplementary Figure 4f: “Aggregate peak analysis¹⁴ was calculated and visualized by Juicer Tools¹⁵ to show that SATB1 HiChIP loops had stronger signal in WT Hi-C datasets than in *Satb1* cKO Hi-C. CTCF HiChIP loops retained the same APA score indicating that CTCF-based high order chromatin organization remained unchanged in *Satb1* cKO.”

4) Line 166) Regarding Extended Data Fig. 1e, what is the significance of this analysis?

Please see also previous comment 3. The importance of this finding is described in the text. It is one of the findings/analyses indicating that SATB1-mediated 3D chromatin interactions are diminished in the knockout cells.

6) Line 82-83) Change sentence to “We have identified SATB1 to be enriched at gene promoters ---
-“.

We modified the sentence as follows: “We have identified SATB1 to be enriched at gene promoters and enhancers involved in long-range chromatin interactions”. (Lines 65-66).

Reviewer #2

This manuscript presents a detailed analysis of the genomic topology features that depend on the transcription factor SATB1 in developing T cells, and a survey of their impacts on gene expression in the cells. As SATB1 is important in multiple unrelated tissues, the authors begin by establishing a highly specific T-lineage-restricted conditional knockout SATB1 mouse line in which the *Satb1* gene is deleted selectively at the beginning of the DP stage in the thymus, which is a common source of all mature alpha beta lineage T cells. This mouse proves to replicate the autoimmunity phenotype of previously described germline *Satb1*-knockouts, confirming the functional abnormality of the *Satb1*-deficient T cells in vivo. The rest of the paper focuses on identifying the origins of these abnormalities by study of total thymocytes (>80% DP cells) from these *Satb1* conditional knockout mice. The analysis combines Hi-C, SATB1 Hi-ChIP (compared with CTCF Hi-ChIP), and RNA-seq analyses of these thymocytes with previous data sources identifying “predicted thymic enhancers” and ChIP-seq analyses of H3K27ac, H3K4me1 and H3K4me3, similarly in unfractionated thymocytes, but from wildtype cells.

The results show that SATB1 protein is needed for particular local loops to form, especially affecting genes involved in T cell function (a marked difference from the loops requiring CTCF), and that when these loops connect predicted enhancers with promoters, the corresponding genes are often altered in expression, usually although not always losing expression in the absence of SATB1. Of particular interest, the loss of SATB1 in DP stage thymocytes has particularly strong effects on TCR α gene rearrangement success. First, it causes a marked depression of TCR α -V segment transcription and bisects the Traj region into a segment that shows increased accessibility and transcription (encoding distal J segments) and a segment that shows decreased accessibility and transcription (proximal J segments). Second, it also inhibits Rag1 expression substantially.

There is a great deal of value in this paper, and the statistical and modeling analyses particularly are described with unusual care and lucidity. The results are intrinsically very interesting. However, the effects of the SATB1 loss on local formation of loops seem surprisingly weak, and the KO also shows only mild effects on ATAC accessibility. It is not completely clear, then, whether the authors conclude that their meticulous looping analysis has revealed a distinctive role of SATB1 different from other sequence-specific transcription factors. In addition, there are several questions about how basic aspects of the study were characterized that the authors should address, and some additional questions about the order of the presentation.

1. While the pathology of the *Satb1* cKO mice is clear evidence that the KO makes a difference, the actual thymocytes used for the analysis are not well defined. The problem is that most, but not all of the cells in the thymus are DP cells, but there is a considerable body of evidence (ranging from Scollay, 1990's to Hogquist, recently) that the average dwell time of normal thymocytes in the DP stage is no more than 3-5 days at most. With the CD4-Cre knockout only deleting *Satb1* in this stage, it is possible that some cells in the majority population of the thymus at a given time have lost *Satb1* too recently to deplete the protein from the chromatin. This would yield an underestimate of the importance of SATB1 for loop formation.

(a) How well is the SATB1 protein depleted in actual SATB1 cKO thymocyte samples? Any evidence, e.g. from Western blots or intracellular protein staining of these thymocyte populations, would be very helpful. If feasible, intracellular staining could also be helpful to indicate whether the depletion is uniform across the population, also relevant as in the next comment.

We thank the Reviewer for this valuable point, which indeed if true would underestimate the effect/importance of SATB1 on loop formation in thymocytes. As suggested, we have performed the experiments presented in the revised **Supplementary Figure 2**.

To check the efficiency of SATB1 depletion in *Satb1* cKO thymocytes we have performed the following experiments:

(a,b) FACS analysis for both WT and *Satb1* cKO thymocytes for the detection of **EYFP expression** in the cells. For these experiments we have utilized *Satb1* cKO mice that were also encompassing the EYFP gene as a transgene in the ROSA26 locus which is not expressed unless Cre-mediated excision of a *loxP*-flanked transcriptional "stop" sequence permits EYFP expression and the detection of EYFP. This does not directly answer the Reviewer's question but at least identifies the fraction of thymocytes that express the Cre recombinase which can act upon the deletion of *Satb1*. We found that more than 96% of the total *Satb1* cKO thymocytes and more than 97% of DP thymocytes expressed the EYFP reporter, indicating the efficient expression of the Cre recombinase expressed under the *Cd4* promoter.

(c,d) As suggested, we have performed **intracellular staining for SATB1** protein, in both WT and *Satb1* cKO thymocytes and found that 8.7-13.9% of total *Satb1* cKO thymocytes still expressed the SATB1 protein. 5.6-10.8% of the double positive (CD4⁺CD8⁺) *Satb1* cKO thymocytes still expressed the SATB1 protein.

(e) **Immunofluorescence experiments** analyzed by confocal microscopy indicated that SATB1 is expressed in 97.11% of wild type thymocytes and 4.16% of *Satb1* cKO thymocytes, utilizing a commercially available antibody detecting all SATB1 isoforms. The relative SATB1 fluorescence signal was calculated in a quantitative manner utilizing the Volocity software from PerkinElmer.

(b) The genes that are downregulated in *Satb1* cKO cells compared to WT include genes that normally increase sharply from DP to SP (*Ccr7*, *Cd40lg*) as well as genes that decrease sharply from DP to SP (*Rag1*, *Bcl6*). Are these gene expression changes happening in the same cells? Conceivably subsets of DP cells could be separated based on these other surface molecules, or based on cell-surface TCR α expression, CD5, and/or CD69 staining, sufficiently to use qRT-PCR to check which cells express these genes. Extended data table 6, which is extremely informative, suggests that the two late-expressed genes might be affected indirectly as a result of a general block of differentiation, in which case they could be downregulated without direct promoter-enhancer loop interactions with SATB1 itself.

To address these concerns, we utilized ImmGen RNA-seq data from sorted T cell populations. This allowed us to identify signature genes for DN, DP and SP T cell subsets (**Supplementary Figure 4a**). Note that we considered them as signature genes when preferentially expressed in the corresponding T cell subsets. We found a significant overlap enrichment of the DP T cell subset signature genes and the differentially expressed genes in the *Satb1* cKO (**Figure 1j,k**). Moreover, this analysis revealed that the regulatory SATB1 loops were highly specific for DP T cells (**revised Figure 4g, Supplementary Figure 7c,d**). Additionally, a correlation plot for significantly transcriptionally deregulated genes showed a strong positive correlation (Spearman's $\rho = 0.62$) between the looping changes and transcriptional deregulation of *Satb1* cKO animals. An indication of causality for this effect is demonstrated by the **revised Figure 4f**, where we specifically compared the differential *Satb1* cKO under- and over-interacting H3K27ac loops that are or are not mediated by SATB1/CTCF, and the expression changes of genes affected by these loops.

These images clearly indicate that genes at the anchors of SATB1-mediated H3K27ac loops that were diminished in *Satb1* cKO, were significantly underexpressed, unlike the genes in either the CTCF-mediated loops or any factor-mediated overinteracting H3K27ac loops. Although it cannot indicate whether the deregulation happened in the same cell, it specifically shows that the SATB1-mediated regulatory loops are highly important for the DP T cell subset.

2. In this paper, the authors use a novel home-made antibody against the long isoform of SATB1. However, the report of the characteristics of this antibody has not been published, and it is not clear whether all the thymocyte populations or only a minority express this long isoform. If the characterization paper still is not available, then some details about the properties of the new antibody should be described. It is especially important to know whether this antibody is recognizing the same SATB1 sites that are reported using other reagents, and how well its specificity for SATB1 is established.

Our second manuscript describing the long SATB1 isoform is now deposited in bioRxiv (*T. Zelenka, P. Tzerpos, G. Panagopoulos, K-C. Tsolis, D-A. Papamatheakis, V.M. Papadakis, D. Stanek, C. Spilianakis, SATB1 undergoes isoform-specific phase transitions in T cells*, doi: <https://doi.org/10.1101/2021.08.11.455932>). As already discussed in the reply to comment 4 of Reviewer #1, in the bioRxiv manuscript we described the specificity of the long SATB1 isoform-specific antibody (**Figures 1e-f, Figure 2a, Figure 3e, Supplementary Figure 1a,b, Supplementary Figure 3**) and we also provided pixel-based colocalization analysis of super-resolution microscopy experiments based on the long SATB1 isoform-specific antibody and general SATB1 antibodies non-

selectively targeting all SATB1 protein isoforms. Western blot experiments comparing the two antibodies have also been provided.

Additionally, we have included into our revised manuscript a comparison of the SATB1 long isoform-specific binding sites (identified in this manuscript) with two publicly available SATB1 ChIP-seq experiments (*Hao et al., 2015; Kitagawa et al., 2017*). Based on the Relative Distance metric of Bedtools (see **Supplementary Notes**), the long SATB1 isoform-specific binding sites were non-randomly spatially correlated with both publicly available SATB1 ChIP-seq datasets (which utilized antibodies targeting all SATB1 isoforms; **revised Supplementary Figure 4d**). Moreover, the genomic feature association analysis (Genome Ontology) for the three datasets of SATB1 binding sites, analyzed by the AnnotatePeak function of Homer showed highly comparable binding preference for all SATB1 antibodies (**revised Supplementary Figure 4e**).

Overall, we should note that the SATB1 HiChIP loops are long isoform-specific and these are the loops that were positively correlated with transcriptional deregulation of affected genes in *Satb1* cKO as well as with H3K27ac loop deregulation. Using an unbiased linear regression modeling approach, these long isoform-specific SATB1-mediated loops were identified as good predictors for *Satb1* cKO underexpressed genes as visualized in the **revised Supplementary Figures 6 and 7a**.

3. It is important to explain how many independently derived biological replicates were involved in all the new data, Hi-C, Hi-ChIP, ATAC-seq, and RNA-seq. Some of the effect sizes are small enough (e.g. Ext. Fig. 1e,g; Fig. 3c) so that their reproducibility is a strong concern.

We performed two biological replicates for each genotype (described in the methods section of the revised manuscript, **lines: 602-603**) for each SATB1 HiChIP, CTCF HiChIP, H3K27ac HiChIP and Hi-C experiments. We performed three biological replicates for each genotype for ATAC-seq (described in the methods, **line 794**) and RNA-seq (added to the methods, **line 769**). Such a number of replicates is a standard in similar omics studies and they allowed us to uncover a number of deregulated genes and regions. Additionally, we have also included the number of experimental animals for tissue section and flow cytometry experiments in each figure.

4. The organization of the paper is a little strange, making it hard for the reader to follow some aspects of the story.
 - (a) A reader would expect to see the full information about the organ that is being studied in the cKO, namely the thymus, presented all together at the beginning, rather than having cell percentages presented in Fig. 1 and Fig. S1 but cell numbers and viabilities reserved for Fig. 6.

We have modified the manuscript accordingly and in the revised manuscript we moved all the information regarding the thymus in **revised Figure 1**.

- (b) Similarly, an obvious early question to ask about removing a transcription factor is what genes are transcriptionally affected. It is surprising that this whole subject is deferred till Figs. 5 and 6, and Ext. Fig. 4, leading to questions about whether the chromatin-level changes that are seen have any impact on function – and whether they are specifically occurring at loci that change expression.

In the revised version of the manuscript, we placed the information regarding transcriptional deregulation earlier in the text (**Lines 130-140, revised Figure 1j,k and Supplementary File 2-3**).

- (c) In general, the reader wishes that the authors would make a more direct statement about how important they judge the SATB1-mediated looping to be in mediating gene expression control in these thymocytes. Are they being reticent because the effects on ATAC-seq and even RNA-seq tracks shown in Ext. Fig. 4 are modest? It would be valuable in any case to make a clear statement of the relationship that is seen.

In the revised version of the manuscript and in our responses to the Reviewers' comments, we provided evidence that the transcriptional changes are not modest and that they are highly positively correlated with the function of SATB1. In the revised version of the manuscript, we emphasized the vast deregulation much earlier in the text:

Lines 130-134: *“Impaired T cell development in Satb1 cKO animals is associated with changes to the transcription programs as previously documented^{17,27}. Here and in our accompanying publication⁵⁴, we have described the vast transcriptional deregulation in the Satb1^{fl/fl} Cd4-Cre⁺ animals. Stranded total RNA sequencing revealed that 922 genes were significantly underexpressed and 719 genes were significantly overexpressed in the Satb1 cKO compared to WT thymocytes (FDR<0.05; Supplementary File 2).”*

Lines 215-220: *“Both this and the chromatin accessibility changes were reflected at the transcriptional level, as demonstrated by the vast transcriptional changes in the Satb1 cKO compared to WT thymocytes⁵⁴ (Supplementary File 2). Such a strong deregulation of the transcriptional landscape in Satb1 cKO cells in contrast to the modest transcriptional changes observed upon depletion of conventional genome organizers⁶⁻⁸ emphasizes the regulatory importance of SATB1-dependent chromatin organization.”*

5. In considering the results in Extended Data 6, it seems worth considering some additional criteria that might sharpen the relationship between SATB1-mediated enhancer-promoter looping and gene regulation.

First, is it possible, that Tcra and Bcl6 show a stronger effect than the other targets precisely because they are the only genes in this set that turn on for the first time in the DP stage, when the SATB1 deletion is being induced? Tcf7, Lef1, Ikzf1 and even Cd8a would all normally be activated before the CD4-Cre could be strongly enough expressed to delete Satb1, and this could contribute to a high background of expression.

We thank Reviewer #2 for this quite interesting point of view. We agree that this fact may contribute to a smaller/underestimated effect for some genes. Though, we should emphasize that there were hundreds of genes that were significantly transcriptionally deregulated irrespective of this effect. Although the effect may be weaker for some genes, the objective of this manuscript was to identify the regulatory chromatin network in developing T cells. Using an unbiased approach, we have identified SATB1 as a great candidate mediating the regulatory loops. Although our methodology does not strictly reflect the situation in DP cells, DP cells represent ~90% of our analyzed population. By employing a thorough computational analysis, we could partially resolve these complex datasets and convincingly relate the transcriptional data with multiple 3C-based datasets. Moreover, as discussed in **comment 1b** of this Reviewer, by selecting signature genes for DN, DP or SP T cell subsets, we clearly showed that the signature genes for DP cells strongly overlapped with differentially expressed genes in Satb1 cKO cells and SATB1- associated loops.

Second, as noted above, some of the genes that are affected without showing clear evidence of SATB1-dependent loops (overlinked – underlinked >1) normally are not expressed at all during the DP stage, when most Satb1 cKO thymocytes are arrested. Would removal of these maturation-dependent genes yield a higher correlation between looping and transcriptional responsiveness? Thus, it might be useful to employ ImmGen RNA-seq data to filter the target genes on their normal developmental patterns of expression and then check for correlation with SATB1-dependent looping.

We thank the Reviewer for these great suggestions. We have implemented these suggestions as discussed in **comment 1b**. In this regard, we have generated the correlation plots between differential H3K27ac loops and gene expression changes (**revised Figure 4h** and **Supplementary Figure 7d**).

Unlike in the original analysis where we used all the genes to generate this plot, in the revised version of the manuscript we used plots specific for separately DN, DP and SP T cell subset signature genes and additionally also for all the significantly transcriptionally deregulated genes (**revised Supplementary Figure 7d**). From all these plots (together with the overlap enrichments analyses provided in **Figure 1j-k**, **Figure 4g**, **Supplementary Figure 7c**) it is clear that DP T cell signature genes are the most affected by SATB1-mediated regulatory loops. Additionally, it is clear that compared to the correlation plot with all the genes plotted (**revised Figure 4h**), the significantly deregulated genes evince much stronger correlation with SATB1-mediated H3K27ac regulatory loops (**revised Supplementary Figure 7d** – the rightmost graph), collectively supporting our former conclusions.

6. **Minor:** Extended Fig. 3 shows results of a very interesting model, but the figure is extremely difficult to see. First, the legend does not do a good enough job to explain panel (a). Presumably this is showing striking compaction of the chromosome predicted in the presence of SATB1, but one needs to be able to visualize the DNA strand better against the dark background even when it is printed out. How realistic is it to assume that the *Bcl6* locus in the absence of SATB1 would exist in such a stretched-out, linearly extended form? Second, panel (b) must be enlarged substantially, as it is really hard to see the viridis color coded regions around the *Bcl6* promoter and SE1 enhancer. (I could only see them by zooming in on the pdf on the monitor; the printout was not at all clear.) It could be worthwhile to add a set of zoom-in panels for the lower left sections of each of these panels in Extended Fig. 3b, so that these key features of the model predictions can be appreciated unambiguously.

We thank the Reviewer for these suggestions. We have improved the quality of both models to make them bigger and clearer. We do not think that the *Bcl6* locus being stretched-out as in the model, represents the true chromatin shape in the nucleus since there are many more factors contributing to the overall 3D structure of the locus and since this is an ensemble of many individual cells. Nevertheless, the model should help us understand the importance of SATB1 in CTCF-dependent chromatin structure and particularly its significance in mediating the proximity between *Bcl6* and its super-enhancer regions.

Reviewer #3

In the current study, Zelenka et. al. investigated the role of genome organizer *Satb1* in thymocytes and how the 'loopscape' mediated by *Satb1* affects the developing T cells. The authors delve into the global chromatin architecture by using various assays including HiC, HiChIP and ATAC-seq and correlated with the transcriptomic modulations upon ablation of *Satb1* in CD4+ T cells. Almost at every level of analysis, they have drawn parallels with the organizer CTCF to demonstrate *Satb1* mediated changes. A positive aspect of the work described in the manuscript is the authors have presented myriad of analyses to demonstrate how *Satb1* regulates the nuances in thymic gene regulation (e.g. using the *Bcl6* and *Tcr* loci) primarily through differential enhancer-promoter contacts. The authors have also extended previous autoimmune related findings to other organs such as pancreas. The methods are written well and extensive. However, the conclusions drawn from this data are not always consistent within the present analysis and also do not correlate with what has been shown by others. The primary conclusion is that the SATB1 mediates promoter-enhancer loops affecting master regulator genes, collectively being critical for cell lineage specification and immune system homeostasis. This key point is related to a first weakness of the manuscript, which is that the authors have chosen to perform this study using one specific isoform of SATB1 that is not the abundant form. The rationale for using this specific isoform is not clear. The functional importance of this isoform is mentioned in the discussion section but the reference cited for the same is a submitted manuscript

hence it can't be cross-checked. Secondly, the cKO model used eliminates all isoforms of *Satb1* hence the functional data can't be correlated with the assays such as HiChIP using antibody against the long isoform. Although the data shown provides additional information on the role of *Satb1* as a genome organizer, the manuscript suffers from major pitfalls and requires substantial revisions to unequivocally support their conclusions.

Reviewer #3 is absolutely right regarding the availability of our manuscript describing a long SATB1 isoform. Though, we should note that we had informed the Editor regarding the second manuscript describing the long SATB1 isoform and we would be more than happy to provide the manuscript when needed. It is currently available in **bioRxiv** (*T. Zelenka, P. Tzerpos, G. Panagopoulos, K-C. Tsolis, D-A. Papamatheakis, V.M. Papadakis, D. Stanek, C. Spilianakis, SATB1 undergoes isoform-specific phase transitions in T cells*, doi: <https://doi.org/10.1101/2021.08.11.455932>), so we have corrected the reference in the revised manuscript. Following that, we made it clear why we have selected the long isoform-specific antibody. We believe that the use of the long isoform-specific antibody will be valuable for the research community especially when connected to expression in malignancies. Different isoforms of the same protein can exhibit remarkably different functions and we provide evidence for such behavior between SATB1 isoforms in our second manuscript where the long SATB1 isoform seems to be more frequently associated with active transcription. We also show that the SATB1 long isoform is expressed at the same if not higher protein levels in murine thymocytes (**Figure 1e,f** of our bioRxiv manuscript). Therefore, we believe that it is an asset performing studies that could discriminate the mode of action of different isoforms of the same protein. This is especially valid for SATB1 protein which displays genome organizer functions and diverse effects on gene transcription regulation. It is true that in the *Satb1* cKO animals, all isoforms were depleted. However, we should note the following:

- 1) The only datasets that are confined to the long isoform are the SATB1 HiChIP and inferred SATB1 binding sites. Our conclusions on chromatin accessibility, H3K27ac-associated loops and gene expression changes in the *Satb1* cKO cells are robust and describe the overall functions of all SATB1 isoforms in murine thymocytes. Please note that our analyses indicate globally an activatory role for SATB1 more than a repressive one. This can be reflected on the number of overexpressed vs underexpressed genes, chromatin accessibility changes observed at promoters and overall deregulated heterochromatin organization (**revised Supplementary Figure 5**).
- 2) Long SATB1 isoform foci based on the immunofluorescence staining utilizing super-resolution microscopy display higher colocalization with 5-fluorouridine-labeled sites of active transcription than collectively all SATB1 isoform foci detected by common, non-isoform-specific, SATB1 antibodies (**Figure 3e** of our bioRxiv manuscript). Moreover, colocalization between sites of active transcription and the long SATB1 isoform in primary murine T cells was more sensitive to 1,6-hexanediol treatment than for the all SATB1 isoform foci detected by isoform non-specific SATB1 antibodies (**Figure 3e** of our bioRxiv manuscript). Following the current model of transcriptional regulation based on liquid-liquid phase separation, this indicates the involvement of the long SATB1 isoform in transcriptional regulation.
- 3) Long SATB1 isoform binding sites and associated chromatin loops tend to be more related to transcriptional activation given that:
 - Long SATB1 isoform binding sites display higher chromatin accessibility than expected by chance (**revised Figure 4a**),
 - Long SATB1 isoform binding sites display drop in chromatin accessibility in the absence of SATB1 (**revised Figure 4b**) which is especially evident at the transcription start sites of genes (**revised Figure 4c** and **Supplementary Figure 5e**),
 - Genes found in long SATB1 isoform-mediated H3K27ac loops underinteracting in *Satb1* cKO display significantly lower expression than genes in H3K27ac

underinteracting loops which are not SATB1-mediated, and there are no significant differences in the expression of genes found in H3K27ac overinteracting loops (**revised Figure 4f**),

- Unbiased models show association of the long SATB1 isoform-mediated chromatin loops containing enhancers and drops in gene expression levels (**revised Supplementary Figure 6f and Supplementary Figure 7a**).

Comments

1. What is Satb1 'loopscape'?

It is a term describing chromatin structure which was previously adopted in the SATB1 literature.

(<https://doi.org/10.1016/j.gde.2007.08.003>, S. Galande, P.K. Purbey, D. Notani, P.P. Kumar (2007) *The third dimension of gene regulation: organization of dynamic chromatin loopscape by SATB1*, *Current Opinion in Genetics & Development*, 17: 408-414; <https://doi.org/10.3389/fimmu.2021.669881>, L. Scourzic, E. Salataj, E. Apostolou (2021) *Deciphering the Complexity of 3D Chromatin Organization Driving Lymphopoiesis and Lymphoid Malignancies*, *Front Immunol.* 12:669881).

2. Line 115. The authors mention "We drew our attention to SATB1, which displayed significant enrichment at the H3K27ac 115 loop anchors (Fig. 1b) and also represents a known genome organizer." The authors have excluded key publications which have looked at SATB1 binding alongside the H3K27ac mark (eg Kitagawa 2017). The authors should introduce these findings in order to build the premise for their work.

We thank the Reviewer for the comment. We have added references indicating that SATB1 was previously found to be associated with enhancers in DP, CD4SP and in developing thymic regulatory T cells. "Moreover, it is a known genome organizer^{50,51}, which was already found to be associated with enhancers in double positive CD4+CD8+ (DP)^{17,27}, single positive CD4+ and developing thymic regulatory T cells¹⁷, yet with a limited number of genome-wide studies targeting its role in 3D chromatin organization of T cells." (**Lines 100-103** on the revised manuscript).

0. The authors have used pre-analyzed datasets from ChIP-Atlas for generating Figure 1a. The conditions under which the ChIP seq for the factors in question are not stated anywhere. The authors should perform their own ChIP seq (at least for SATB1 is much essential) replicating the conditions which were used to make H3K27ac HiChIP samples.

Please note that we utilized the publicly available tool ChIP-Atlas for the following reason: the enrichment analysis of this tool allows unbiased search for factors that are enriched at the investigated regions – in our case at WT H3K27ac loop anchors. In this regard, it is not clear what conditions should be specified. In the methods section of our manuscript, we clearly describe the conditions under which the ChIP-Atlas was used (**lines 900-906**, revised manuscript). The conditions of the individual ChIP-seq experiments deposited in SRA database (used by ChIP-Atlas) are in this case not important for the prediction. To facilitate the search for individual ChIP-seq experiments, we have added a complete output of ChIP-Atlas into the supplementary section (**revised Supplementary file 1**) and we removed the original Figure 1a which would otherwise be redundant. We should also clarify that we have generated our own set of SATB1 binding sites as described in the methods section (**lines 707-711**, revised manuscript). These were produced based on a HiChIP experiment with >60% Dangling End Pairs (~280 million reads), therefore these were prepared under the exact conditions as H3K27ac HiChIP samples. Note that binding sites inferred from HiChIP experiments (given the nature of HiChIP experiments) can be compared to traditional ChIP-seq peaks and especially the very

high number of reads that we used compared to common ChIP-seq experiments ensures great coverage of our data.

In the revised version of this manuscript, we additionally provide a qualitative comparison to two publicly available SATB1 ChIP-seq experiments. Therefore, we have included a comparison of the SATB1 long isoform-specific binding sites (identified in this manuscript) with two publicly available SATB1 ChIP-seq experiments (*Hao et al., 2015; Kitagawa et al., 2017*). Based on the Relative Distance metric of Bedtools, the long SATB1 isoform-specific binding sites were non-randomly spatially correlated with both publicly available SATB1 ChIP-seq datasets (which utilized antibodies targeting all SATB1 isoforms; **revised Supplementary Figure 4d**). Moreover, the genomic feature association analysis (Genome Ontology) for the three datasets of SATB1 binding sites, analyzed by the AnnotatePeak function of Homer showed highly comparable binding preference for all SATB1 antibodies (**revised Supplementary Figure 4e**).

4. While expounding the autoimmune phenotype of *Satb1* cKO mice, the authors again fail to cite the original and subsequent findings which showed that SATB1 deletion in CD4 leads to autoimmunity. The authors should include and discuss their findings in the text to support their observations. Furthermore, the authors mention in line 125 that there is a reduction of SP cells in the thymus.

We fully agree with the Reviewer on this. We have revised the entire part discussing the *Satb1* cKO mouse phenotype and it should now be clear which parts are consistent with others and which are novel in this work. *Satb1^{f/f} Cd4-Cre⁺* mice have been previously utilized and uncovered deregulated T cell development and an autoimmune phenotype. In the revised manuscript these publications are included in the references as:

17. Kitagawa, Y. et al. Guidance of regulatory T cell development by *Satb1*-dependent superenhancer establishment. *Nat. Immunol.* 18, 173–183 (2017).
27. Kakugawa, K. et al. Essential roles of SATB1 in specifying T lymphocyte subsets. *Cell Rep.* 19, 1176–1188 (2017).

We make clear what is new to the prior studies. The work described in the latter publications is discussed and cited in the revised manuscript in:

Introduction

Lines 68-70: *“Mice with conditionally depleted SATB1 from T cells display impaired T cell development accompanied with an autoimmune-like phenotype^{17,26,27}.”*

Results

Lines 100-102: *“Moreover, it is a known genome organizer^{50,51}, which was already found to be associated with enhancers in double positive CD4⁺ CD8⁺ (DP)^{17,27}, single positive CD4⁺ and developing thymic regulatory T cells¹⁷.”*

Lines 111-113: *“In line with previous studies^{26,27}, mice with SATB1 depleted from T cells had increased number of DP cells and decreased numbers of CD4⁺ and CD8⁺ single positive (SP) cells in the thymus, indicating a developmental blockade at the DP stage^{26,27,52}. Additionally, here we showed that...”*

Lines 122-124: *“The absence of naïve CD4⁺ T cells together with increased levels of DP T cells in the spleen (Fig. 1g) were suggestive of an autoimmune-like phenotype⁵³, consistently with other studies^{17,26,27}.”*

Lines 130-132: *“Impaired T cell development in *Satb1* cKO animals is associated with changes to the transcription programs as previously documented^{17,27}. Here and in our accompanying*

publication⁵⁴, we have described the vast transcriptional deregulation in the *Satb1^{fl/fl}/Cd4-Cre* + animals.”

Lines 327-330: “It is worth noting that in line with a previous study¹⁷, 1D H3K27ac ChIP-seq peaks derived from HiChIP experiments available for WT and *Satb1* cKO did not reveal any major differences between the genotypes, which further reinforces the importance of SATB1-mediated 3D chromatin organization regulating *Bcl6* expression.”

Discussion

Lines 425-427: “Previous research suggested a cell-extrinsic mechanism of autoimmunity based on the deregulation of regulatory T cells¹⁷.”

Lines 444-446: “Moreover, the blockade at stage 0 (ST0) of iNKT development in *Satb1*-deficient mice was previously reported²⁷, collectively suggesting a potential link between SATB1-mediated regulation of *Bcl6* and these developmental programs.”

Methods

Lines 907-913: “Publicly available ChIP-seq datasets: To validate the reliability of our SATB1 binding sites dataset, we compared it to two SATB1 ChIP-seq datasets that were prepared using standard SATB1 antibodies non-selectively targeting all SATB1 isoforms (Abcam, ab109122⁸⁵ & ab70004¹⁷). The first SATB1 ChIP-seq (GSM1617950⁸⁵) is already provided with the processed SATB1 peaks, hence these were only converted from mm9 to mm10 mouse genome assembly using CrossMap¹²⁷ to match our dataset. The second dataset (DRR061108¹⁷) was completely re-analyzed.”

5. Although in Figure 1c, there is no percentage indicated. Just by looking at Fig 1c, it seems that there is an increase in CD4 and CD8SP upon cKO of *Satb1*. This should be clarified and gatings should be carefully checked. Additionally, showing levels of intracellular *Satb1* in these validation figures upon gating is essential.

The Reviewer is correct, we have not included the percentages of cell populations on the graphs, in Figure 1c. Though, the percentages of cell populations were indicated on the right part of Figure 1c (graphs on the right) and in great detail in Supplementary Figure 1b. In the revised version of the manuscript, we made this clearer by referring to the mean percentages in the figure (**revised Figure 1d,g,h** and **revised Supplementary Figure 3c**). We also clarified the gating strategy in the methods section (**lines 485-503**, revised manuscript) and in the Reporting Summary. Otherwise for both CD4SP and CD8SP there is a decrease in cell populations (**revised Supplementary Figure 3c**) which is in line with publications published earlier.

We thank the Reviewer for the suggestion. We have performed the experiments as presented in the revised **Supplementary Figure 2c,d**. As suggested, we have performed intracellular staining for SATB1 protein, in both WT and *Satb1* cKO thymocytes and found that 8.7-13.9% of total *Satb1* cKO thymocytes still expressed the SATB1 protein. 5.6-10.8% of the double positive (CD4⁺CD8⁺) *Satb1* cKO thymocytes still expressed the SATB1 protein.

0. The authors have used a conditional knockout (cKO) model of *Satb1* for all studies reported here. However, the phenotypes described here do not correlate with the other published models of *Satb1* cKO. Furthermore, the phenotype shown in Extended figure 1a., the phenotype of CD4cre+*Satb1*fl mice shown has not been discussed in any of the earlier studies (eg. Kakugawa 2017). The authors should provide the data regarding the KO regions (fl and cre), the genotyping figures and validations for a successful KO.

As mentioned in the comment 4 of this Reviewer, *Satb1^{fl/fl}* Cd4-Cre⁺ mice have been previously utilized and uncovered deregulated T cell development and an autoimmune phenotype. In the revised manuscript these publications are included in the references as:

17. Kitagawa, Y. et al. Guidance of regulatory T cell development by *Satb1*-dependent superenhancer establishment. *Nat. Immunol.* 18, 173–183 (2017).
27. Kakugawa, K. et al. Essential roles of SATB1 in specifying T lymphocyte subsets. *Cell Rep.* 19, 1176–1188 (2017).

These studies have not provided images of animals or morphology of their organs, therefore the phenotype displayed in our revised Supplementary Figure 3a cannot be correlated to prior findings. Moreover, the study by *Kitagawa et al., 2017* was mostly focused on investigating regulatory T cells, hence there is missing characterization of the thymus or peripheral lymphoid organs. In line with *Kitagawa et al.*, we supported the inflammatory environment of *Satb1* cKO and autoimmune phenotype affecting internal organs. This and the absence of naive T cells was also in line with *Kondo et al., 2015* (although they used *Satb1^{fl/fl}* Vav-Cre⁺ animals). Additionally, the thymic deregulation with increased levels of DP and decreased levels of SP T cells was in line with *Kakugawa et al., 2017* (*Satb1^{fl/fl}* Cd4-Cre⁺ animals) as well as *Alvarez et al., 2000* (SATB1-null animals). In the revised version of the manuscript, we included citations of all the respective studies and made clear what is known and what is novel regarding the *Satb1* cKO phenotype. Nevertheless, we believe that our data nicely correlate with the published studies.

Moreover, we now provide the information regarding the strategy for the generation of the *Satb1* cKO mouse (**revised Supplementary Figure 1**).

To check the efficiency of SATB1 depletion in *Satb1* cKO thymocytes we have performed the following experiments described in **revised Supplementary Figure 2**:

- (a,b) **FACS analysis** for both WT and *Satb1* cKO thymocytes for the detection of **EYFP expression** in the cells. For these experiments we have utilized *Satb1* cKO mice that were also encompassing the EYFP gene as a transgene in the ROSA26 locus which is not expressed unless Cre-mediated excision of a *loxP*-flanked transcriptional “stop” sequence permits EYFP expression and the detection of EYFP. This does not directly answer the Reviewer’s question but at least identifies the fraction of thymocytes that express the Cre recombinase which can act upon the deletion of *Satb1*. We found that more than 96% of the total *Satb1* cKO thymocytes and more than 97% of DP thymocytes expressed the EYFP reporter, indicating the efficient expression of the Cre recombinase expressed under the *Cd4* promoter.
- (c,d) As suggested, we have performed **intracellular staining for SATB1** protein, in both WT and *Satb1* cKO thymocytes and found that 8.7-13.9% of total *Satb1* cKO thymocytes still expressed the SATB1 protein. 5.6-10.8% of the double positive (CD4⁺CD8⁺) *Satb1* cKO thymocytes still expressed the SATB1 protein.
- (e) **Immunofluorescence experiments** analyzed by confocal microscopy indicated that SATB1 is expressed in 97.11% of wild type thymocytes and 4.16% of *Satb1* cKO thymocytes, utilizing a commercially available antibody detecting all SATB1 isoforms. The relative SATB1 fluorescence signal was calculated in a quantitative manner utilizing the Volocity software from PerkinElmer.

7. Lines 130-131: The authors state “.. prevailing IL-17 cell responses and increased levels of pro-inflammatory cytokines such as IFN γ and TNF α detected in *Satb1* cKO sera...” Yet again, there is no reference to the article Yasuda, 2019 which showed Th17 mediated autoimmunity in the absence of *Satb1*. The authors should thoroughly revise the text to incorporate all the important information.

We thank the reviewer for the comment. We are indeed aware of this publication, which, in fact, concludes that SATB1 promotes pathogenesis of Th17 cells, i.e. in *Satb1* knockout, there is no Th17

mediated autoimmunity. Moreover, this article focused on the roles of SATB1 in peripheral T cells, while our manuscript mainly focuses on intra-thymic T cell development. Please also find attached a summary of important parts from Yasuda et al:

*“Collectively, these results indicate that Satb1 is dispensable for the differentiation of Th17 cells in vivo and for the induction of the Th17 subset in vitro..... Th17^{Satb1KO} mice were resistant to the development of EAE with fewer eYFP⁺ Th17 cells infiltrating the spinal cord compared with control (Il17a^{Cre} R26R^{eYFP} Satb1^{wt/wt})..... Collectively, these results suggest that Satb1 does not affect the maintenance and migratory capacity of Th17 cells to inflamed tissues, but **Satb1 expression** is increased upon IL-23 stimulation and **plays a pivotal role in the pathogenicity** of EAE and effector functions of Th17 cells **by regulation of GM-CSF production.**”*

8. The authors show in Fig 1e ‘disturbance of the islets of Langerhans’ upon Satb1 KO. Authors should show more sections and point the differences in WT vs cKO of Satb1 for better clarity. Furthermore, there is no discussion/inference from the glucose tolerance test as to why only one time point differ which also seems to recover later?

Following the suggestion of the Reviewer, we have included more sections from different animals to better indicate the disturbance of the islets of Langerhans and we added labels to make it clearer (**revised Supplementary Figure 3d**).

In the **revised Supplementary Figure 3e**, we show that glucose metabolism is impaired and we should note that even at the steady state time point before fasting, the knockout animals have significantly elevated glucose levels compared to their WT counterparts. It is noteworthy that the animals that were used for the glucose tolerance test were 85 days (± 8) of age (**Methods Lines 535542**, revised manuscript) therefore at an age prior to the onset of phenotype described in the revised Supplementary Figure 3a.

9. The authors indicate an increase in autoantibodies upon Satb1 cKO (Fig 1F). The imaging is poorly done and is unclear. This is also a crude and indirect method. Instead, the authors should use more robust and quantitative methods such as ELISA or LIA.

We would like to point out that the objective of this manuscript was to investigate promoter-enhancer looping. As indicated in comment 4 of this Reviewer, the autoimmune phenotype was also previously described utilizing *Satb1* knockout mice. This experiment was used in order to support our phenotypic characterization of the *Satb1* cKO mice that included experiments such as the histology sections of thymus (**revised Figure 1b**), spleen (**revised Figure 5d**) and pancreas (**revised Supplementary Figure 3d**).

We did try to perform the ELISA assay utilizing Mouse ANA (Anti-nuclear Antibody) ELISA Kit (MOFI01271-48) from AssayGenie but unfortunately we were not successful. Given the high number of serum samples (cohorts of different ages and sexes) we have decided not to pursue it again. Though we should indicate the fact that the presence of anti-nuclear antibodies was already demonstrated in the serum of SATB1-depleted animals in the past (<https://doi.org/10.4049/jimmunol.1501429>, **revised manuscript reference #26**: Kondo, M. et al. SATB1 plays a critical role in establishment of immune tolerance. *J. Immunol.* 196, 563–572 (2016).).

1. In line 146, the authors state “We did not identify any major changes in high-order chromatin organization....” In the related figure 2a, the authors should describe the parameters for this analysis, and multiple regions should be shown to arrive at such conclusion. Same for the TAD analysis in the supplementary information. The authors should also compare this data with HiC analysis for the same regions under CTCF null condition (the data is publicly available).

The statistical significance of our claims regarding the high-order chromatin organization is discussed in the response to comment 12 of this Reviewer. The parameters of Hi-C analysis are described in the methods section (**lines 683-697**, revised manuscript). In the revised version of the manuscript, we also added the normalization method (balanced) and resolution of the presented matrices (500 kbp). We have modified the text on the first call of the **revised Figure 2a** to include the information about resolution and normalization and we also included this information in the figure legend, as follows:

Lines 148-150: “To address this, we performed Hi-C experiments in both WT and *Satb1* cKO thymocytes (Supplementary File 4). Notably, we did not identify any major changes in high-order chromatin organization (Fig. 2a; 500 kbp resolution, balanced normalization).”

Legend Figure 2a: “Comparison of WT and *Satb1* cKO Hi-C heatmaps at 500 kbp resolution (balanced normalization) of chromosomes 9 and 11 indicates no major changes at the high order chromatin level of murine thymocytes.”

Please note that Figure 2a does not include a handpicked region but the entire chromosome. In the revised version of **Figure 2a** we show two entire chromosomes. Moreover, we re-analyzed two publicly available Hi-C datasets from untreated/WT and CTCF/RAD21 depleted cells and compared the deregulation of high-order chromatin features among all the datasets (**revised Figure 2** and **revised Supplementary Figure 4b**). We could not locate any Hi-C dataset from CTCF-depleted thymocytes. However, given the conservation of TADs, we concluded that comparison with CTCF-depleted mESCs (<https://doi.org/10.1016/j.cell.2017.05.004>) would also be valuable. Next, we utilized Hi-C datasets from *Rad21^{fl/fl} Cd4-Cre⁺* DP murine thymocytes (<https://doi.org/10.1101/gr.161620.113>). As anticipated, the knockout Hi-C datasets for RAD21 and CTCF revealed a perturbation of TADs and short- to mid-range contact frequencies (<10 Mbp), both of which were mostly unaffected in the *Satb1* cKO thymocytes (**revised Figure 2**). We show an example of a domain that is disrupted in both CTCF and RAD21 depleted cells but which is maintained in the *Satb1* cKO. The differential analysis of TADs was also extended for CTCF and RAD21 depleted cells, showing a significant difference between the proportion of unchanged TADs in *Satb1* cKO and in the other two datasets (**revised Supplementary Figure 4b**). Lastly, in *Satb1* cKO we did not detect any major changes in chromatin compartmentalization as demonstrated by new saddle plot analysis (**revised Figure 2d** and **Supplementary Figure 4c**).

Collectively these new data support our original hypothesis that SATB1 acts differently from conventional genome organizer proteins and it does not participate in high order chromatin organization.

11. Importantly, the authors used a custom antibody specific for only SATB1 long isoform for their HiChIP experiments. Since this is the central focus of their findings, it necessitates the validation of the efficacy of the antibody. As almost 90% of *Satb1* expressed in thymocytes is not the long isoform, it warrants that either the HiChIP experiments to be repeated with *Satb1* antibody or show that the effect is specific only to the *Satb1* long form.

Our second manuscript describing the long SATB1 isoform is now deposited in bioRxiv (*T. Zelenka, P. Tzerpos, G. Panagopoulos, K-C. Tsohis, D-A. Papamatheakis, V.M. Papadakis, D. Stanek, C. Spilianakis, SATB1 undergoes isoform-specific phase transitions in T cells*, doi: <https://doi.org/10.1101/2021.08.11.455932>). As already discussed in the reply to comment 4 of Reviewer #1 and comment 2 of Reviewer #2, in the bioRxiv manuscript we described the specificity of the long SATB1 isoform-specific antibody (**Figures 1e-f, Figure 2a, Figure 3e, Supplementary Figure 1a,b, Supplementary Figure 3**) and we also provided pixel-based colocalization analysis of super-resolution microscopy experiments based on the long SATB1 isoform-specific antibody and general SATB1 antibodies non-selectively targeting all SATB1 protein isoforms. Western blot experiments comparing the two antibodies have also been provided.

Additionally, we have included into our revised manuscript a comparison of the SATB1 long isoform-specific binding sites (identified in this manuscript) with two publicly available SATB1 ChIP-seq experiments (Hao *et al.*, 2015; Kitagawa *et al.*, 2017). Based on the Relative Distance metric of Bedtools (see **Supplementary Notes**), the long SATB1 isoform-specific binding sites were non-randomly spatially correlated with both publicly available SATB1 ChIP-seq datasets (which utilized antibodies targeting all SATB1 isoforms; **revised Supplementary Figure 4d**). Moreover, the genomic feature association analysis (Genome Ontology) for the three datasets of SATB1 binding sites, analyzed by the AnnotatePeak function of Homer showed highly comparable binding preference for all SATB1 antibodies (**revised Supplementary Figure 4e**).

Overall, we should note that the SATB1 HiChIP loops are long isoform-specific and these are the loops that were positively correlated with transcriptional deregulation of affected genes in *Satb1* cKO as well as with H3K27ac loop deregulation. Using an unbiased linear regression modeling approach, these long isoform-specific SATB1-mediated loops were identified as good predictors for *Satb1* cKO underexpressed genes as visualized in the **revised Supplementary Figures 6 and 7a**.

Moreover, data in our second manuscript (T. Zelenka, P. Tzerpos, G. Panagopoulos, K-C. Tsolis, D-A. Papamatheakis, V.M. Papadakis, D. Stanek, C. Spilianakis, *SATB1 undergoes isoform-specific phase transitions in T cells*, doi: <https://doi.org/10.1101/2021.08.11.455932>) strongly challenge the general view of SATB1 isoforms. While the long isoform RNA levels are 3-5-fold lower than the RNA levels of collectively all short SATB1 isoforms, both imaging and western blot experiments indicated high abundance of the long isoform at the protein level similar if not higher compared to the short isoform (please also see our initial response comment to Reviewer #3).

12. How was the strength of the interaction pair calculated (fig2b)? This should be explained more in the text to support the conclusion of the figure. It is important that the authors plot for higher resolution contacts to better conclude the reduced contact strength of *Satb1* loops.

We extended the respective method section using the following description: “*To identify differentially interacting regions between SATB1 and CTCF HiChIP experiments, we utilized raw matrices at 100 kbp and 500 kbp resolutions. Matrices were analyzed using diffHic and differentially interacting regions were determined as differential interactions with FDR≤0.05.*” (**Lines 698-700**, revised manuscript). Note that diffHic is a count-based method that utilizes the edgeR framework to identify differentially interacting regions (<https://doi.org/10.1186/s12859-015-0683-0>). Count-based methods for the identification of differentially interacting regions from Hi-C/HiChIP experiments, such as diffHiC, assume independence of each pairwise comparison. At low resolutions, this assumption is logical, as we only capture the spatial proximity between distant chromatin fibers. In the case of high-resolution contact maps, this assumption is not valid anymore. For example, if two regions interact strongly in the 3D space, the adjacent genomic loci would also be affected and thus being more proximal. Therefore, at high resolutions, methods with the independence assumption can have a high error rate (<http://www.genome.org/cgi/doi/10.1101/gr.212241.116>). For these reasons, we deemed necessary to keep the low resolutions in the revised manuscript. The results of this analysis are in line with our other findings, collectively indicating that SATB1 does not mediate high-order chromatin organization (see also response to the comment 10 of this Reviewer). This led us to shift our focus to the loop-level of chromatin organization and the involvement of SATB1 in chromatin looping is supported by a plethora of other analyses and datasets.

13. In lines 168-170, the authors state “...genes residing in both CTCF- and SATB1-mediated loops were transcriptionally insulated from their gene neighbors (Extended Data Fig. 1f) and this characteristic was not altered in the *Satb1* cKO. ...” Does this mean if the contacts by *Satb1* are lost (as described by the APA analysis), there is no change in these genes’ expression?

We have mentioned that genes inside loops were transcriptionally insulated from genes outside the loops. We did not state that there were no changes in gene expression – we actually found and claim the opposite. We have modified the text in the following way to make it more clear: “*genes residing in CTCF- and SATB1-mediated loops were transcriptionally insulated from their gene neighbors outside the loops*” (lines 190-193, revised manuscript). For more details about how this analysis was performed, please see the respective method section (lines 944-952, revised manuscript).

14. Figure 2e, The authors have used ‘Satb1 binding site’ for their nucleosome binding analysis from ATAC-seq data. This becomes dicey as Satb1 binding sequence is not perfect but dynamic. Hence, Satb1 ChIP-seq peaks are necessarily to be used instead of a singular input sequence, which is applicable to Ctf only, here. The authors should also indicate the motif for Satb1 that they have used for their analyses.

To clarify this part, we have not used a singular input sequence in this analysis, neither for CTCF nor for SATB1. To determine the SATB1 binding sites we utilized a HiChIP experiment with a large number of so-called Dangling End Pairs (~280 million reads; described in the methods and all datasets available from GEO database). Such analysis provides data comparable to a standard ChIP-seq experiment (we just did not call them peaks not to confuse readers who usually synonymize peaks with the ChIP-seq output). To demonstrate the reliability of the data, in the revised version of this manuscript we added a comparison of the long-isoform specific binding sites retrieved from our HiChIP data with the standard ChIP-seq peaks based on antibodies non-selectively targeting all SATB1 isoforms (see **Supplementary Notes** and earlier responses to comment 4 of the Reviewer #1, comment 2 of the Reviewer #2 and comments 3 and 11 of this Reviewer #3). For the nucleosome binding analysis of CTCF, we used publicly available ChIP-seq peaks as we did not have similar HiChIP data as for SATB1 with high enough number of Dangling End Pairs. We should note that we still identified the CTCF binding sites based on the HiChIP data and used it for a motif analysis. Note that the top hit was the CTCF motif, supporting the validity and reliability of our approach.

Our dataset for the long SATB1 isoform binding sites evinced closer relative distance to both public SATB1 ChIP-seq datasets (*Hao et al., 2015; Kitagawa et al., 2017*) than expected by chance. Irrespective of the differential peak size among the datasets, this method allowed us to validate a non-random overlap between the long and all SATB1 isoform datasets (**revised Supplementary Figure 4d**). Moreover, the genomic feature association analysis (Genome Ontology) for the three datasets of SATB1 binding sites, analyzed by the AnnotatePeak function of Homer showed highly comparable binding preference for all SATB1 antibodies (**revised Supplementary Figure 4e**).

Similarly, a pixel-based colocalization analysis based on super-resolution microscopy confirmed the significant correlation and overlap between the immunofluorescence staining utilizing antibodies targeting the long and all SATB1 isoforms (*T. Zelenka, P. Tzerpos, G. Panagopoulos, KC. Tsolis, D-A. Papamatheakis, V.M. Papadakis, D. Stanek, C. Spilianakis, SATB1 undergoes isoform-specific phase transitions in T cells*, doi: <https://doi.org/10.1101/2021.08.11.455932>) (**Figure 2a, Supplementary Figure 1b, Supplementary Figure 3a**).

Collectively, all these data support the overall quality of our dataset. Ultimately, the validity of our dataset was also confirmed by the correlation between the SATB1 peaks and peak-based chromatin loops and their functional deregulation in SATB1-depleted cells.

15. Figure 3a. The authors show immunofluorescence of HP1 and PolII to demonstrate ‘more chromatin compactness’ as stated in the text. Firstly, the images are of very poor resolution with a lot of background noise. More importantly, relative increase in HP1a does not necessarily demonstrate heterochromatinization, since PolII remains the same. The authors need to show nuclear stainings alongside these stains. The authors should also perform quantitative Western blotting to ascertain these observations.

We do agree with the comment that was also raised by Reviewer #1 in comment 6. However, this was one piece of information supporting the other approaches. We did show in our manuscript the decrease in chromatin accessibility as supported by ATAC-seq experiments (**revised Figure 4a/b/c, Supplementary Figure 5c**) and we could also see the compaction from multiple images from electron microscopy (**revised Supplementary Figure 5a**). This is also in line with the decreased MNase digestion of the *Satb1* cKO chromatin (**revised Supplementary Figure 5b**), the increased volume of heterochromatin regions as deduced by quantitative measurements of DAPI staining in *Satb1* cKO nuclei versus WT thymocyte nuclei (**revised Supplementary Figure 5d**, left graph) and the reduced volume of pRNA Pol II in the *Satb1* cKO nuclei versus WT thymocyte nuclei (**revised Supplementary Figure 5d**, right graph). All this is also supported by the higher number of genes with decreased expression seen from RNA-seq (**revised Supplementary File 2**). In the revised version of the manuscript, we omitted the HP1a staining as it did not provide additional value to our conclusions.

1. The immunostaining the authors have used total thymocytes from CD4+ cKO mouse. Therefore, it is unclear whether the cells shown/quantitated are of which thymic subset. A straightforward way would be to use sorted thymocytes.

Unfortunately, it is not clear to us to which immunostaining the Reviewer refers to, unless it is the one presented in Figure 3a (related to comment 15). However, it should be noted that the DP T cells represent ~90% of the thymocyte population used in this study. Additionally, we have used advanced computational approaches to specifically demonstrate that the chromatin organization role of SATB1 takes place in DP T cells. (Please also take under consideration our response to the previous comment #15).

2. Line 195: The authors wrote “This observation further reinforced our hypothesis that SATB1 acts at a finer-scale level of genome organization” the conclusions need to be re-written according to the results in a specific manner.

This entire paragraph was modified to comply with **comment 10** of this Reviewer, therefore this sentence was now removed in the revised version of the manuscript.

3. Lines 213-214: The authors mention “especially evident at the transcription start site of genes (TSS), suggesting a direct role of SATB1 in gene transcription regulation.” Which genes are the used for overlapping ATAC peaks; all or up or down upon *Satb1* cKO? The accessibility (shown in fig 3 b and 1k) should be validated at the transcriptome level via qRT-PCRs to demonstrate the repressive role of *Satb1* for the genes as mentioned in line 220.

The original Figures 3b-c (**revised Figures 4a-b**) referred to SATB1 binding sites and not genes. Figure 3d (**revised Figure 4c**) refers to all the genes irrespective of their transcriptional deregulation in *Satb1* cKO and we now specified it in the text (**lines 227-231**, revised manuscript). It is not clear how we could validate decreased chromatin accessibility by qRT-PCR. We stated in the manuscript that ~5% of SATB1 binding sites evinced decreased chromatin accessibility (**lines 231-236**, revised manuscript). Since we refer to SATB1 binding sites but not genes, no expression changes can be validated.

4. Figure 4a: -The authors show a correlation of RNA seq to over-vs-under-interacting K27 loops in the KO condition. Given its low correlation (although positive), it seems that there is only a subset of genes expression that correspond to change in the looping. The authors should discuss this.

Further, it would be more informative to plot upgenes vs over-interacting and downgenes vs under-interacting.

This is a valid point and similar comments have been made by Reviewer #2 (comments 1 and 5). We should note that in the original analysis, all the genes were used to generate the correlation plot (Spearman's $\rho = 0.26$). Considering this fact, the correlation was relatively high already. In the revised version of the manuscript, we used only the significantly deregulated genes, which display much stronger correlation with SATB1-mediated H3K27ac regulatory loops (Spearman's $\rho = 0.62$), collectively supporting our conclusion about the role of SATB1 in mediating regulatory chromatin loops (**revised Supplementary Figure 7d**). An indication of causality for this effect is demonstrated by the **revised Figure 4f**, where we specifically compared the differential *Satb1* cKO under- and over-interacting H3K27ac loops that are or are not mediated by SATB1/CTCF, and the expression changes of genes affected by these loops. These images clearly indicate that genes at the anchors of SATB1-mediated H3K27ac loops that were diminished in *Satb1* cKO, were significantly underexpressed, unlike the genes in either the CTCF-mediated loops or any factor-mediated overinteracting H3K27ac loops.

Additionally, to unambiguously indicate that our observed regulatory loops are specific to DP T cells, utilizing the ImmGen RNA-seq sorted T cell datasets, we identified signature genes for DN, DP and SP T cells (**Supplementary Figure 4a**). Utilizing these sets of signature genes, in the revised version of the manuscript we clearly show that the signature genes for DP cells are the most transcriptionally deregulated in *Satb1* cKO (**Figure 1j,k**) and they are also the most affected by SATB1-mediated regulatory looping (**revised Figure 4g, Supplementary Figure 7c,d**).

16. For Figure 4b and 4c, the authors show expression change for genes at the loops of *Satb1*, CTCF and the rest of the under-interacting and over-interacting loops. For the conclusion (lines 261263) “...Moreover, the expression of genes located at anchors of SATB1-mediated loops was decreased more dramatically than genes located at CTCF loops supporting direct involvement of SATB1 in the regulatory chromatin loops”, it is essentially required to include CTCF KO thymic RNAseq data or CTCF KO HiChIP data of K27ac to infer this correctly.

In the revised version of the manuscript we modified this part as follows: “Moreover, the expression of genes located at anchors of SATB1-mediated loops was dramatically decreased supporting the direct involvement of SATB1 in the regulatory chromatin loops (Fig. 4f).” (**lines 274-276**, revised manuscript).

17. In Figure 4F, what ‘disturbance’ are the authors stating here? The differences should be labeled and clearly shown. Cell quantitation from germinal centers from both WT and cKO mice is needed to support the data. Same statements were also used for Fig 3F. The authors need to revise their results and conclusions in the text thoroughly.

The Reviewer has a valid point. We have revised our manuscript and we have now provided new spleen tissue sections (**revised Figure 5d**). Accordingly, the legend of the revised Figure 5d (**lines 1421-1432**) now reads: “H&E staining of mouse spleen sections. In WT sections, Red Pulp (RP), lymphoid Nodules of the white pulp (N), the Hilus (H) and Trabecular Vein (TV) are labeled. Arteries (black arrows) are present in each nodule. The nodule structure is clear with extensive periarteriolar lymphocyte sheath (PALS), rich in round, dark stained T lymphocytes surrounding the arterioles (WT2). The distinct marginal zone surrounding the marginal sinus (MZ and magenta arrow, respectively) is shown. Dotted lines mark lighter stained B-lymphocyte rich follicle regions within the nodule. In *Satb1* cKO (SKO), there is disturbed spleen structure with few apparent small sized lymphoid nodules with periarteriolar region depleted of T lymphocytes. The follicular region has

accumulated large phagocytic cells and displays many foci of phagocytosis of apoptotic cells (green arrows). The red pulp contained higher numbers of haemopoietic cell clusters and megakaryocytes (yellow arrows) suggesting increased haemopoietic activity. Scale bar WT1 & SKO1 100 μ m and WT2 & SKO2 50 μ m.”

Therefore, we have modified the text in the revised version of the manuscript to adhere with the results and it now reads: “*Satb1 cKO animals had disturbed spleen structure, containing fewer lymphoid nodules harboring impaired B cell follicular regions with accumulated phagocytic cells and increased apoptosis (Fig. 5d).*” (Lines 341-342, revised manuscript).

It is not clear what should be improved about the original Fig. 3F since it is unrelated to Figure 4F.

18. For Fig 4e, It is imperative to validate some of these contacts (like Bcl6) via 3C-qPCRs on WT vs SATB1 cKO conditions. BCL6 expression should be shown at the protein level.

We have performed western blot analysis for BCL6 in WT and *Satb1* cKO thymocyte protein extracts supporting the reduced expression of BCL6 protein in the knockout (**revised Figure 5c**). We have also performed immunofluorescence experiments and quantitative signal analysis in confocal microscopy images for BCL6, in WT and *Satb1* cKO thymocytes supporting the great reduction of BCL6 expression in the knockout cells (**revised Supplementary Figure 9c**). We have also performed 3C analysis in WT and *Satb1* cKO thymocytes to check the interaction between the *Bcl6* gene promoter and its two enhancers and indeed we have confirmed the Hi-C and HiChIP data (**revised Supplementary Figure 9a, b**).

19. Figure 6D. The panels shown for WT and cKO do not show any difference to support the conclusion “we also identified the disrupted thymic structure and impaired cell-to-cell communication”.

We have revised our manuscript and we have now provided new thymus tissue sections (**revised Figure 1b**). Accordingly, the legend of the revised Figure 1b (**lines 1342-1344**) now reads: “*Methylene blue stained thymus sections from WT and Satb1 cKO mice displaying a disrupted thymic environment in the Satb1 cKO mouse (C: cortex, M: medulla, CM: corticomedullary region; scale bar 50 μ m).*”

Therefore, we have modified the text in the revised version of the manuscript to adhere with the results and it now reads: “*The thymi of Satb1 cKO animals had impaired structural integrity (Fig. 1b)*”. (Lines 108-109, revised manuscript)

20. In the *Satb1* cKO thymus, the H&E staining shows no difference. Methylene Blue staining only shows the relatively low number of cells and the TEM images are set to different contrasts, hence are not reliable measure of differences. The authors need to include higher quality and properly labeled pictographs which show the claimed difference unambiguously.

We fully agree with the Reviewer. We have revised our manuscript and we have now provided new thymus tissue sections (**revised Figure 1b**) and Transmission Electron Microscopy images (**revised Figure 1c** and **Supplementary Figure 5a**).

21. Lines 378-382: The authors state “The presence of two SATB1 protein isoforms was recently described by our group.....We showed that the long SATB1 protein isoform had a higher propensity to undergo phase transitions compared to the short isoform”. What is the significance

here? It is not appropriate to cite papers that are submitted (not to any preprint server) and the data can't be verified.

Reviewer #3 is absolutely right regarding the availability of our manuscript describing a long SATB1 isoform. Though, we should note that we had informed the Editor regarding the second manuscript describing the long SATB1 isoform and we would be more than happy to provide the manuscript when needed. It is currently available in **bioRxiv** (*T. Zelenka, P. Tzerpos, G. Panagopoulos, K-C. Tsohis, D-A. Papamatheakis, V.M. Papadakis, D. Stanek, C. Spilianakis, SATB1 undergoes isoform-specific phase transitions in T cells*, doi: <https://doi.org/10.1101/2021.08.11.455932>). The significance is clear, given the current model of transcriptional regulation which might be directly linked to the gene expression regulation by regulatory SATB1-mediated chromatin loops.

Once more we would like to thank the Reviewers for their helpful suggestions. We hope that we have addressed all the points in a satisfactory way and we hope that they will now find our manuscript suitable for publication.

Sincerely,

Charalampos G. Spilianakis, Ph.D

REVIEWER COMMENTS

Reviewer #1 (Remarks to the Author):

Authors have made a large effort trying to improve the manuscript. Although authors have responded to many questions raised by reviewers, unfortunately, from the viewpoint of this reviewer, multiple questions and ambiguity still remain. Most importantly, the results as they are shown do not necessarily demonstrate that SATB1 dependent genome organization (even including chromatin looping connecting enhancers and promoters) is directly correlated to the transcription program in the T cell lineage.

While this paper was under revision, an important manuscript that is highly relevant to the Zelenka et al paper has been uploaded in bioRxiv (by Kohwi et al.). This can be found using the two key words "SATB1 bioRxiv" on website. This bioRxiv paper is very important for the authors to read and study in depth in order to re-interpret their results described in the current Zelenka et al paper. Although the bioRxiv paper has not been published yet, there are couple of important points that are worth attention.

1. As questioned by REV 1 as well as REV3, there has been a puzzle concerning SATB1 binding sites in the genome. Kohwi et al (bioRxiv) demonstrated that SATB1 binds in vivo to DNA segments called "base unpairing regions or BURs", which have a strong unwinding property, confirming the result originally described in vitro. Based on the bioRxiv paper, SATB1 directly interacts with BURs, and these direct interactions can be detected only by the modified ChIP-seq protocol. This conclusion is worth believing because the BURs detected in vivo precisely mapped to the genomic sequences that SATB1 binds in vitro. Therefore, the binding target sequences are now clarified. The references should be selectively chosen to avoid any further confusion in the field.
2. Based on the bioRxiv paper, an important point to consider for the Zelenka et al paper is that the normal ChIP-seq protocol (and this will extend to SATB1 HiChIP) does not identify direct SATB1 binding sites (or direct SATB1-mediated chromatin looping). Also, another point is that CTCF and SATB1 are totally independent in terms of genome organization. Therefore, the terminology used in Zelenka et al, such as SATB1-regulatory looping or SATB1-mediated looping is not necessarily scientifically accurate. SATB1-dependent XYZ should be fine. As Zelenka et al also indicated, in agreement with the bioRxiv paper, these SATB1-binding sites detected by normal protocols show major overlap with CTCF sites and H3K27ac. These could be indirect or could be even artifacts, but at least for some of the gene loci examined in the bioRxiv paper, ChIP-seq peaks at these loci were found to be still SATB1 dependent.
3. Based on the point 2 above, Zelenka et al paper needs to re-evaluate the entire manuscript carefully (including ABSTRACT) and extract the most important points that can be concluded and avoid over-interpretation. Also, in Discussion, point 2 should be discussed. In particular, one has to carefully revise the wording in line 201-203, page 9 and page 12, line 260-263. It describes a notion that CTCF forms a basal higher-order chromatin structure (TADs), and SATB1 forms a refined layer of organization (promoter-enhancer) within CTCF TADs. This conclusion can cause confusion because this manuscript is not dealing with SATB1's directly regulated genome structure and SATB1-directly regulated looping is likely to be independent of CTCF-TADs. On the other hand, "H3K27ac loops (interaction frequencies) that are dependent on SATB1" can, in principle, provide informative results as these are derived independently of the SATB1 ChIP-seq or SATB1 HiChIP-seq protocol.
4. In response to my first comment made in the previous review, it was bit surprising that authors replied "in most cases, the H3K27ac loops could be detected in both WT and Satb1 cKO at the same genomic positions". With the use of differential H3K27ac loop analysis, authors found a positive correlation between changes in H2K27ac interaction frequency and gene expression was found (Figure 4h). A table would be useful to list those genes with a threshold value used to define the differential loops and exact fold change in expression. Furthermore, for most of genes shown their expression is not greatly changed based on the RNAseq data (probably 50% reduction at the maximum). There are many DP genes in the RNAseq data showing more dramatic changes in gene expression depending on

SATB1. It seems that none of these genes were found in this group showing the positive correlation. Consistently, the magnitude of change in expression detected in Figure 4f is also small (reduction in expression of genes at anchors of SATB1 loops appears not dramatic). Viewing the entire manuscript, only *Bcl6* shown in Fig. 5a, *Ccr7* in Supplementary Fig. 12 and *Rag 1* in Figure 6d are genes that show major effect on gene expression in *Satb1* cKO. This needs to be explained in the context of all SATB1-dependent genes (Elaborate on Supplementary Figure 7a with fold change). Also, for Fig. 4g, T cell subset signature genes at SATB1 loop anchors are enriched in DP compared to DN or SP and this is also the case for H3K27ac loops (both underinteraction or overinteraction) in Supp Fig. 7c. From this set of data, it would be crucial to list up gene names that are strictly SATB1 dependent for looping and show the changes in gene expression depending on SATB1 for each of these genes.

5. Although it is stated in line 226, page 10, a visible drop in chromatin accessibility in the *Satb1* cKO (Fig. 4b), ATAC-seq data shown elsewhere for individual genes (e.g. *Bcl6*, Fig.6) show very similar profiles and there is no indication of any difference in SATB1-dependent accessibility. Explain.

6. Since this manuscript aims to demonstrate that SATB1-dependent chromatin organization is correlated (or linked) to gene expression, it is essential to show strictly SATB1-dependent looping (identified by comparing WT H3K27ac versus SKO H3K27ac) is correlated to gene expression. Similar to the SATB1-dependent looping data shown in Figure 6c for the TCR locus organization for rearrangement would be nice to show for SATB1-dependent genes in DP cells. At this time, even with *Bcl6* gene with the addition of 3C analysis, it is still not fully convincing (too large error bars for *Enh2*, and Hi-C is not quite convincing).

7. Revisit the model shown in Supplementary Fig .10 a and think if this really holds true. This reviewer thinks this model is problematic, as it still tries to relate to CTCF (in light of point 2).

8. It is a concern that cKO thymocytes exhibit some SATB1 protein expression. If this is indeed the case, experiments that involve amplification can suffer from contamination of results from WT thymocytes. Western blot should be negative. It is best to contact the two labs which used *Satb1* cKO mice and see if they saw this leakage. The cKO mice are not expected to have target protein expression in cre-expressing cells.

9. Because this research was done using a particular isoform of SATB1, it has to be stated clearly that these results were derived for the specific isoform of SATB1.

Minor points

1. Fig. 1c is not very clear to show disrupted intercellular contact.

Overall comment:

The author responded that Kitagawa et al paper dealt with Treg cells while this paper (Zelenka et al) is mostly on DP cells. Therefore, SATB1 binding sites at the *Foxp3* site is irrelevant to their paper. Their response is not necessarily correct. In Kitagawa et al paper, SATB1 is bound to the super-enhancer at the *Foxp3* site already at the DP stage, before the super-enhancer at the *Foxp3* is established at Treg cells, *Foxp3* expression and greatly increased ATAC-seq signals. Similar to *Foxp3* in Treg cells, it is expected that many DP signature genes are activated in DP cells in a SATB1-dependent manner, showing similar response as *Foxp3*. So, it was expected that those genes would show SATB1-dependent H3K27ac modification, and that looping connecting H3K27ac sites would be greatly altered by SATB1 depletion. Apparently, for some reason, Zelenka et al paper did not see such dramatic results in any genes shown in DP cells (change in H3K27ac, and ATAC-seq and gene expression, as seen for *Foxp3* in Treg cells). This is a puzzle and remains to be explained. Therefore, it is not really clear to this reviewer whether SATB1 dependent genome structure (looping, accessibility, histone modification) indeed reflects SATB1-dependent gene expression in a strict sense.

Reviewer #2 (Remarks to the Author):

I thank the authors for their careful attention to the comments of all the reviewers and for the extensive revisions that they have made. They have thoughtfully addressed all my previous comments, and the paper is considerably stronger in my opinion.

My remaining comments are about figure clarity. These kinds of HiC analytical tools are often difficult to appreciate visually without more guidance.

1. In the legend to Fig. 4e, or at least in the similar legend to Supplementary Fig. 4g, please spell out what each of the four quadrants mean and what the ADA score is actually measuring. Once that is clear, the L and R labels are sufficient, but first the reader has to understand what the heat map measures and what the difference is between upper left and upper right, for example.

2. In Fig. 4h, the light transparent blue circles are very hard to distinguish from the light transparent blue triangles where the two sets of symbols overlap. Could the authors make them different colors, and also could they provide different regression lines for the relationships in the SATB1 loops and the relationships outside the SATB1 loops? The same change would be useful in Fig. 6b.

3. Also in Supplementary Fig. 4, there is not enough information on the axes of the plots in panels b and d for the reader to get the point. Readers should not have to look up other references to know what the category labels on the figure axis mean. Could "Non-differential", "merge", "split", "shifted", and "complex" be defined briefly in the legend to panel b? And please explain what the "relative distance" is relative to, in panel d. The p values are clear, but it would be good also to understand what is being shown.

4. In Supplementary Fig. 6, the labels on panels a, b, c, and d are not visible at all. Please add labels with legible (≥ 8 pt) type to these screenshots.

5. In Supplementary Fig. 8 and in the top panel of Supplementary Fig. 12 (Cd28), it is hard to see a convincing difference in RNA levels just from the tracks. The legends are very minimal. If the point is to demonstrate a change, could the fpkm (rpkm, tpkm) values be indicated alongside these RNA tracks?

Reviewer #3 (Remarks to the Author):

In the revised version of the manuscript, that authors have addressed many of my queries and incorporated a few suggestions. With additions and/or modifications to the existing data presented and its associated text based on mine and other reviewers' comments, the manuscript has been reasonably improved. However, it requires addressal of some key aspects related to the experiments, analyses and the conclusions drawn thereof, for it to be recommended for publication. As such, the authors do not provide unequivocal data to support their title either. The specific comments are listed below:

1. In line 66 of the revised manuscript, the authors claim to identify SATB1 enrichment at the promoter/enhancer, but have not cited the first and defining reports in this context (such as Kitagawa 2017) here but only in generic earlier context (line 56). They should correctly introduce the context of these prior reports.

2. The authors based on previous comments modified text in context of Fig 1a toward identification of Satb1 at SE (using K27ac loops). Since the use of HiChIP datasets was used in region identification and not looping per se, it becomes quintessential to mention the approach and differences from Kitagawa et al, in which they described enrichment of Satb1 at SEs and TEs using K27ac ChIP-seq, more clearly.

3. In response to comment #3, the authors performed a reldist analysis with 'all-iso' available datasets (Supp. 4d). Although this shows some level of qualitative similarity, it would be more robust to show a quantified Statistic such as jaccard. Additionally, the authors should also correlate these two datasets as a positive control. Further, we understand that the trend remains the same, but to answer our question, the authors should show an absolute overlap (percentage) between the features (Supp. 4e).

4. In line 109, which intracellular contacts are the authors referring to? Its not shown or indicated in

the figure referred here (1c). The authors should show clearer images along with indicators for the phenotype mentioned inside the figure.

5. In figure 1d (top panel); Since the authors generated *Satb1* fl/fl mice and claim that the change thymic cell numbers are in coherence with previous studies, they should reconcile their findings with the Kakugawa et al. (2017) paper that showed a dramatic decrease in SP cell numbers (~50%).

6. Whereas the figure presented in the manuscript using flow cytometry (1d) shows only minimal variation from the WT controls. Furthermore, the authors should cite the Alvarez et al., 2000 study because of more similarity of KO conditions. To further characterize their cKO model, the authors should also show immature and mature thymic populations using the same gating.

7. In line 119, the authors refer to Kondo et al., 2016 for reconciling their peripheral phenotype. However, the thymic data presented here doesn't correlate with this (in the context of DP change). The authors should mention and discuss this discrepancy. It is imperative that the authors should also show the populations of LN for the same gatings.

8. Following our earlier comment, authors have added immunostaining data for *Satb1* (all iso) in their KO conditions. Since most of the following work and the premise of it is set around the long isoform, they should also demonstrate the percentages of cells specifically expressing this isoform.

9. The authors have added clearer and much better depicted images of pancreas in the revised manuscript. Although in line 126, and Supp. 3e, the authors mentioned impaired glucose tolerance. It starts and ends same through the time-course with broader variation. In response to our comment #8 they wrote 'at the steady state time point before fasting, the knockout animals have significantly elevated glucose levels compared to their WT' which doesn't reflect in the figure.

10. In their response to our earlier comment #7, the authors mention 'Moreover, this article focused on the roles of SATB1 in peripheral T cells, while our manuscript mainly focuses on intra-thymic T cell development.' However in actual sense, the authors move (and cite) to and fro between thymic and peripheral references according to what suits the specific part of their data. In this particular case, since they talk about blood serum levels of cytokines, it would rise from peripheral mature cells. Having said that, the authors state one of the conclusions from Yasuda 2019 about the dispensability of *Satb1* for Th17 response. In such instance, the authors should state what do they infer from their data and how it reconciles with the previous study, citing the same.

11. We agree with the authors that maximum enrichment is of DP genes. But given our and reviewer #1's concern regarding the use of total cells, and the contention by the authors that it is 90% DP, the enrichment for SPs is still quite high (4 fold compared to 7 fold for DP). This also reflects in the subsequent figure 1k. The authors should critically revisit their selections for data/signature for downstream analysis and revise these figures.

12. As a part of response to our comment #11, the authors wrote 'While the long isoform RNA levels are 3-5-fold lower than the RNA levels of collectively all short SATB1 isoforms, both imaging and western blot experiments indicated high abundance'. While both imaging and WB are at best semi-quantitative, the reverse correlation of transcript and protein could easily be attributed to the efficacy of the lab-made antibody (especially when compared directly to a commercial one). It is necessary to show using more quantifiable methods, at least some validation in the supplementary data of this study since the other manuscript in reference is itself probably under review.

13. In line 153 and related figures 2b,c,d, the authors have compared *Satb1* KO dataset they generated with CTCF and RAD21 KO datasets from previous literature. Firstly, the enzymes used for digestion in those data are different from the ones used in the new data (eg HindIII for CTCF KO and DpnII for current dataset). It is widely known that 6 base cutters have very different biases than a 4 base cutter. Therefore, the authors should expound on the caveats of such comparison. Additionally, they can further normalize for the bias and plot the data again to increase the robustness of their analysis.

14. Importantly, reasoning by the authors in their rebuttal for use of mESC dataset about the conservation of TADs (so far apart in cell stages) is not acceptable, since their own premise is changes in 3D loops in one single cell type.

15. Line 187, the authors mention CTCF RNA levels are unperturbed, but do not refer to any figure, and/or dataset.

16. In line 198, the authors did not cite the original references attributing SATB1 mediated T cell

loopscape (cited in reviewer response but not the manuscript).

17. With respect to Supp. Fig 5 a,b and d, The authors claim that there is compaction and heterochromatinization. As with HP1 figure in the older version, S5a is also unclear. Instead of nuclear compaction, as posited by the authors, there seems to be that the whole cell/section is shrunk longitudinally. Furthermore, the MNase (5b) treatment seems to be almost same, with no significance in quantitation. For 5d, at least visibly it appears to have exactly the same fluorescence, which according to the quantitation is nearly halved. The authors should provide better representative images.

18. In response to our comment #18, the authors wrote 'It is not clear how we could validate decreased chromatin accessibility by qRT-PCR'. The authors should note that this can be easily accomplished by designing primers at/around the binding site enriched in their analysis for PCR. It is widely used to show accessibility changes just like 3C-qPCR.

19. In response to our comment #19, the authors state the usage of ImmGen data to pick out signature genes for downstream usage in Figure 1j,k. Whereas they posit in the main text that they did analysis on publicly available data for manually picking up gene clusters for the same. This should be clarified and appropriate sources cited.

20. With respect to Supp. 7b, the authors should also show what is the correlation of DP signature genes enriched by Satb1 overlap with those thymic enhancers.

21. In many places of the text, as well as Supp. Fig 7, the authors emphasise the DP restricted role of Satb1. But it is of most importance to realize in the text that the authors have used total thymocytes and did not directly compare Satb1 enrichment in different cell types. Logically, it would definitely lead to DP signature. Unless single cell or single population based analyses are done, it would be inappropriate to mention selective differential role in cell types (line 282).

22. We appreciate that the authors have performed 3C experiment for validation. However, they have used a different enzyme here for digestion compared to their HiC protocol. Why have they not used DpnII? Furthermore, its only mildly decreasing with no significance. Using a different enzyme might explain the same. At least regions with significant enrichment should be shown.

23. The new spleen sections are much clearer and explain the phenotype much better. In contrast, the Western blots (e.g. Bcl6) show a lot of pixilation hence better WB images should be shown. For Rag proteins (especially Rag2), the WB visibly does not correlate the transcript data. There is no change in Rag2, whereas Actin is halved. It is therefore required that the authors repeat the WB for better clarification.

24. It is also important to validate the expression of these select targets via qRT-PCR in KO condition.

Response to Reviewers' Comments

Nature Communications manuscript NCOMMS-21-25391A

"The 3D enhancer network of the developing T cell genome is controlled by SATB1"

We have now completed the second revision process of our manuscript and we submit the revised version for your consideration. Once more, we would like to thank the Reviewers for their time in reviewing our manuscript but most importantly for their constructive comments that helped us improve the quality of our work. We hope that our responses, based on the experiments we have now completed, will adequately address the remaining concerns.

Please find below a point-by-point response to the Reviewers' comments. (Original comments are in **black** and responses are in **blue**).

Reviewer #1

Authors have made a large effort trying to improve the manuscript. Although authors have responded to many questions raised by reviewers, unfortunately, from the viewpoint of this reviewer, multiple questions and ambiguity still remain. Most importantly, the results as they are shown do not necessarily demonstrate that SATB1 dependent genome organization (even including chromatin looping connecting enhancers and promoters) is directly correlated to the transcription program in the T cell lineage.

While this paper was under revision, an important manuscript that is highly relevant to the Zelenka et al paper has been uploaded in bioRxiv (by Kohwi et al.). This can be found using the two key words "SATB1 bioRxiv" on website. This bioRxiv paper is very important for the authors to read and study in depth in order to re-interpret their results described in the current Zelenka et al paper. Although the bioRxiv paper has not been published yet, there are couple of important points that are worth attention.

We do thank the Reviewer for the helpful suggestion of reading the manuscript submitted in bioRxiv (Yoshinori Kohwi, Mari Grange, Hunter W. Richards, Ya-Chen Liang, Cheng-Ming Chuong, Yohko Kitagawa, Shimon Sakaguchi, Vladimir A. Botchkarev, Ichiro Taniguchi, and Terumi Kohwi-Shigematsu. *Deeply hidden genome organization directly mediated by SATB1*, <https://doi.org/10.1101/2021.12.19.473323>). We are indeed aware of this manuscript posted in bioRxiv on December 21, 2021.

1. As questioned by REV 1 as well as REV3, there has been a puzzle concerning SATB1 binding sites in the genome. Kohwi et al (bioRxiv) demonstrated that SATB1 binds in vivo to DNA segments called "base unpairing regions or BURs", which have a strong unwinding property,

confirming the result originally described in vitro. Based on the bioRxiv paper, SATB1 directly interacts with BURs, and these direct interactions can be detected only by the modified ChIP-seq protocol. This conclusion is worth believing because the BURs detected in vivo precisely mapped to the genomic sequences that SATB1 binds in vitro. Therefore, the binding target sequences are now clarified. The references should be selectively chosen to avoid any further confusion in the field.

It is indeed true that SATB1 binding sites in the genome are a puzzle for the research community since a clear consensus has not been suggested based on the studies that employed ChIP-seq analyses for the SATB1 protein. Although, we highly appreciate the work done by Kohwi et al., we should note that this study is still not peer-reviewed and it was only posted on bioRxiv at the late stage of our revisions. Given i. the harsh urea-based conditions used by Kohwi et al., ii. the lack of urea-based ChIP-seq for a transcriptional activator as a control, the early nature of the work and iv. The fact that all SATB1 ChIPseq experiments so far are based upon the standard protocol, the notion that the binding target sequences are clarified is an over-statement.

2. Based on the bioRxiv paper, an important point to consider for the Zelenka et al paper is that the normal ChIP-seq protocol (and this will extend to SATB1 HiChIP) does not identify direct SATB1 binding sites (or direct SATB1-mediated chromatin looping). Also, another point is that CTCF and SATB1 are totally independent in terms of genome organization. Therefore, the terminology used in Zelenka et al, such as SATB1-regulatory looping or SATB1-mediated looping is not necessarily scientifically accurate. SATB1-dependent XYZ should be fine. As Zelenka et al also indicated, in agreement with the bioRxiv paper, these SATB1-binding sites detected by normal protocols show major overlap with CTCF sites and H3K27ac. These could be indirect or could be even artifacts, but at least for some of the gene loci examined in the bioRxiv paper, ChIP-seq peaks at these loci were found to be still SATB1 dependent.

We believe that the term “SATB1-mediated” does not necessarily imply direct interactions and it can still be interpreted under the light of indirect interactions. Nevertheless, following this Reviewer’s suggestion, we have changed the term “SATB1-mediated” to “SATB1-dependent” as it does not counteract our conclusions and is general enough to be correct under any circumstances.

Regarding the reviewer’s comment on complete independence of CTCF and SATB1, we should note that we based our conclusions on the intersection of SATB1 HiChIP data with Hi-C, H3K27ac HiChIP and other datasets which are available for WT and *Satb1* cKO and which thus allowed us to detect and investigate the implications of SATB1-depletion on 3D nuclear architecture and looping. It is hard to justify how SATB1-dependent loops are correlated with a decrease in expression levels in the cKO cells assuming our peaks are “artifacts”. **Supplementary Figure 7a** shows in an unbiased fashion how genes found in SATB1 loop anchors that are connected to an enhancer element tend to be underexpressed upon depletion of SATB1. Moreover, the reviewer is kindly directed to **Fig. 4f**, where it is clearly shown that under-interacting H3K27ac loops that were also SATB1 loops in WT cells are associated with a larger decrease in expression levels. We also show in the same figure, that this effect is not random and more pronounced for SATB1 dependent loops than CTCF dependent loops. We respectfully ask the reviewer to consider these findings, which, we believe, add validity to our SATB1 HiChIP.

Also please note that SATB1 ChIPseq experiments have been published in the following papers, where only the standard ChIPseq protocol has been utilized:

- Kiyokazu Kakugawa, Satoshi Kojo, Hirokazu Tanaka, Wooseok Seo, Takaho A Endo, Yohko Kitagawa, Sawako Muroi, Mari Tenno, Nighat Yasmin, Yoshinori Kohwi, Shimon Sakaguchi, Terumi Kowhi-Shigematsu, Ichiro Taniuchi (2017). Essential Roles of SATB1 in Specifying T Lymphocyte Subsets. *Cell Rep.* 2017, 19(6):1176-1188.
doi: 10.1016/j.celrep.2017.04.038.
- A standard ChIPseq protocol was used for SATB1 protein.

- Kitagawa, Y., Ohkura, N., Kidani, Y., Vandenbon, A., Hirota, K., Kawakami, R., Yasuda, K., Motooka, D., Nakamura, S., Kondo, M., Taniuchi I, Kohwi-Shigematsu T, Sakaguchi S. (2017). Guidance of regulatory T cell development by *Satb1*-dependent super-enhancer establishment. *Nat. Immunol.* 18, 173–183.
doi: 10.1038/ni.3646
- A standard ChIPseq protocol was used for SATB1 protein.
- Feng D, Chen Y, Dai R, Bian S, Xue W, Zhu Y, Li Z, Yang Y, Zhang Y, Zhang J, Bai J, Qin L, Kohwi Y, Shi W, Kohwi-Shigematsu T, Liao S, Hao B (2021) Chromatin organizer SATB1 controls the cell identity of CD4⁺ CD8⁺ double-positive thymocytes by compacting super-enhancers. Preprint from *Research Square*, 15 Nov 2021
doi: 10.21203/rs.3.rs-1069634/v1
- A standard ChIPseq protocol was used for SATB1 protein (colocalization of SATB1 peaks with H3K27ac peaks was presented).
- Bingtao Hao, Abani Kanta Naik, Akiko Watanabe, Hirokazu Tanaka, Liang Chen, Hunter W. Richards, Motonari Kondo, Ichiro Taniuchi, Yoshinori Kohwi, Terumi Kohwi-Shigematsu, and Michael S. Krangel (2015) An anti-silencer– and SATB1-dependent chromatin hub regulates Rag1 and Rag2 gene expression during thymocyte development. *J Exp Med* 212(5):809-24.
doi: 10.1084/jem.20142207
- A standard ChIPseq protocol was used for SATB1 protein.
- Rajarshi P Ghosh, Quanming Shi, Linfeng Yang, Michael P Reddick, Tatiana Nikitina, Victor B Zhurkin, Polly Fordyce, Timothy J Stasevich, Howard Y Chang, William J Greenleaf, Jan T Liphardt (2019) *Satb1* integrates DNA binding site geometry and torsional stress to differentially target nucleosome-dense regions. *Nat Commun* 10(1):3221.
doi: 10.1038/s41467-019-11118-8.
- A standard ChIPseq protocol was used for SATB1 protein.

3. Based on the point 2 above, Zelenka et al paper needs to re-evaluate the entire manuscript carefully (including ABSTRACT) and extract the most important points that can be concluded and avoid over-interpretation. Also, in Discussion, point 2 should be discussed. In particular, one has to carefully revise the wording in line 201-203, page 9 and page 12, line 260-263. It describes a notion that CTCF forms a basal higher-order chromatin structure (TADs), and SATB1 forms a refined layer of organization (promoter-enhancer) within CTCF TADs. This conclusion can cause confusion because this manuscript is not dealing with SATB1's directly regulated genome structure and SATB1-directly regulated looping is likely to be independent of CTCF-TADs. On the other hand, "H3K27ac loops (interaction frequencies) that are dependent on SATB1" can, in principle, provide informative results as these are derived independently of the SATB1 ChIP-seq or SATB1 HiChIP-seq protocol.

At this stage (considering our answers to comments 1 and 2 of this Reviewer), it is highly speculative to imply that our manuscript is dealing with SATB1's indirectly regulated genome organization and thus we will not reconsider our findings. It is not clear based on which data the assumptions presented by this Reviewer are made, however, it is reasonable to assume that high-order chromatin structures will at least partially impact finer-scale looping. We should also note that **we did re-evaluate our manuscript and toned down the interpretation of our data** already in the first revision of this manuscript, following the three Reviewers' suggestions:

Lines 202-205: "Taking these results under consideration we conclude that the high-order chromatin organization of murine thymocytes is primarily maintained via CTCF long-range chromatin interactions with minor input from the SATB1-dependent loops."

Lines 263-265: "Collectively, these findings suggested that CTCF participates in mechanisms responsible for supporting a basal high-order T cell chromatin structure, whereupon SATB1 likely exerts its action in a more refined organization layer consisting of promoter-enhancer chromatin loops."

4. In response to my first comment made in the previous review, it was bit surprising that authors replied “in most cases, the H3K27ac loops could be detected in both WT and Satb1 cKO at the same genomic positions”. With the use of differential H3K27ac loop analysis, authors found a positive correlation between changes in H2K27ac interaction frequency and gene expression was found (Figure 4h). A table would be useful to list those genes with a threshold value used to define the differential loops and exact fold change in expression. Furthermore, for most of genes shown their expression is not greatly changed based on the RNAseq data (probably 50% reduction at the maximum). There are many DP genes in the RNAseq data showing more dramatic changes in gene expression depending on SATB1. It seems that none of these genes were found in this group showing the positive correlation. Consistently, the magnitude of change in expression detected in Figure 4f is also small (reduction in expression of genes at anchors of SATB1 loops appears not dramatic). Viewing the entire manuscript, only Bcl6 shown in Fig. 5a, Ccr7 in Supplementary Fig. 12 and Rag 1 in Figure 6d are genes that show major effect on gene expression in Satb1 cKO. This needs to be explained in the context of all SATB1-dependent genes (Elaborate on Supplementary Figure 7a with fold change). Also, for Fig. 4g, T cell subset signature genes at SATB1 loop anchors are enriched in DP compared to DN or SP and this is also the case for H3K27ac loops (both underinteraction or overinteraction) in Supp Fig. 7c. From this set of data, it would be crucial to list up gene names that are strictly SATB1 dependent for looping and show the changes in gene expression depending on SATB1 for each of these genes.

Please note that the differential H3K27ac loops with overlapping genes, their differential RNA-seq values and also their dependency on SATB1 loops are all already available in **Supplementary File 8**. To identify differential H3K27ac loops, we used the default parameters of the differential FitHiChIP pipeline, which means that loops are considered differential with the EdgeR threshold being $FDR < 0.05$ and fold change $\geq \log_2(2)$. We have added in the Methods the fact that we used the default parameters, so it now reads: “*For the differential analysis of H3K27ac WT and Satb1 cKO HiChIP datasets, we utilized the differential analysis pipeline from FitHiChIP¹⁰⁵ using default parameters and only utilized loops that were classified as differential in 3D but not in 1D.* (lines: 706-709)”

Please note that in the revised version of the manuscript, we modified **Fig. 4h** in which it is now more clearly seen that genes in SATB1 loops are predominantly present in H3K27ac underinteracting loops and these genes are mostly downregulated.

This Reviewer speculates that the expression changes observed in our manuscript are too small to be considered significant and/or physiologically relevant. We do not agree with this statement. RNA-seq logFC values can be tricky to interpret, as very relevant genes, such as *Tcf7* and *Lef1*, are expressed at very high levels and changes in their expression are translated into “shrank” logFC values. Moreover, it is well known that for master regulator genes like those, even minimal changes in their expression may result in substantial physiological changes. It is therefore important not to bias ourselves by subjective evaluations and seek only targets that are deregulated 5-fold or more. We should note that we have made a large number of statistical evaluations, used a number of biological replicates and a number of independent approaches to draw our conclusions. Not only are the majority of our presented genes significantly transcriptionally deregulated, according to the standards of contemporary bioinformatics, but their deregulation is also accompanied by several Hi-C and HiChIP experiments for different factors, all converging to the same patterns. Note also our unbiased approaches, such as the regression models, which also gave us the same answers and which are quite unique in comparison to other studies solely relying on the experimental resources. We would also like to highlight that all our experiments were performed in primary cells which distinguishes our work from other studies relying solely on cell lines.

5. Although it is stated in line 226, page 10, a visible drop in chromatin accessibility in the Satb1 cKO (Fig. 4b), ATAC-seq data shown elsewhere for individual genes (e.g. Bcl6. Fig.6) show very

similar profiles and there is no indication of any difference in SATB1-dependent accessibility. Explain.

Figure 4b depicts a global **average**. As such it only depicts a **trend** and not something that could hold true for all of the binding sites studied. We showed in **Fig. 4b**, that chromatin accessibility at SATB1 binding sites (not to be mistaken with genes) is decreased in *Satb1* cKO indicating that SATB1 promotes chromatin opening. This is also in line with other datasets, as discussed in our reply to comment 15 of Reviewer #3 in the Revision 1 document. Moreover, these changes were apparent at the transcription start sites of genes as demonstrated in **Fig. 4c** and **Supplementary Fig. 5f**, and the chromatin accessibility also positively correlated with gene expression as indicated in **Supplementary Fig. 5g**. In terms of the individually depicted genes displayed in **Supplementary Fig. 8** and **Supplementary Fig. 13**, in the majority of them we could also detect decreased chromatin accessibility in *Satb1* cKO (please see the overall log2 fold change for the ATAC-seq of these genes below), even though not always the accessibility changes reflected the extent of RNA-seq data deregulation. Nevertheless, we should bear in mind that the deregulation in chromatin looping, that we study in this work and that is anticipated in *Satb1* cKO, does not necessarily need to correlate with changes to chromatin accessibility: In fact, we were particularly interested in differential looping events that didn't show a big change in H3K27ac marks in the loop anchors.

Additionally, we should also mention that quite similar conclusions were already published for BCL11B, a factor similar to SATB1. In *Hu et al., 2018* (<https://doi.org/10.1016/j.immuni.2018.01.013>), the authors concluded that upon *Bcl11b* deletion, chromatin accessibility was not changed despite the decrease in chromatin interactions: “Consistent with the observation that the regions bound by more *BCL11B* in naive $CD4^+$ T cells displayed higher chromatin interaction than those in DN2 cells (Figure S6E), we detected more decreases in interaction induced by *Bcl11b* deletion in naive $CD4^+$ T cells for TADs with more *BCL11B* binding (Figure 7A), thus supporting the hypothesis that *BCL11B* facilitates chromatin interaction. We found no remarkable decrease in accessibility at DHSs bound by *BCL11B* in *BCL11B*-deficient cells (data not shown), suggesting that *BCL11B* binding is not required for the maintenance of chromatin accessibility at regulatory sites after its establishment.”

6. Since this manuscript aims to demonstrate that SATB1-dependent chromatin organization is correlated (or linked) to gene expression, it is essential to show strictly SATB1-dependent looping (identified by comparing WT H3K27ac versus SKO H3K27ac) is correlated to gene expression. Similar to the SATB1-dependent looping data shown in Figure 6c for the TCR locus organization for rearrangement would be nice to show for SATB1-dependent genes in DP cells. At this time,

even with *Bcl6* gene with the addition of 3C analysis, it is still not fully convincing (too large error bars for *Enh2*, and Hi-C is not quite convincing).

Correlation between differences in H3K27ac looping and gene expression is provided in **Fig. 4h** and specifically for the DN, DP and SP T cell subset signature genes also in **Supplementary Fig. 7d**. Additionally, the differential H3K27ac loops with overlapping genes, their differential RNA-seq values and also their dependency on SATB1 loops, are all available in **Supplementary File 8** as mentioned earlier. Please, see also our replies to Reviewer #2 in the Revision 1 document, specifically the comments 1 and 5 discussing this subject. We should also remind here that it was not a single Hi-C experiment based on which we identified *Bcl6* and other genes presented in **Supplementary Fig. 8**, **Supplementary Fig. 13** and all together listed in **Supplementary File 8**. All these targets were identified by elaborate analysis of multiple biological replicates of SATB1 HiChIP, H3K27ac HiChIP and Hi-C experiments which makes the identification of these targets extremely robust (moreover, we used quite stringent criteria to identify the loops).

Validation by 3C experiments is not common in the modern era of nuclear organization research, as it only achieves validation of the sequencing itself, since the other sources of potential bias remain the same, given the nature of the protocol. Unlike advanced computational and statistical analyses of Hi-C and HiChIP datasets, allowing for correction of biases caused by unequal probability to detect interactions that are further away, this cannot be achieved by the 3C approach. It would be therefore expected that with increasing distance the error bars will also increase. However, in the revised version of the manuscript in response to comment 22 of Reviewer #3, we repeated the 3C experiment using the same restriction enzyme, DpnII, that we used in the actual Hi-C/HiChIP experiments and we now provide **Supplementary Fig. 9** in addition to the **Supplementary Fig. 10a,b**. Note also the 3D modeling in **Fig. 5b** and **Supplementary Fig. 11a,b** accompanying our experimental datasets, once again confirming our conclusions.

Please also refer to our reply to **comment 22** of **Reviewer #3** in this Revision 2 document.

7. Revisit the model shown in Supplementary Fig .10 a and think if this really holds true. This reviewer thinks this model is problematic, as it still tries to relate to CTCF (in light of point 2).

We find that this suggestion is not justified on the basis of an objective argument and so, in line with our replies to comments 1-3 of this reviewer, we would not like to reconsider this model.

8. It is a concern that cKO thymocytes exhibit some SATB1 protein expression. If this is indeed the case, experiments that involve amplification can suffer from contamination of results from WT thymocytes. Western blot should be negative. It is best to contact the two labs which used *Satb1* cKO mice and see if they saw this leakage. The cKO mice are not expected to have target protein expression in cre-expressing cells.

We have repeated the Western blot analysis utilizing wild type and *Satb1* cKO thymocytes, blotted for the long and all SATB1 isoforms. The data are presented in **Supplementary Fig. 1f,g**. Please also take under consideration the following. *Satb1* is deleted at the DP stage but until deletion the gene might already been expressed and a certain percentage of cells might still express SATB1 at very low levels as was suggested by Reviewer 2, point 1a of revision 1. As we have pointed out in our Response to Reviewers in revision 1:

To check the efficiency of SATB1 depletion in *Satb1* cKO thymocytes we have performed the following experiments:

(a,b) FACS analysis for both WT and *Satb1* cKO thymocytes for the detection of **EYFP expression** in the cells. For these experiments we have utilized *Satb1* cKO mice that were also encompassing the EYFP gene as a transgene in the ROSA26 locus which is not expressed unless Cre-mediated excision of a *loxP*-flanked transcriptional “stop” sequence permits EYFP expression and the detection of EYFP. This does not directly answer the

Reviewer's question but at least identifies the fraction of thymocytes that express the Cre recombinase which can act upon the deletion of *Satb1*. We found that more than 96% of the total *Satb1* cKO thymocytes (like it should be based on the fact that total thymocytes still have 1-3% DN cells) and more than 97% of DP thymocytes expressed the EYFP reporter, indicating the efficient expression of the Cre recombinase expressed under the *Cd4* promoter.

- (c,d) As suggested, we have performed **intracellular staining for SATB1** protein, in both WT and *Satb1* cKO thymocytes and found that 2.83-5.4% of total *Satb1* cKO thymocytes still expressed the SATB1 protein. 2.7-3.6% of the double positive - DP (CD4⁺CD8⁺) *Satb1* cKO thymocytes still expressed the SATB1 protein. As shown in Supplementary Figure 2c even the percentage of cells that expresses SATB1, the Mean Fluorescence Intensity (MFI) of these remnant expressors is half compared to the MFI for SATB1 in WT mice. In DP cells, in the *Satb1* cKO it looks like 2.7-3.6% of cells still express SATB1 which would support a leakage in CD4-Cre activity. Though, we should take under consideration the fact that this experiment is based on intracellular SATB1 staining. Due to treatments, for permeabilizing the cells and cell nuclei, the SATB1 positive cells might lose SATB1 protein and thus fluorescence detected in FACS analysis. This is also evidenced by the 2.85% of WT DP cells in which we did not detect SATB1.
- (e) **Immunofluorescence experiments** analyzed by confocal microscopy indicated that SATB1 is expressed in 97.11% of wild type thymocytes and 4.16% of *Satb1* cKO thymocytes, utilizing a commercially available antibody detecting all SATB1 isoforms. The relative SATB1 fluorescence signal was calculated in a quantitative manner utilizing the Volocity software from PerkinElmer. The fraction of *Satb1* cKO thymocytes expressing SATB1 corresponds to the number of DN cells (as deduced from **Fig. 1d**). Alternatively, this could be explained as previously suggested by Reviewer 2, that upon Cre-mediated deletion, the remaining mRNA or protein, based on their rate of decay might still support some remaining SATB1 protein.

Regarding the suggestion of contacting other labs, that utilized *Satb1* knockout cells, we have to note that this specific analysis (measure the SATB1 expression levels in the knockout cells) was not shown by any group that published data utilizing a conditional *Satb1* knockout, or a *Satb1^{f/f};CdCre⁺* mouse model, like the one we used. We are actually the first to present this piece of information. **The following references represent all the papers published so far that utilized conditional *Satb1* knockout cells:**

- Feng D, Chen Y, Dai R, Bian S, Xue W, Zhu Y, Li Z, Yang Y, Zhang Y, Zhang J, Bai J, Qin L, Kohwi Y, Shi W, Kohwi-Shigematsu T, Liao S, Hao B (2021) Chromatin organizer SATB1 controls the cell identity of CD4⁺ CD8⁺ double-positive thymocytes by compacting super-enhancers. Preprint from *Research Square*, 15 Nov 2021
doi: 10.21203/rs.3.rs-1069634/v1
- **They used *Satb1^{f/f};VavCre* mice.**
- **They do not present any data regarding the SATB1 protein levels in the *Satb1* cKO.**
- Delong Feng, Zhaoqiang Li, Litao Qin, Bingtao Hao (2021) The role of chromatin organizer *Satb1* in shaping TCR repertoire in adult thymus. *Genome* 64(9):821-832.
doi: 10.1139/gen-2020-0139.
- **They used *Satb1^{f/f};VavCre* mice.**
- **They do not present any data regarding the SATB1 protein levels in the *Satb1* cKO.**
- Yoshinori Kohwi, Mari Grange, Hunter W. Richards, Ya-Chen Liang, Cheng-Ming Chuong, Yohko Kitagawa, Shimon Sakaguchi, Vladimir A. Botchkarev, Ichiro Taniguchi, and Terumi Kohwi-Shigematsu (2021) Deeply hidden genome organization directly mediated by SATB1. *bioRxiv* <https://doi.org/10.1101/2021.12.19.473323>
- **They used *Satb1^{f/f};Cd4Cre* mice.**
- **They do not present any data regarding the SATB1 protein levels in the *Satb1* cKO.**

- Yukiko Doi, Takafumi Yokota, Yusuke Satoh, Daisuke Okuzaki, Masahiro Tokunaga, Tomohiko Ishibashi, Takao Sudo, Tomoaki Ueda, Yasuhiro Shingai, Michiko Ichii, Akira Tanimura, Sachiko Ezo, Hirohiko Shibayama, Terumi Kohwi-Shigematsu, Junji Takeda, Kenji Oritani, Yuzuru Kanakura (2018) Variable SATB1 Levels Regulate Hematopoietic Stem Cell Heterogeneity with Distinct Lineage Fate. *Cell Rep* 23(11):3223-3235.
doi: 10.1016/j.celrep.2018.05.042.
- **They used *Satb1^{fl/fl}Tie2Cre* mice, *Vav1Cre* mice, *Satb1^{fl/fl}Mx1Cre* mice**
- **They do not present any data regarding the SATB1 protein levels in the *Satb1* cKO.**
- Yuriko Tanaka, Takehiko Sotome, Akiko Inoue, Takanori Mukozu, Taku Kuwabara, Tetuo Mikami, Terumi Kowhi-Shigematsu, Motonari Kondo (2017) SATB1 Conditional Knockout Results in Sjögren's Syndrome in Mice. *J Immunol* 199(12):4016-4022.
doi: 10.4049/jimmunol.1700550.
- **They used *Satb1^{fl/fl}VavCre* mice.**
- **They do not present any data regarding the SATB1 protein levels in the *Satb1* cKO.**
- Tom L Stephen, Kyle K Payne, Ricardo A Chaurio, Michael J Allegrezza, Hengrui Zhu, Jairo Perez-Sanz, Alfredo Perales-Puchalt, Jenny M Nguyen, Ana E Vara-Ailor, Evgeniy B Eruslanov, Mark E Borowsky, Rugang Zhang, Terri M Laufer, Jose R Conejo-Garcia (2017) SATB1 Expression Governs Epigenetic Repression of PD-1 in Tumor-Reactive T Cells. *Immunity* 46(1):51-64.
doi: 10.1016/j.immuni.2016.12.015.
- **They used *Satb1^{fl/fl}Cd4Cre* mice.**
- **They show Western Blot for SATB1 in WT and *Satb1* cKO in CD8+ splenocytes but not thymocytes.**
- Kiyokazu Kakugawa, Satoshi Kojo, Hirokazu Tanaka, Wooseok Seo, Takaho A Endo, Yohko Kitagawa, Sawako Muroi, Mari Tenno, Nighat Yasmin, Yoshinori Kohwi, Shimon Sakaguchi, Terumi Kowhi-Shigematsu, Ichiro Taniuchi (2017). Essential Roles of SATB1 in Specifying T Lymphocyte Subsets. *Cell Rep.* 2017, 19(6):1176-1188.
doi: 10.1016/j.celrep.2017.04.038.
- **They used *Satb1^{fl/fl}Cd4Cre* mice.**
- **They do not present any data regarding the SATB1 protein levels in the *Satb1* cKO.**
- Kitagawa, Y., Ohkura, N., Kidani, Y., Vandenberg, A., Hirota, K., Kawakami, R., Yasuda, K., Motooka, D., Nakamura, S., Kondo, M., Taniuchi I, Kohwi-Shigematsu T, Sakaguchi S. (2017). Guidance of regulatory T cell development by *Satb1*-dependent super-enhancer establishment. *Nat. Immunol.* 18, 173–183.
doi: 10.1038/ni.3646
- **They used *Satb1^{fl/fl}Cd4Cre* mice.**
- **They do not present any data regarding the SATB1 protein levels in the *Satb1* cKO.**
- Motonari Kondo, Yuriko Tanaka, Taku Kuwabara, Taku Naito, Terumi Kohwi-Shigematsu, Akiko Watanabe (2016) SATB1 Plays a Critical Role in Establishment of Immune Tolerance. *J Immunol* 196(2):563-72.
doi: 10.4049/jimmunol.1501429.
- **They used *Satb1^{fl/fl}VavCre* mice.**
- **They do not present any data regarding the SATB1 protein levels in the *Satb1* cKO.**
- Bingtao Hao, Abani Kanta Naik, Akiko Watanabe, Hirokazu Tanaka, Liang Chen, Hunter W. Richards, Motonari Kondo, Ichiro Taniuchi, Yoshinori Kohwi, Terumi Kohwi-Shigematsu, and Michael S. Krangel (2015) An anti-silencer– and SATB1-dependent chromatin hub regulates *Rag1* and *Rag2* gene expression during thymocyte development. *J Exp Med* 212(5):809-24. doi: 10.1084/jem.20142207
- **They used *Satb1^{fl/fl}VavCre* mice.**
- **They do not present any data regarding the SATB1 protein levels in the *Satb1* cKO.**

9. Because this research was done using a particular isoform of SATB1, it has to be stated clearly that these results were derived for the specific isoform of SATB1.

It should be noted that the *Satb1* cKO animals are not specific to any isoform and thus also the majority of the data, such as RNA-seq, ATAC-seq and H3K27ac HiChIP. The only dataset, truly specific for the long SATB1 isoform, was SATB1 HiChIP and accordingly the analyses carried out utilizing these loops. This fact is clearly stated at the first occurrence of SATB1 HiChIP, i.e. **lines 171-173**:

“It is important to note that in the SATB1 HiChIP experiments we used custom-made antibodies specifically targeting the long SATB1 isoform that we recently characterized⁶⁴ (Supplementary File 6).”

Moreover, it is also mentioned in the discussion in **lines 408-409**: *“we should note that in our study we were mostly focused on functions of the long SATB1 protein isoform.”*

Minor points

- 1) Fig. 1c is not very clear to show disrupted intercellular contact.

It should be noted that these images represent tissue sections and not individual cells. Therefore, the dark regions represent nuclei, grey regions cytoplasm and in *Satb1* cKO the whitish spaces are then regions with missing intercellular contacts. In the revised version of the manuscript, we added arrows indicating the sites of missing contacts and we have also indicated cellular membranes of two cells to make this clearer. Please note that we present additional transmission electron microscopy images in **Supplementary Fig. 5a** showing the same pattern.

Overall comment

The author responded that Kitagawa et al paper dealt with Treg cells while this paper (Zelenka et al) is mostly on DP cells. Therefore, SATB1 binding sites at the FoxP3 site is irrelevant to their paper. Their response is not necessarily correct. In Kitagawa et al paper, SATB1 is bound to the super-enhancer at the Foxp3 site already at the DP stage, before the super-enhancer at the Foxp3 is established at Treg cells, Foxp3 expression and greatly increased ATAC-seq signals. Similar to Foxp3 in Treg cells, it is expected that many DP signature genes are activated in DP cells in a SATB1-dependent manner, showing similar response as Foxp3. So, it was expected that those genes would show SATB1-dependent H3K27ac modification, and that looping connecting H3K27ac sites would be greatly altered by SATB1 depletion. Apparently, for some reason, Zelenka et al paper did not see such dramatic results in any genes shown in DP cells (change in H3K27ac, and ATAC-seq and gene expression, as seen for Foxp3 in Treg cells). This is a puzzle and remains to be explained. Therefore, it is not really clear to this reviewer whether SATB1 dependent genome structure (looping, accessibility, histone modification) indeed reflects SATB1-dependent gene expression in a strict sense.

As this Reviewer mentioned, SATB1 is bound to the *Foxp3* super-enhancer already at the DP stage, i.e. **before the super-enhancer is established**. Therefore, we could not detect it with our H3K27ac HiChIP experiments as the region is not yet decorated by the H3K27ac mark. Since H3K27ac signal has to be present for a HiChIP loop to be detected, no loops should be detected at that locus. Moreover, in line with *Kitagawa et al., 2017*, we also detected decreased chromatin accessibility at the *Foxp3* locus in *Satb1* cKO as demonstrated below (see the comparison to the data from *Kitagawa et al., 2017*). See also in the *Kitagawa* data how H3K27ac ChIP-seq signal is basically absent in the DP stage.

Regarding the Reviewer's comment on DP signature genes not being affected by SATB1 depletion, we are concerned that the Reviewer completely oversaw our added analyses dictated towards DP-signature genes. We isolated DP signature genes in an automated manner using ImmGen data, as we already described. It is clear that a) SATB1 loops tend to form more often at DP signature genes (**Fig. 4g, Supplementary Fig. 7d and Supplementary File 8**), b) DP signature genes are very often differentially expressed after SATB1 depletion (**Fig. 1k, l and Supplementary Fig. 7d**) and c) under-

interacting H3K27ac loops are very often found at those gene sites (**Supplementary Fig. 7c,d**) . If the reviewer is referring to the levels of expression not changing that much, we would like to refer the Reviewer to our previous comment regarding logFC changes.

Data from this manuscript to illustrate SATB1 binding and ATAC-seq signal deregulation in *Satb1* cKO

Reviewer #2

I thank the authors for their careful attention to the comments of all the reviewers and for the extensive revisions that they have made. They have thoughtfully addressed all my previous comments, and the paper is considerably stronger in my opinion.

My remaining comments are about figure clarity. These kinds of HiC analytical tools are often difficult to appreciate visually without more guidance.

We thank the Reviewer for appreciating our efforts to answer every comment and improve the quality of our manuscript.

1. In the legend to Fig. 4e, or at least in the similar legend to Supplementary Fig. 4g, please spell out what each of the four quadrants mean and what the ADA score is actually measuring. Once that is clear, the L and R labels are sufficient, but first the reader has to understand what the heat map measures and what the difference is between upper left and upper right, for example.

Fig. 4e and **Supplementary Fig. 4g** are both z-score and bullseye transformed heatmaps to overcome some issues inherently linked to the common square visualization. Here is the description provided in the original publication:

“The bullseye transformation of a heat map is a visualization technique intended to more accurately represent the secondary interactions around a strong loop in the genome. The plot is a simple transformation of the rectangular heat map such that each bin’s Euclidean distance to the center now directly corresponds to its Manhattan distance in the original map. Each ring in the bullseye plot has segments corresponding to the $4 \times N$ bins with a Manhattan distance of N ”

from the central bin. Each bin in a ring takes up exactly the same angular area and they are evenly distributed around the circle. Although this represents some distortion from their actual angles in the original plot, this creates the same visual area for each bin. Z-score transformation is done for each ring separately and the ADA score is obtained by percentage of Z-scores > 1 in the bottom right quarter versus the total plot. (Rowley et al., 2020; <https://doi.org/10.1101/gr.257832.119>)”

We therefore, utilized this explanation and included it in the legend of **Supplementary Fig. 4g** for better clarity, which now reads:

“Intra-domain interaction frequencies of Hi-C matrices derived from WT and Satb1 cKO (SKO) thymocytes, restricted to the SATB1/CTCF-dependent HiChIP loops. Provided bullseye visualization by SIPMeta represents a transformed rectangular heatmap such that each bin’s Euclidean distance to the center now directly corresponds to its Manhattan distance in the original map. Each ring in the bullseye plot has segments corresponding to the 4 × N bins with a Manhattan distance of N from the central bin. Each bin in a ring takes up exactly the same angular area and they are evenly distributed around the circle. Z-score transformation is done for each ring separately and the aggregate domain analysis (ADA) score is obtained by percentage of Z-scores > 1 in the bottom right quarter vs the total plot. L and R denotes left and right loop anchors, respectively. Note the disturbed interaction pattern within SATB1-dependent and not within CTCF-dependent loops in Satb1 cKO cells.”

Moreover, the following statement was added in the legend of **Fig. 4e**:
“See also Supplementary Fig. 4g for more details.”

2. In Fig. 4h, the light transparent blue circles are very hard to distinguish from the light transparent blue triangles where the two sets of symbols overlap. Could the authors make them different colors, and also could they provide different regression lines for the relationships in the SATB1 loops and the relationships outside the SATB1 loops? The same change would be useful in Fig. 6b.

In the revised version of the manuscript, we have implemented the suggested changes to **Fig. 4h** and **Fig. 6b**.

3. Also in Supplementary Fig. 4, there is not enough information on the axes of the plots in panels b and d for the reader to get the point. Readers should not have to look up other references to know what the category labels on the figure axis mean. Could “Non-differential”, “merge”, “split”, “shifted”, and “complex” be defined briefly in the legend to panel b? And please explain what the “relative distance” is relative to, in panel d. The p values are clear, but it would be good also to understand what is being shown.

In the revised version of the manuscript, we have provided a graphical legend explaining each TAD category within the **Supplementary Fig. 4b**.

Relative distance in **Supplementary Fig. 4d** represents a ratio between the smaller distance between a feature from dataset A and upstream or downstream feature from dataset B, and the overall distance between the downstream and upstream features from dataset B, as described in the manual of bedtools reldist. It therefore indicates a spatial correlation between two datasets which is independent of a feature length and which makes it useful in comparing datasets with different peak sizes, such as in our case. In the revised version of the manuscript, we added this description in the legend of **Supplementary Fig. 4d**.

4. In Supplementary Fig. 6, the labels on panels a, b, c, and d are not visible at all. Please add labels with legible (>=8 pt) type to these screenshots.

In the revised version of the manuscript, we increased the size of all labels in these panels to make them more visible.

5. In Supplementary Fig. 8 and in the top panel of Supplementary Fig. 12 (Cd28), it is hard to see a convincing difference in RNA levels just from the tracks. The legends are very minimal. If the point is to demonstrate a change, could the fpkm (rpkm, tpkm) values be indicated alongside these RNA tracks?

In the revised version of the manuscript, we have added DESeq2 normalized RNA-seq counts in **Supplementary Fig. 8** and **Supplementary Fig. 13**.

Reviewer #3

In the revised version of the manuscript, that authors have addressed many of my queries and incorporated a few suggestions. With additions and/or modifications to the existing data presented and its associated text based on mine and other reviewers' comments, the manuscript has been reasonably improved. However, it requires addressal of some key aspects related to the experiments, analyses and the conclusions drawn thereof, for it to be recommended for publication. As such, the authors do not provide unequivocal data to support their title either. The specific comments are listed below:

1. In line 66 of the revised manuscript, the authors claim to identify SATB1 enrichment at the promoter/enhancer, but have not cited the first and defining reports in this context (such as Kitagawa 2017) here but only in generic earlier context (line 56). They should correctly introduce the context of these prior reports.

Line 65-66 reads: "*We have identified SATB1 to be enriched at gene promoters and enhancers involved in long-range chromatin interactions.*" It should be noted that *Kitagawa et al.*, 2017 did not study any involvement of SATB1 in long-range chromatin interactions. To our knowledge, *Kitagawa et al.* did not use either Hi-C, HiChIP or any other Chromosome Conformation Capture-based method, thus it is not a relevant publication to this statement. The simple association between SATB1 and enhancers is introduced (citing also *Kitagawa et al.*) in the relevant paragraph at **lines 100-103** which reads: "*Moreover, it is a known genome organizer^{50,51}, which was already found to be associated with enhancers in double positive CD4⁺ CD8⁺ (DP)^{17,27}, single positive CD4⁺ and developing thymic regulatory T cells¹⁷, yet with a limited number of genome-wide studies targeting its role in 3D chromatin organization of T cells.*"

Moreover, the involvement of SATB1 in long-range chromatin loops is introduced at **lines 73-75** which reads: "*In a proposed model, SATB1 dimers bound to DNA interact with each other to form a tetramer in order to mediate long-range chromatin loops^{33,34}.*"

2. The authors based on previous comments modified text in context of Fig 1a toward identification of Satb1 at SE (using K27ac loops). Since the use of HiChIP datasets was used in region identification and not looping per se, it becomes quintessential to mention the approach and differences from Kitagawa et al, in which they described enrichment of Satb1 at SEs and TEs using K27ac ChIP-seq, more clearly.

We should note that in contrary to what the Reviewer implies, the identification of SATB1 was based upon H3K27ac looping per se. SATB1 was found to be enriched at the anchors of H3K27ac loops which directly suggested that SATB1 is responsible for formation of a part of these regulatory H3K27ac loops. This approach is clearly described both in the main text at **lines 91-94**:

"Loop calling at 5 kbp resolution (FDR ≤ 0.01) yielded 16,458 regulatory loops. To identify the prospective protein factors associated with these regulatory loops, we intersected the anchors of these loops with all the available murine ChIP-seq datasets from blood cells, using the enrichment analysis of ChIP-Atlas⁴²."

and in the methods at **lines 908-912**:

“Loop anchors from WT H3K27ac HiChIP loops were pre-processed by extracting the anchors from both sides of loops and then merging them into non-overlapping unique regions (mergeBed command of bedtools)¹⁰⁹. The resulting regions were converted from mm10 to mm9 using CrossMap¹²⁷ and were analyzed using the enrichment analysis ChIP-Atlas⁴² against all the available murine ChIP-seq datasets from the blood cell type class and compared to 100x random permutations.”

3. In response to comment #3, the authors performed a reldist analysis with ‘all-iso’ available datasets (Supp. 4d). Although this shows some level of qualitative similarity, it would be more robust to show a quantified Statistic such as jaccard. Additionally, the authors should also correlate these two datasets as a positive control. Further, we understand that the trend remains the same, but to answer our question, the authors should show an absolute overlap (percentage) between the features (Supp. 4e).

We should note that there is a large difference in size and variance of SATB1 binding sites identified in this work (peak size **201±57 bp**) and in the public datasets (*Hao et al., 2015: 1464±767 bp*; *Kitagawa et al. 2017: 363±238 bp*). This indicates that the datasets were prepared under different conditions and thus any reliable comparisons are difficult/impossible. As a means of set overlap, the Jaccard statistic is suitable for sets with similar, identical or point-like sizes, in which the intersection over the union calculation is straight-forward. This is not the case with the genomic segments under question with great size variance, which makes the assessment of nested peaks problematic. In such cases it is always necessary to accompany the Jaccard index computations with permutation tests as the underlying distribution is hard to estimate. To partially compensate for the differential length, we extended the peak sizes to mimic the peak length of *Hao et al., 2015* dataset which has the largest peaks. Nevertheless, this approach is not ideal. The Jaccard statistic is then summarized below and also added in the **Supplementary Fig. 4d**: - Long iso vs Kitagawa et al., 2017: 0.0475077

- Long iso vs Hao et al., 2015: 0.039982
- Shuffled long iso vs Hao et al., 2015: 0.0077481
- Shuffled long iso vs Kitagawa et al., 2017: 0.00796765
- Kitagawa et al., 2017 vs Hao et al., 2015: 0.381913

Moreover, we replaced **Supplementary Fig. 4d** with a figure showing the reldist analysis for the two publicly available datasets, i.e. *Kitagawa et al., 2017* vs *Hao et al., 2015*.

For the aforementioned reasons, we also find it misleading to display the absolute overlap of features with SATB1 binding sites as it is not comparable among the datasets. Yet, in the revised version of the manuscript we added these numbers in **Supplementary Fig. 4e**.

4. In line 109, which intracellular contacts are the authors referring to? Its not shown or indicated in the figure referred here (1c). The authors should show clearer images along with indicators for the phenotype mentioned inside the figure.

We refer to disrupted **intercellular** contacts instead of intracellular, i.e. the broken tissue integrity with cells detached from each other. As discussed in the minor comment 1 of Reviewer #1, it should be noted that these images represent tissue sections and not individual cells. Therefore, the dark regions represent nuclei, grey regions cytoplasm and in *Satb1* cKO the whitish spaces are then regions with missing intercellular contacts.

In the revised version of the manuscript, we added arrows indicating the sites of missing contacts and we have also indicated cellular membranes of two cells to make this clearer. Please note that we present additional transmission electron microscopy images in **Supplementary Fig. 5a** showing the same pattern.

5. In figure 1d (top panel); Since the authors generated *Satb1* fl/fl mice and claim that the change thymic cell numbers are in coherence with previous studies, they should reconcile their findings with the Kakugawa et al. (2017) paper that showed a dramatic decrease in SP cell numbers (~50%).

We thank the Reviewer for bringing this up. Please note that the distribution of our data is very similar to the data from *Kakugawa et al.* This confusion could be caused either by different order of x,y axis or by the different amounts of cells scored. The difference in percentages was caused by differently set quadrants. In the revised version of the manuscript, the quadrants were set in the same way as in *Kakugawa et al.* and the percentages are now much more similar. Note that this modification affected the **Fig. 1d** and **Supplementary Fig. 3c**.

6. Whereas the figure presented in the manuscript using flow cytometry (1d) shows only minimal variation from the WT controls. Furthermore, the authors should cite the Alvarez et al., 2000 study because of more similarity of KO conditions. To further characterize their cKO model, the authors should also show immature and mature thymic populations using the same gating.

We should note that the pattern of our flow cytometry data resembles the one from other published studies. Possible differences that could be confusing are that we scored higher number of cells, we have put the x and y axis in the opposite order and then the setup of quadrants as explained in the comments 5 and 7 of this Reviewer. Moreover, detailed immunophenotyping is not the goal of this paper since we do not present a novel knockout mouse model and we demonstrated that our mouse model is similar to previously published studies.

7. In line 119, the authors refer to Kondo et al., 2016 for reconciling their peripheral phenotype. However, the thymic data presented here doesn't correlate with this (in the context of DP change). The authors should mention and discuss this discrepancy. It is imperative that the authors should also show the populations of LN for the same gatings.

In response to comment 5 of this Reviewer, we have slightly modified the quadrants of flow cytometry data of our thymic samples. Upon this modification, the percentages of DP T cells in the thymus of our *Satb1^{fl/fl}Cd4-Cre⁺* knockout animals highly resemble the data from *Kondo et al., 2016* (**despite the fact that we are comparing different knockout animals**; see the comparison below). Therefore, we do not foresee any discrepancy between our data and the data previously published regarding *Satb1* knockout animals. To demonstrate the deregulation of peripheral lymphoid organs, we have presented data from spleen in **Fig. 1g** and **Supplementary Fig. 3c** and in the revised version of our manuscript we have now added also data from lymph nodes in **Fig. 1h** and **Supplementary Fig. 3c**.

8. Following our earlier comment, authors have added immunostaining data for Satb1 (all iso) in their KO conditions. Since most of the following work and the premise of it is set around the long isoform, they should also demonstrate the percentages of cells specifically expressing this isoform.

We have performed additional immunofluorescence experiments in thymocytes and we now present the percentage of cells expressing the long SATB1 isoform in both wild type and *Satb1* cKO thymocytes in **Supplementary Fig 1e**.

9. The authors have added clearer and much better depicted images of pancreas in the revised manuscript. Although in line 126, and Supp. 3e, the authors mentioned impaired glucose tolerance. It starts and ends same through the time-course with broader variation. In response to our comment #8 they wrote 'at the steady state time point before fasting, the knockout animals have significantly elevated glucose levels compared to their WT' which doesn't reflect in the figure.

In the revised version of the manuscript, we have added *P* values by Wilcoxon rank sum test to the **Supplementary Fig. 3e** comparing WT and *Satb1* cKO values for each time point, indicating the significant difference in glucose levels at the steady state.

10. In their response to our earlier comment #7, the authors mention 'Moreover, this article focused on the roles of SATB1 in peripheral T cells, while our manuscript mainly focuses on intra-thymic T cell development.' However in actual sense, the authors move (and cite) to and fro between thymic and peripheral references according to what suits the specific part of their data. In this particular case, since they talk about blood serum levels of cytokines, it would rise from peripheral mature cells. Having said that, the authors state one of the conclusions from Yasuda 2019 about the dispensability of Satb1 for Th17 response. In such instance, the authors should state what do they infer from their data and how it reconciles with the previous study, citing the same.

In the revised version of the manuscript, we modified the legend of **Fig. 1j** in **Lines 1365-1367** which now reads: "Differences in the cytokine milieu in the blood serum of 5 WT and 11 *Satb1* cKO (SKO) animals measured with bead-based immunoassay. Note the elevated IL17 response and increased inflammatory cytokines."

We should make clear that in *Yasuda et al., 2019* (<https://doi.org/10.1038/s41467-019-08404-w>), the authors based their conclusions on an *Il17a-Cre* mouse (*Satb1* was therefore deleted in IL-17-producing T cells) that displayed impaired production of GM-CSF, a key pathogenic cytokine associated with autoimmunity. Moreover, they concluded that SATB1 is dispensable for the differentiation and non-pathogenic function of Th17 cells. In our work, we only characterized the cytokine levels in the blood serum of our CD4-Cre *Satb1* cKO animals and identified increased levels of IL-17. We implied that this could be linked to the deregulation of the intra-thymic developmental programs, since this is where SATB1 is predominantly expressed, where it is depleted for the first time during ontogenesis, and where it arguably plays its major biological role. Our system is thus not comparable to the data from *Yasuda et al.* since the two manuscripts investigated the roles of SATB1 in completely different cell types. We wanted to highlight with our

experiments that T-cell development is disrupted and as such peripheral T-cells may show defects. This is apparent from **Lines 127-129**: “Deregulation of the thymic developmental programs was also supported by the altered cytokine milieu in the blood serum, with prevailing IL-17 cell responses and increased levels of pro-inflammatory cytokines such as IFN γ and TNF α detected in *Satb1* cKO sera (Fig. 1j).”

Additionally, Yasuda *et al.*, 2019 used a *Thpok*-Cre conditional knockout mice (*Satb1* was deleted in CD4⁺CD8⁻ SP thymocytes and in peripheral CD4⁺ T cells) which did not display any deregulation of either DP, CD4SP, CD8SP T cell numbers in the thymus, or IL-17-producing CD4⁺ T cells in the periphery. This is in direct contrast to the phenotype of the *Cd4*-Cre mouse (*Satb1* is deleted already in CD4⁺CD8⁺ DP T cells) that we and other studies, employing this knockout line, showed in terms of thymic deregulation (please see **Fig. 1d** and **Supplementary Fig. 3c**). Therefore, these data indicate that SATB1 plays a major role in the intra-thymic development before ThPOK is expressed, i.e. mostly during the DP stage. Nowhere in the text we explicitly link the function of SATB1 with peripheral Th17 cells. In the discussion, in **Lines 451-463** we hypothesized how the IL-17 deregulation might be caused, suggesting two directions, though, both linked to the dysregulated **intra-thymic development** in *Satb1* cKO mice. One due to deregulated *Bcl6*, which could be carried over through epigenetic mechanisms to peripheral cells, as previously reported (Mathew *et al.*, 2014; <https://doi.org/10.1084/jem.20132267>); and which could possibly explain the increased IL-17 response due to the decreased levels of Th17-antagonist BCL6. Alternatively, we hypothesized that this could be caused by deregulated $\gamma\delta$ T17 cells, thus also intrinsically linked to the **intra-thymic T cell development** and its deregulation in *Satb1* cKO. Additionally, we have added a statement about the findings reported by Yasuda, which are based on different mouse models. The aforementioned **Lines 451-463** read: “*BCL6* is known to function as an antagonist of factors specifying other cell lineage fates^{95,96}, especially of PRDM1 and underlying Th17 lineage specification⁷³. Indeed, the increased IL-17 response we observed in the cytokine milieu of *Satb1* cKO mouse sera (Fig. 1j), suggests a favored Th17 specification due to downregulation of BCL6. However, since the depletion of SATB1 and its underlying regulome took place already during the intra-thymic development (*Satb1*^{fl/fl}Cd4-Cre⁺), we hypothesize that the increased IL-17 cytokine levels could be primarily due to elevated $\gamma\delta$ T17 cells. Based on the transcriptomic analysis depicted in Fig. 6c (RNA-seq, WT/*Satb1* cKO) we have detected the overexpression of the *TCRc5* locus and correspondingly also of the V γ 4⁺ and V γ 6⁺ chains which are expressed in $\gamma\delta$ T17 cells⁹⁷⁻¹⁰⁰. Moreover, the most overexpressed gene in *Satb1* cKO thymocytes, as indicated by our RNA-seq experiments, was *Maf* (encoding c-MAF; log₂FC = 3.696 in female thymus and log₂FC = 7.349 in male DP cells), which was shown to be essential for the commitment of $\gamma\delta$ T17 cells¹⁰¹. Note that we used a Cd4-Cre⁺ mouse and our findings cannot be directly related to the former report on dispensability of SATB1 for the differentiation of peripheral Th17 cells which were based on different mouse models targeting later developmental stages of T cells¹⁰².”

11. We agree with the authors that maximum enrichment is of DP genes. But given our and reviewer #1's concern regarding the use of total cells, and the contention by the authors that it is 90% DP, the enrichment for SPs is still quite high (4 fold compared to 7 fold for DP). This also reflects in the subsequent figure 1k. The authors should critically revisit their selections for data/signature for downstream analysis and revise these figures.

The selection of these genes cannot be revisited given the unbiased nature of the selection process as described in the Methods (**lines 980-989**), so that the genes truly represent the DN/DP/SP signature genes. SP signature genes' expression, peaks in SP T-cells, but that does not completely exclude that they are already expressed in the DP stage. Thus, part of the enrichment can stem from this fact. However, it should be noted that the primary aim of this analysis was to support the existence of a SATB1-dependent promoter-enhancer network in DP T cells by demonstrating that indeed the DP signature genes are the most affected in *Satb1* cKO.

This does not rule out the option that other cell types lacking SATB1 could be transcriptionally affected too. Especially the SP T cells have SATB1 depleted too, and as direct descendants of DP T cells it is anticipated that they would display certain transcriptional deregulation due to various cell-intrinsic and -extrinsic reasons. Yet we still showed almost 2-fold enrichment of DP DEGs over SP DEGs. But given the main objective of this analysis, more relevant graphs in this context are included in **Fig. 4g**, displaying an almost 3-fold enrichment of DP signature genes present at SATB1 loop anchors over SP signature genes; and **Supplementary Fig. 7d**, showing the correlation between SATB1-dependent looping and transcriptional deregulation as indicated by the abundance of SATB1-dependent loops (triangles) in the lower left quadrant of DP vs SP T cells.

12. As a part of response to our comment #11, the authors wrote ‘While the long isoform RNA levels are 3-5-fold lower than the RNA levels of collectively all short SATB1 isoforms, both imaging and western blot experiments indicated high abundance’. While both imaging and WB are at best semi-quantitative, the reverse correlation of transcript and protein could easily be attributed to the efficacy of the lab-made antibody (especially when compared directly to a commercial one). It is necessary to show using more quantifiable methods, at least some validation in the supplementary data of this study since the other manuscript in reference is itself probably under review.

First, it is not clear how quantifying the levels of long SATB1 isoform would improve our conclusions on the link between transcriptional regulation and 3D chromatin organization, already based on Hi-C, HiChIP, ATAC-seq and RNA-seq experiments. Second, it is not clear what kind of validation is proposed and what kind of quantifiable methods could be utilized. However, we believe that in the revised version of the manuscript we provide more than enough evidence about the long SATB1 isoform’s existence (whose description is however not the subject of this work), as well as validity of our antibody.

The long *Satb1* isoform is expressed at the mRNA level, as indicated by a thorough and isoform-specific analysis of our total-stranded-RNA-seq experiments (**Supplementary File 2**). In **Supplementary Fig. 1d,f** and **Supplementary Fig. 1e,g** we provide a direct comparison of immunofluorescence experiments and Western blot analysis in WT and *Satb1* cKO between a commonly utilized SATB1 antibody, non-specifically recognizing all isoforms, and our long isoform-specific antibody, respectively. These results well support the existence of the long SATB1 isoform protein. Additionally, they support the specificity of the long SATB1 isoform antibody as it detected no protein in SATB1-depleted T cells (**Supplementary Fig.1g**).

Furthermore, if the Reviewer by “*validation*” refers to the **validation of the antibody detecting the long SATB1 isoform**, we have performed additional experiments utilizing bacterially expressed His-tagged recombinant proteins for both the long and short SATB1 isoforms. We have performed Western blot analysis of these recombinant proteins and have detected them utilizing three different antibodies: i. The long SATB1 isoform-specific antibody, ii. A commercially available antibody detecting all SATB1 isoforms and iii. An anti-Histidine antibody detecting the His-tagged SATB1 proteins. The data are presented in the **revised Supplementary Fig. 2e.f**. Based on this data, we concluded that in murine thymocytes a long SATB1 isoform is expressed at both the mRNA and protein levels.

If the question refers to the relative expression levels of the long and short SATB1 protein isoforms in the same cell population, this is impossible to be unraveled based on immunofluorescence experiments coupled to confocal analysis, simply because the antibodies targeting the short SATB1 isoform, also target the long SATB1 isoform (a *Satb1* knockout deleting the extra exon of the long *Satb1* isoform would be helpful to partly answer this question but is not yet available).

The best way we thought in order to check for the relative protein expression levels of the long versus the short SATB1 isoforms in primary T cells is described below and has been included in the accompanying manuscript we have posted to *bioRxiv* (*T. Zelenka, P. Tzerpos, G.*

Panagopoulos, K-C. Tsolis, D-A. Papamatheakis, V.M. Papadakis, D. Stanek, C. Spilianakis, SATB1 undergoes isoform-specific phase transitions in T cells, doi: <https://doi.org/10.1101/2021.08.11.455932>. A description of the rationale and the actual experiment is provided below:

To validate the existence of the long SATB1 isoform, we generated a custom-made antibody specifically targeting the extra peptide of 31 amino acids, originating from the extra *Satb1* exon. None of the commercially available antibodies can distinguish between the two SATB1 isoforms *in vivo*, as they target parts of SATB1 shared by both isoforms. It should also be noted that when investigating the SATB1 protein levels, we have to bear in mind that the antibodies targeting either the N- or the C-terminus of the SATB1 protein cannot discriminate between the short and long isoforms, thus we can only compare the amount of the long SATB1 isoform to the total SATB1 protein levels *in vivo*. To overcome this limitation and to specifically validate the presence of the long SATB1 protein isoform in primary murine T cells, we designed a serial immunodepletion-based experiment (Panel A, of the figure that follows). We found that the antibody raised against the extra peptide of the long SATB1 isoform, specifically detects only the long SATB1 isoform (Panel B, lane 2) and not the short SATB1 isoform (Panel B, lane 3). This experiment can also be used for the quantitation of the cellular protein levels of SATB1 isoforms in primary murine thymocytes. Comparison of the two bands for the long (lane 2) and the short SATB1 (lane 3) isoform in Panel B suggests that upon densitometric quantitation of the band intensity, the long SATB1 isoform protein levels are 1.5x to 2.62x less abundant than the short isoform, according to the two replicates of our immunodepletion experiments, respectively.

FIGURE LEGEND. (A) Scheme for the approach utilized to detect SATB1 long isoform and validate the custom-made long isoform-specific antibody (Davids Biotechnology). Whole cell thymocyte protein extracts (sample 1) were prepared and incubated with a custom-made antibody against the long SATB1 isoform. The immunoprecipitated material (sample 2) was kept and the immunodepleted material was subjected on a second immunoprecipitation reaction utilizing an antibody detecting epitopes on both long and short SATB1 isoforms (Santa Cruz Biotechnology, sc-376096). The material from the second immunoprecipitation reaction was kept (sample 3). (B) Western blot analysis for the samples described in (A). The whole thymocyte protein extract (1), immunoprecipitated long SATB1 protein (2) and immunoprecipitated short SATB1 protein (3). UPPER PANEL: Western blot

analysis utilizing a SATB1 antibody detecting only the long SATB1 isoform. LOWER PANEL: Western blot analysis utilizing a SATB1 antibody detecting all SATB1 isoforms.

13. In line 153 and related figures 2b,c,d, the authors have compared Satb1 KO dataset they generated with CTCF and RAD21 KO datasets from previous literature. Firstly, the enzymes used for digestion in those data are different from the ones used in the new data (eg HindIII for CTCF KO and DpnII for current dataset). It is widely known that 6 base cutters have very different biases than a 4 base cutter. Therefore, the authors should expound on the caveats of such comparison. Additionally, they can further normalize for the bias and plot the data again to increase the robustness of their analysis.

First, we should note that it was this Reviewer's suggestion to include these datasets in our analysis. It is indeed true that there are multiple biases that affect the Hi-C data. As described in **lines 744-749** of the manuscript, the matrices used for these analyses were normalized to the smallest dataset from each compared pair (WT vs KO data) and additionally corrected by the KR or ICE method. We do not think that further normalization considering the use of different restriction enzymes is needed because in our analysis we mostly compared WT vs KO data that were obtained using the same protocol, i.e. having the same restriction enzyme bias. The only figure in which we performed a direct comparison among the SATB1, CTCF and RAD21 KO datasets is **Supplementary Fig. 4b**. However, in this figure we have made a comparison of differential TADs (i.e. following the WT vs KO differential analysis) which should no longer be confounded by any bias.

14. Importantly, reasoning by the authors in their rebuttal for use of mESC dataset about the conservation of TADs (so far apart in cell stages) is not acceptable, since their own premise is changes in 3D loops in one single cell type.

We should note that there is a big difference between regulatory chromatin loops, which are highly cell-type specific, and TADs which are conserved. In Methods, **lines 699-700** that read: "*since TADs are conserved among different cell types¹, the comparison between T cells and mESCs could be performed*", we refer to the study by Rao et al., 2014 (<https://doi.org/10.1016/j.cell.2014.11.021>) that showed domain conservation between lymphoblastoid and lung fibroblast cell lines (apart from others). However, the TADs' conservation across cell types, during differentiation and along evolution was also reported elsewhere, e.g.:

- Dixon et al., 2012 (<https://dx.doi.org/10.1038%2Fnature11082>);
- Nora et al., 2012 (<https://dx.doi.org/10.1038%2Fnature11049>);
- Dixon et al., 2015 (<https://doi.org/10.1038/nature14222>);
- Krefting et al., 2018 (<https://doi.org/10.1186/s12915-018-0556-x>); and references therein.

On the other hand, regulatory and transcription-dependent loops are highly cell-type specific, as one would expect given the distinct transcriptional programs of different cell types and as deduced from HiChIP and ChIA-PET experiments probing the finer-scale chromatin looping, e.g.:

- Li et al., 2012 (<https://doi.org/10.1016/j.cell.2011.12.014>);
- Mumbach et al., 2017 (<https://doi.org/10.1038/ng.3963>);
- Weintraub et al., 2017 (<https://doi.org/10.1016/j.cell.2017.11.008>);
- Grubert et al., 2020 (<https://doi.org/10.1038/s41586-020-2151-x>).

This is also in line with the recent views of transcriptional regulation via transcription factors-based 3D genome organization, e.g.:

- Stadhouders et al., 2019 (<https://doi.org/10.1038/s41586-019-1182-7>);
- Kim et al., 2019 (<https://doi.org/10.1016/j.molcel.2019.08.010>);
- Campigli Di Giammartino et al., 2020 (<https://dx.doi.org/10.1080%2F15384101.2020.1805238>).

11. Line 187, the authors mention CTCF RNA levels are unperturbed, but do not refer to any figure, and/or dataset.

In the revised version of the manuscript, we now refer to the **Supplementary File 2**, containing the RNA levels.

We have also performed RT-qPCR in wild type and *Satb1* cKO thymocyte RNA which is now presented in **Supplementary Fig. 4h**.

We have also performed Western blot analysis utilizing whole thymocyte protein extracts from wild type and *Satb1* cKO mice. The result is presented in the **revised Supplementary Figure 4i**.

12. In line 198, the authors did not cite the original references attributing SATB1 mediated T cell loopscape (cited in reviewer response but not the manuscript).

Publications referring to SATB1-dependent nuclear organization were properly cited at earlier occasions. Moreover, in the revised version of the manuscript we have now replaced the term loopscape so that the final version reads:

“Gene ontology analysis of the genes intersecting with loop anchors uncovered the high propensity of SATB1 to participate in chromatin loops involving immune-related genes, while CTCF-dependent chromatin loops exhibited omnipresent looping patterns resulting in the enrichment of general metabolic and cellular processes (Fig. 3e). (Lines 199-202)”

13. With respect to Supp. Fig 5 a,b and d, The authors claim that there is compaction and heterochromatinization. As with HP1 figure in the older version, S5a is also unclear. Instead of nuclear compaction, as posited by the authors, there seems to be that the whole cell/section is shrunk longitudinally. Furthermore, the MNase (5b) treatment seems to be almost same, with no significance in quantitation. For 5d, at least visibly it appears to have exactly the same fluorescence, which according to the quantitation is nearly halved. The authors should provide better representative images.

For **Supplementary Fig. 5a** we wrote about “deregulated heterochromatin organization” which should not be confused with compaction. We also referred to disrupted intercellular contacts, i.e. the broken tissue integrity with cells detached from each other. It should also be noted that these images represent tissue sections and not individual cells. Therefore, the dark regions represent nuclei, grey regions cytoplasm and in *Satb1* cKO the whitish spaces are then regions with missing intercellular contacts. In the revised version of the manuscript, we added arrows indicating the sites of missing contacts and we have also indicated the plasma membranes of two cells to make this clearer (both in **Fig. 1c** and in **Supplementary Fig. 5a**).

In relation to the compaction, in order to increase the clarity, we replaced the original figure with MNase treatment with another approach utilizing DNase I treatment. In **Supplementary Fig. 5b**, we now present gel electrophoresis of genomic DNA from WT and *Satb1* cKO thymocytes treated with increasing concentration of DNase I. The red rectangle indicates that for the same DNase I concentration the WT chromatin is digested more and therefore is considered more accessible compared to the *Satb1* cKO thymocyte chromatin which tends to be more compact.

For Supplementary Fig. 5d we now provide new representative images (**revised Supplementary Fig. 5e**) so that the picture conforms to the differences uncovered upon quantitation of 952 cells for DAPI staining and 144 cells for RNA Pol II staining.

As a general comment we would like to mention that we have provided numerous methods to draw the same conclusion regarding the minor differences of the heterochromatin content in wild type and *Satb1* cKO thymocytes. We do not base any major conclusion in our manuscript on this difference other than being an interesting observation. Therefore, it is not clear to us why the Reviewer insists on the addition of these numerous biochemical assays which all support the same piece of data already deduced by the ATAC-seq experiments, provided in the manuscript. ATAC-seq is considered these days as the state of the art for studying chromatin accessibility.

14. In response to our comment #18, the authors wrote 'It is not clear how we could validate decreased chromatin accessibility by qRT-PCR'. The authors should note that this can be easily accomplished by designing primers at/around the binding site enriched in their analysis for PCR. It is widely used to show accessibility changes just like 3C-qPCR.

We should note that RT-qPCR stands for **reverse transcription quantitative PCR** where the starting material is RNA. In the case of ATAC-seq, we are talking purely about DNA accessibility and DNA regions which do not necessarily correspond to genes and thus are not even necessarily transcribed. Therefore, the suggested approach cannot be used to validate the identified regions. Moreover, no PCR-based approach was used as a validation technique in any of the original ATAC-seq publications such as *Buenrostro et al., 2013* (<https://dx.doi.org/10.1038%2Fnmeth.2688>) or *Buenrostro et al., 2015* (<https://dx.doi.org/10.1002%2F0471142727.mb2129s109>).

Though, since our intention is not to argue with the Reviewers but make our manuscript better and suited for publication, we have utilized qPCR to study DNase I accessibility of the *Bcl6* promoter which displays SATB1 binding and additionally shows reduced chromatin accessibility based on ATAC-seq experiments. Indeed, *Bcl6* gene promoter is less accessible to DNase I in the *Satb1* cKO thymocytes (**Supplementary Fig. 5d**).

15. In response to our comment #19, the authors state the usage of ImmGen data to pick out signature genes for downstream usage in Figure 1j,k. Whereas they posit in the main text that they did analysis on publicly available data for manually picking up gene clusters for the same. This should be clarified and appropriate sources cited.

The methodology to identify the T cell subset signature genes was comprehensively described in the Methods section with all the sources cited (**lines 980-987**). Additionally, at its first occurrence in the main text in **lines 135-139**, we have also cited the respective study (*Yoshida et al., 2019*; <https://doi.org/10.1016/j.cell.2018.12.036>) from which the sorted ImmGen datasets were extracted:

*“Additionally, here we identified double negative (DN), DP and SP T cell signature genes (Supplementary Fig. 4a) using data available from sorted T cell subsets⁵⁵. In line with the hypothesis that the DP stage is the most affected in *Satb1* cKO animals, this analysis indicated the highest overlap enrichment between DP signature genes and differentially expressed genes compared to other T cell subset signature genes (Fig. 1k, Supplementary File 3). Moreover, the DP signature genes displayed the highest drop in RNA levels in *Satb1* cKO animals compared to DN and SP signature genes (Fig. 1l).”*

16. With respect to Supp. 7b, the authors should also show what is the correlation of DP signature genes enriched by *Satb1* overlap with those thymic enhancers.

Supplementary Fig. 7b depicts the overlap enrichment between loop anchors and enhancers. The overlap enrichment between genes and enhancer loops is depicted in **Fig. 4d**. In line with our data, the thymus-specific enhancers used in our study (*Shen et al., 2012*; <https://doi.org/10.1038/nature11243>) were identified from the entire thymus – proportionally including enhancers from DN/DP/SP T cells.

17. In many places of the text, as well as Supp. Fig 7, the authors emphasise the DP restricted role of *Satb1*. But it is of most importance to realize in the text that the authors have used total thymocytes and did not directly compare *Satb1* enrichment in different cell types. Logically, it would definitely lead to DP signature. Unless single cell or single population based analyses are done, it would be inappropriate to mention selective differential role in cell types (line 282).

We thank the Reviewer for the valuable comment regarding **Supplementary Fig. 7**. We rephrased its title to: “*SATB1-dependent promoter-enhancer chromatin loops in developing T cells*”, and we modified the legend of **Supplementary Fig. 7c,d** which now reads:

“**c** *Overlap enrichment of T cell subset signature genes at anchors of H3K27ac under- and over-interacting loop anchors. Note the high enrichment of DP signature genes for both categories.* **d** *Scatter plots indicating positive correlation (Spearman’s ρ) between changes in RNA levels based on RNA-seq and differential H3K27ac looping between WT and Satb1 cKO. Three leftmost graphs show correlation for DN, DP and SP T cell subset signature genes (described in Supplementary Fig. 4a). The rightmost graph shows that the correlation is highest when only significantly differentially expressed genes (DEGs) are selected. Note that in DP subset, the majority of genes is located in the bottom left quadrant and present in SATB1-dependent loops (triangles).”*

We nevertheless respectfully disagree that our analysis, utilizing the T cell subset signature genes, is biased towards the DP subset, as explained in **comment #11** of this Reviewer. We employed objective and unbiased approaches to support our claims that SATB1 mostly plays genome organization roles in DP T cells. We have modestly toned down these statements in the legend of **Supplementary Fig. 7c,d**. We also removed the part “*collectively indicating a DP T cell stage-specific regulatory role of SATB1*” in line 282, although this statement already seems to be toned down to us. It would be helpful if instead of “many places of the text” the Reviewer referred also to other specific locations. To our knowledge, in the rest of the main text we refer to SATB1’s roles in developing T cells which is scientifically accurate. We only mention DP cells in parts of the text where we have been asked by the Reviewers to perform additional analyses considering the DP and other T cell subsets. These instances cover the signature genes of the specific subsets and as such we mention it this way (such as **in lines 135-139** and **280-285**).

Although one could sort only DP cells, there is not enough data showing that FACS sorting and extended cell manipulation would not affect the 3D genome organization which was our primary goal in this study.

18. We appreciate that the authors have performed 3C experiment for validation. However, they have used a different enzyme here for digestion compared to their HiC protocol. Why have they not used DpnII? Furthermore, its only mildly decreasing with no significance. Using a different enzyme might explain the same. At least regions with significant enrichment should be shown.

Hi-C is designed to detect chromatin loops and cumulatively estimate the loss or gain of loops between two chromatin regions unlike the design for detecting the interaction of only two restriction fragments. Hi-C loops are based on hundreds of millions of reads detecting any possible interaction between chromatin fragments while 3C was designed to detect the interaction of only two chromatin fragments. That is why Hi-C is mostly based on the use of 4-cutters while 3C is based on the use of 6 cutters since the optimum length of chromatin fragments that can efficiently be analyzed have to range between 2-10Kb in length, having to do with the efficiency of a specific protein to bind and mediate a specific 3C interaction. This is the reason why we utilized a 6-cutter in the 3C experiment and we reasoned that it would be a great alternative proof for the interactions detected by Hi-C and the use of a 4-cutter as it should be. Following this Reviewer’s suggestion, we have designed again the 3C experiment, utilizing DpnII and performed the 3C experiment for the *Bcl6* gene locus. The data for multiple replicates are presented in **Supplementary Fig. 9**, showing decrease in interactions between the *Bcl6* promoter and its super-enhancers in *Satb1* cKO T cells.

19. The new spleen sections are much clearer and explain the phenotype much better. In contrast, the Western blots (e.g. Bcl6) show a lot of pixilation hence better WB images should be shown. For Rag proteins (especially Rag2), the WB visibly does not correlate the transcript data. There is no change in Rag2, whereas Actin is halved. It is therefore required that the authors repeat the WB for better clarification.

We have repeated the Western blot analysis utilizing whole thymocyte protein extracts from wild type and *Satb1* cKO mice. The Western blot analysis for BCL6 is presented in the **revised Figure 5c**. The Western blot analysis for RAG2 is presented in the **revised Figure 6f**.

24. It is also important to validate the expression of these select targets via qRT-PCR in KO condition.

Although we again do not understand the value of reconfirming the RNAseq data based on RT-qPCR, such analysis was performed and the data are presented in **Supplementary Figs. 4h and 10c**.

Once more we would like to thank the Reviewers for their suggestions. We hope that we have addressed all the points in a satisfactory way and we hope that they will now find our manuscript suitable for publication.

Sincerely,

Charalampos G. Spilianakis, Ph.D

REVIEWERS' COMMENTS

Reviewer #2 (Remarks to the Author):

The authors have done an extraordinarily patient, thorough, and scholarly job of responding to the comments of the various reviewers. I am happy with all their responses to my own previous comments and greatly appreciate the increased clarity of the effects they demonstrate. There are just three word choices left that I have questions about. If they should indeed be changed, this can easily be done during production:

1. In the new (and very helpful) updated legends to Fig. 4e and Fig. S4g, did the authors mean counts >1 in the bottom RIGHT quadrants, or the bottom LEFT quadrants?

2. On lines 111-112, the authors write that "mice with SATB1 depleted from T cells had increased NUMBER of DP cells and decreased numbers of CD4+ and CD8+ single positive (SP) cells in the thymus". However, I believe that they mean an increased PERCENTAGE of DP cells instead, as shown in the figure; if there were an absolutely increased number of DP cells, it would be difficult to explain how the total cellularity of the thymus was slightly reduced.

3. In one respect, I agree with Reviewer 3 that the implication of "is controlled by" in the title is a little stronger than the actual effects that the authors so carefully document and dissect. The authors show convincingly that these differences are statistically and mechanistically significant, with major consequences, but across much of the genome *Satb1* appears to modulate and focus enhancer-promoter loops, but not necessarily to have an absolute deterministic effect on the whole "3D enhancer network" (Fig. 4). Could a slightly different word choice be used?

With regard to comments by the other reviewers, I have carefully read through those author responses too, and my own impression is that the authors have made exceptional efforts to address each comment in a scientifically responsible and accurate way, far beyond what is normally required. The added statistical support and some of the added measurements they have contributed indeed strengthen the paper. In the rebuttal, too, the detailed scholarly comparison the authors provide between the methods in this paper and the methods used in each of the previous relevant papers in the literature is exceptionally thorough and valuable. Overall, I find these author responses highly convincing.

Response to Reviewers' Comments

Nature Communications manuscript NCOMMS-21-25391C

"The 3D enhancer network of the developing T cell genome is shaped by SATB1"

We have now completed the revision process of our manuscript and we submit the revised version for your consideration. We would like to thank the Reviewers for their time in reviewing our manuscript but most importantly for their constructive comments that helped us improve our work. We hope that our responses, based on the experiments we have now completed, will adequately address the concerns raised.

Below please find a point-by-point response to the Reviewers' comments. (Original comments are in **black** and responses in **blue**).

Reviewer #2 (Remarks to the Author):

The authors have done an extraordinarily patient, thorough, and scholarly job of responding to the comments of the various reviewers. I am happy with all their responses to my own previous comments and greatly appreciate the increased clarity of the effects they demonstrate. There are just three word choices left that I have questions about. If they should indeed be changed, this can easily be done during production:

1. In the new (and very helpful) updated legends to Fig. 4e and Fig. S4g, did the authors mean counts >1 in the bottom RIGHT quadrants, or the bottom LEFT quadrants?
We thank the Reviewer for noticing this mistake. It should be the left quadrant and we changed it in the revised version of our manuscript.
2. On lines 111-112, the authors write that "mice with SATB1 depleted from T cells had increased NUMBER of DP cells and decreased numbers of CD4+ and CD8+ single positive (SP) cells in the thymus". However, I believe that they mean an increased PERCENTAGE of DP cells instead, as shown in the figure; if there were an absolutely increased number of DP cells, it would be difficult to explain how the total cellularity of the thymus was slightly reduced.
We again thank the Reviewer for noticing this. We indeed referred to percentages and not numbers. We have corrected it in the revised version of the manuscript.
3. In one respect, I agree with Reviewer 3 that the implication of "is controlled by" in the title is a little stronger than the actual effects that the authors so carefully document and dissect. The authors show convincingly that these differences are statistically and mechanistically significant, with major consequences, but across much of the genome *Satb1* appears to modulate and focus

enhancer-promoter loops, but not necessarily to have an absolute deterministic effect on the whole “3D enhancer network” (Fig. 4). Could a slightly different word choice be used?

We thank the Reviewer for this objective and valuable suggestion. We have decided to replace the word “control” with “shape” which more directly implies the involvement of SATB1 in spatial organization of the 3D enhancer network while being less deterministic.

With regard to comments by the other reviewers, I have carefully read through those author responses too, and my own impression is that the authors have made exceptional efforts to address each comment in a scientifically responsible and accurate way, far beyond what is normally required. The added statistical support and some of the added measurements they have contributed indeed strengthen the paper. In the rebuttal, too, the detailed scholarly comparison the authors provide between the methods in this paper and the methods used in each of the previous relevant papers in the literature is exceptionally thorough and valuable. Overall, I find these author responses highly convincing.

Once more we would like to thank the Reviewers for their helpful suggestions that ultimately increased the quality of our manuscript.

Sincerely,

Charalampos G. Spilianakis, Ph.D